# Partition First, Embed Later: Laplacian-Based Feature Partitioning for Refined Embedding and Visualization of High-Dimensional Data

Erez Peterfreund [1]   Ofir Lindenbaum [2]   Yuval Kluger [1 3 4]   Boris Landa [1 5]

## Abstract

Embedding and visualization techniques are essential for analyzing high-dimensional data, but they often struggle with complex data governed by multiple latent variables, potentially distorting key structural characteristics. This paper considers scenarios where the observed features can be partitioned into mutually exclusive subsets, each capturing a different smooth substructure. In such cases, visualizing the data based on each feature partition can better characterize the underlying processes and structures in the data, leading to improved interpretability. To partition the features, we propose solving an optimization problem that promotes graph Laplacian-based smoothness in each partition, thereby prioritizing partitions with simpler geometric structures. Our approach generalizes traditional embedding and visualization techniques, allowing them to learn multiple embeddings simultaneously. We establish that if several independent or partially dependent manifolds are embedded in distinct feature subsets in high-dimensional space, then our framework can reliably identify the correct subsets with theoretical guarantees. Finally, we demonstrate the effectiveness of our approach in extracting multiple low-dimensional structures and partially independent processes from both simulated and real data.

## 1. Introduction

Dimensionality reduction methods are crucial for extracting scientific insights from high-dimensional data by embedding it in a low-dimensional space, making it more suitable for visualization and downstream analysis. Such methods aim to reduce the dimensionality of a dataset while preserving its underlying structural characteristics. Specifically, given $N$ observations with $D$ features, $\{\boldsymbol{y}_i\}_{i=1}^{N} \subset \mathbb{R}^D$, standard techniques such as Laplacian Eigenmaps (Belkin & Niyogi, 2003), Diffusion Maps (Coifman & Lafon, 2006), t-distributed Stochastic Neighbor Embedding (tSNE) (Van der Maaten & Hinton, 2008), and UMAP (McInnes et al., 2018), first construct an $N \times N$ similarity graph between the given data points using all $D$ features. Then, they embed the $N$ graph nodes in a low-dimensional space where certain structural characteristics of the graph are preserved.

However, when the data's underlying structure is highly complex, standard methods may struggle to accurately capture and embed it in low dimensions. In particular, we are motivated by settings where several different groups of features are governed by distinct latent variables, each representing a unique low-dimensional substructure. If the number of unique latent variables is large, visualizing the data using all features with tSNE or UMAP can severely distort the data's underlying structure and fail to disentangle distinct latent variables (Kohli et al., 2021; Chari & Pachter, 2023). For Laplacian Eigenmaps and Diffusion Maps, there is significant redundancy in their representation that grows non-linearly with the number of underlying latent variables (Blau & Michaeli, 2017). This phenomenon is due to the eigenstructure of the graph Laplacian, where many eigenvectors are dependent and represent overlapping directions of variation. As a result, when the number of latent variables is large, the embedding dimension may need to substantially exceed it to fully capture the data's structure.

To address this challenge, we propose to *partition first, embed later*, namely a procedure for partitioning the features of a dataset into disjoint subsets, such that they exhibit simpler structures across the samples compared to the entire dataset. Specifically, given a prescribed number of partitions $K$, we propose to learn a partitioning of the $D$ features into $K$ mutually exclusive subsets and $K$ corresponding similarity graphs of size $N \times N$. Each similarity graph describes the pairwise affinities between all $N$ observations when considering only the corresponding subset of features. By creating

---

[1]Program in Applied Math, Yale University, New Haven, CT, USA [2]Faculty of Engineering, Bar Ilan University, Ramat Gan, Israel [3]Department of Pathology, Yale University, New Haven, CT, USA [4]Interdepartmental Program in Computational Biology and Bioinformatics, Yale University, New Haven, CT, USA [5]Department of Electrical Engineering, Yale University, New Haven, CT, USA. Correspondence to: <erezpeter@gmail.com>.

*Proceedings of the $42^{nd}$ International Conference on Machine Learning*, Vancouver, Canada. PMLR 267, 2025. Copyright 2025 by the author(s).

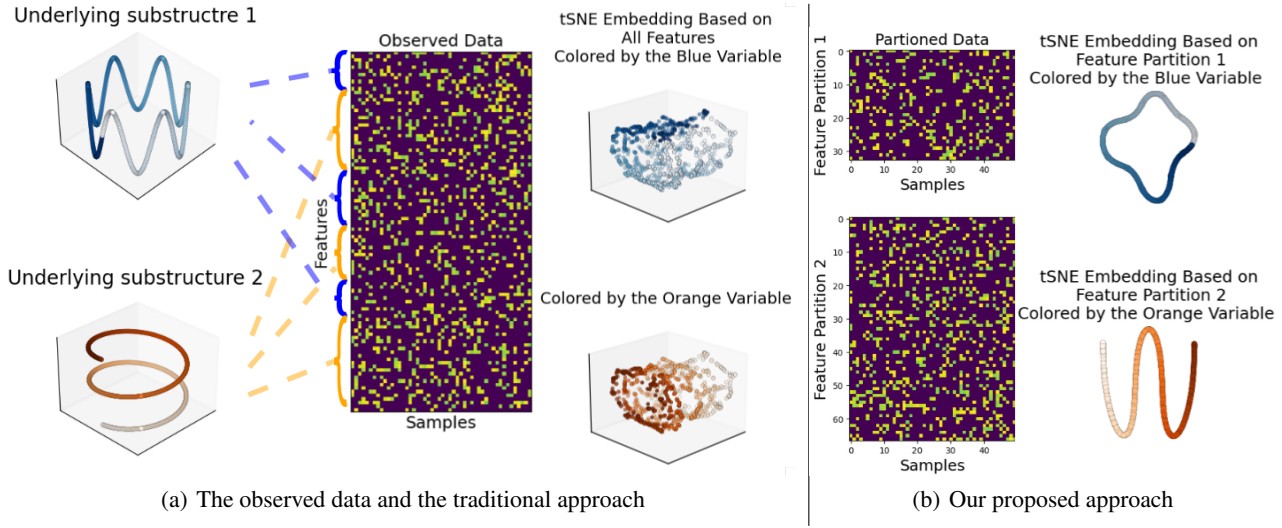

(a) The observed data and the traditional approach        (b) Our proposed approach

*Figure 1.* Illustration of our setting and proposed approach. (a) Simulated data consisting of two distinct substructures governed by independent latent variables: a closed loop (depicted in blue) and a helical curve (depicted in orange). Each substructure is embedded within a specific (unknown) subset of the observed features. The tSNE embedding, based on all features, is presented on the right in two panels: the top panel is colored according to the blue variable, while the bottom panel is colored according to the orange variable. (b) Our approach effectively divides the features into two subsets (K=2), with each subset consistently capturing information from only one of the underlying structures (blue and orange). Using these features, the tSNE embeddings for each feature partition can be used to recover the driving latent variables. The structure of each embedding (illustrated on the right), which is based solely on one subset of features, corresponds to one of the distinct latent variables (indicated by blue or orange).

separate graphs based on different sets of features, we can decompose complex data into simpler substructures, thereby better capturing their underlying patterns. Moreover, if each feature subset describes a low-dimensional geometry, it can be embedded and visualized more effectively than the entire dataset using all features. Multiple low-dimensional embeddings that capture different substructures in the data are often easier to interpret than a single embedding in higher-dimensional space; see Figure 1. For further discussions on selecting the number of partitions $K$ and validating that our approach leads to a simplified data representation, we refer the reader to Appendix G.

The setting where distinct feature subsets may contain unique geometric structures is widespread in applications. In hyperspectral imaging, different feature groups correspond to different wavelengths, which capture distinct chemical or physical phenomena of the observed materials or environment (Gowen et al., 2015; Khan et al., 2018). Similarly, in astrophysics, different spectral bands of electromagnetic radiation serve as feature groups in the data, capturing distinct astrophysical phenomena such as interstellar extinction and gravitational waves (Indebetouw et al., 2005; Burke-Spolaor et al., 2019). In cellular biology and genomics, different groups of genes are associated with distinct cellular processes (Sastry et al., 2019; Lamoureux et al., 2021), as exemplified in Section 5.2.

To find the best feature partitions and their associated simi-

larity graphs, we propose minimizing an objective function that relies on a certain graph Laplacian-based smoothness score. Our approach naturally extends the graph construction step in common embedding and visualization techniques, such as tSNE and Diffusion Maps, by enabling the simultaneous learning of multiple graphs from adaptively chosen feature partitions. We analyze this objective function in a setup where multiple low-dimensional manifolds are embedded in a high-dimensional space, possibly with partial dependence. We show that, in a suitable asymptotic regime with high dimension and large sample size, the minimum is attained only when the features are correctly partitioned into subsets that contain the individual embedded manifolds.

In Figure 1, we illustrate our approach on a high-dimensional dataset where two distinct low-dimensional latent structures are embedded within different, unknown feature subsets (depicted in orange and blue). The traditional tSNE embedding, constructed using all features, fails to reveal these underlying structures, resulting in a convoluted representation. In contrast, our method automatically uncovers and partitions the features associated with each latent structure, effectively disentangling the data. The resulting tSNE embeddings, generated separately for each partition, distinctly capture the corresponding latent variables, providing a more interpretable and structured visualization. Importantly, the latent variables governing each feature subset are not restricted to a single dimension but can represent

more complex low-dimensional geometries, reinforcing the need for feature partitioning before embedding.

## 2. Related Work

Feature partitioning has been explored in bi-clustering (Dhillon, 2001; Kluger et al., 2003), where the objective is to simultaneously cluster features and observations into subsets with similar entries or correlated rows/columns. Additionally, traditional clustering techniques, such as k-means (Lloyd, 1982) and spectral clustering (Ng et al., 2001), can be adapted for feature partitioning by treating each feature vector as a sample in $\mathbb{R}^N$. Our approach differs from such methods in that it does not require features to have similar values or be correlated in order to be grouped together. Instead, we group features based on shared latent variables, which we identify from the inferred geometric structure across the samples. This enables more general and flexible partitioning of features into groups. Indeed, we demonstrate in Section 5.1 and Appendix D that conventional clustering and bi-clustering methods fail to correctly partition features in several experiments with both simulated and real data.

A closely related research area is unsupervised feature selection, which focuses on identifying important data features (Lindenbaum et al., 2021; Shaham et al., 2022). These methods typically utilize a similarity graph constructed using all features to rank the features according to a Laplacian-based smoothness score. However, these methods only retain a subset of the features, which may not represent all the latent variables of the data. Moreover, these methods do not specify which selected features correspond to distinct substructures in the data. In contrast, our approach does not lose any information since it retains all features in the data. Moreover, it separates the features into interpretable groups that can be embedded and visualized more effectively.

Another related line of work focuses on decoupling data from a product-manifold (Zhang et al., 2021; He et al., 2023). This line of work is applicable to our setup if different feature groups are sampled from statistically independent manifolds. These methods attempt to deconvolve the eigenstructure of the graph Laplacian constructed using all the features to recover the graph Laplacians of the individual manifolds. One major limitation of such approaches is that the statistical error in the estimate of graph Laplacian-based quantities grows exponentially with the intrinsic dimension (Singer, 2006). Hence, these methods are prone to large errors if the product manifold is governed by many independent latent variables. In contrast, our approach decomposes the product manifold into individual manifolds with low intrinsic dimensions, allowing for a much more accurate construction of graph Laplacians for each manifold. Moreover, our approach supports cases where the subsets

of features forming the data are partially dependent (see Sections 4 and 5) — a scenario that cannot be addressed by existing product-manifold decoupling techniques.

An alternative approach was introduced by (Van der Maaten & Hinton, 2012), who extended the tSNE algorithm to generate multiple embeddings of the data. Their method constructs a single affinity matrix from the high-dimensional data and produces several low-dimensional embeddings whose integration (via a specialized combination rule) best approximates the original affinity matrix. In contrast, our approach directly constructs multiple affinity matrices by partitioning the features into groups that capture distinct low-dimensional substructures. Each of these feature groups can be used to generate a separate embedding, enabling a more accurate representation of the underlying geometry—often obscured when relying on a single global affinity matrix.

Finally, we mention Independent Subspace Analysis (ISA) and related techniques (Theis, 2006; Niu et al., 2010), which seek linear projections of the data into subspaces that are statistically independent or contain distinct structures. In high dimensions, these subspaces have many more degrees of freedom than the partitions learned by our approach, which can be prohibitive from a computational and statistical perspective (Bickel et al., 2018). Additionally, the analytical properties of these methods are not well understood in high-dimensional settings, especially when the substructures in the data are partially dependent. In contrast, our approach provides theoretical guarantees on accurate recovery of feature partitions in challenging high-dimensional scenarios, even under partial dependence between latent variables across partitions.

**Notations:** Bold symbols represent vectors or matrices. The $d$-th coordinate of a vector $\boldsymbol{y}$ is denoted by $(\boldsymbol{y})_d$ or $y_d$. For any $\boldsymbol{x} \in \mathbb{R}^D$ and $\boldsymbol{\omega} \in \mathbb{R}_+^D$, denote the weighted norm by $\|\boldsymbol{x}\|_{\boldsymbol{\omega}}^2 = \sum_{d=1}^{D} \omega_d(\boldsymbol{x})_d^2$.

## 3. Our Approach

This section presents the necessary background, motivation, and details of our proposed approach. First, in Section 3.1, we review the construction of similarity graphs from data and introduce the key concept of data smoothness defined over these graphs. Next, in Section 3.2, we formulate an optimization problem that adaptively partitions features into disjoint subsets and constructs corresponding graphs to maximize the total smoothness of the data. The proofs for this section are provided in Appendix I.1.

### 3.1. Traditional graph construction and the graph smoothness score

Given a set of observed data points $\boldsymbol{y}_1, \ldots, \boldsymbol{y}_N \in \mathbb{R}^D$, common embedding and visualization techniques initially

construct a graph representing their pairwise similarities. A popular choice of the graph affinity matrix $\boldsymbol{W} \in [0,1]^{N \times N}$ is a row-normalized Gaussian kernel defined by

$$W_{i,j} = \begin{cases} \frac{\exp\left(-\|\boldsymbol{y}_i - \boldsymbol{y}_j\|^2/\epsilon_i\right)}{\sum_{t=1}^{N} \exp\left(-\|\boldsymbol{y}_i - \boldsymbol{y}_t\|^2/\epsilon_i\right)} & i \neq j \\ 0 & i = j \end{cases} \quad (1)$$

for all $i, j = 1, \ldots, N$, where $\{\epsilon_i\}_{i=1}^{N} \subset \mathbb{R}_+$ represent bandwidth parameters controlling the effective neighborhood size around each point. Zeroing out the main diagonal aligns with tSNE's graph construction step and also makes the resulting affinity matrix $\boldsymbol{W}$ more robust to noise (Karoui, 2010; Landa et al., 2021). Other embedding techniques, such as Diffusion Maps and Laplacian Eigenmaps, construct the affinity matrix without zeroing out the main diagonal and with a single global bandwidth parameter.

The tSNE algorithm determines the bandwidth parameters $\epsilon_1, \ldots, \epsilon_N$ from (1) by imposing an entropy constraint on the rows of $\boldsymbol{W}$. This constraint is given by

$$\sum_{j=1}^{N} W_{i,j} \log W_{i,j} \leq -\log(\alpha), \quad (2)$$

for $i \in \{1, \ldots, N\}$, where $\alpha$ denotes a predefined global neighborhood size parameter known as the *perplexity*, typically set between 5 and 30. The tSNE graph construction enforces this constraint by adjusting the bandwidth parameters $\{\epsilon_i\}$ adaptively to the local sampling density. In contrast, other common graph construction techniques usually employ a global bandwidth constraint of the form $\epsilon_1 = \ldots = \epsilon_N$ discussed in (Singer et al., 2009).

Many feature selection methods (He et al., 2005; Lindenbaum et al., 2021) utilize the affinity matrix for identifying a meaningful subset of features. In particular, given some affinity matrix $\tilde{\boldsymbol{W}} \in [0,1]^{N \times N}$ that encodes the similarity between each pair of data points, these methods utilize a score of the form

$$S(\tilde{\boldsymbol{W}}, d, \{\boldsymbol{y}_i\}_{i=1}^{N}) = \sum_{i,j=1}^{N} \tilde{W}_{i,j} \left((\boldsymbol{y}_i)_d - (\boldsymbol{y}_j)_d\right)^2, \quad (3)$$

to measure the smoothness of the $d$th coordinate of the data over the graph. This score is often referred to as the Laplacian score (He et al., 2005) for certain choices of the affinity matrix $\tilde{\boldsymbol{W}}$. Summing this score over all coordinates, we define the graph smoothness score by

$$J(\tilde{\boldsymbol{W}}, \{\boldsymbol{y}_i\}_{i=1}^{N}) = \sum_{i,j=1}^{N} \tilde{W}_{i,j} \|\boldsymbol{y}_i - \boldsymbol{y}_j\|^2. \quad (4)$$

The following proposition shows that the affinity matrix $\boldsymbol{W}$ defined in (1) minimizes the graph smoothness score in (4), subject to constraints of perplexity and stochasticity for each row (while zeroing out the main diagonal).

**Proposition 3.1.** *The matrix $\boldsymbol{W} \in [0,1]^{N \times N}$ defined in (1) is a solution to*

$$\underset{\tilde{\boldsymbol{W}} \in [0,1]^{N \times N}}{\arg\min} \quad J(\tilde{\boldsymbol{W}}, \{\boldsymbol{y}_i\}_{i=1}^{N}), \quad (5)$$

*subject to the constraints $\tilde{W}_{i,i} = 0$, $\sum_{j=1}^{N} \tilde{W}_{i,j} = 1$ and $\sum_{j=1}^{N} \tilde{W}_{i,j} \log \tilde{W}_{i,j} \leq -\log(\alpha)$ for all $i \in \{1, \ldots, N\}$, where $\epsilon_1, \ldots, \epsilon_N \in \mathbb{R}_+$ from (1) are the minimum values that satisfy the entropy constraint.*

Similar results appeared in (Cuturi, 2013; Van Assel et al., 2024) under slightly different constraints.

If the data is sampled from a Riemannian manifold and the bandwidth parameters $\epsilon_1, \ldots, \epsilon_N \in \mathbb{R}_+$ are fixed as constants (ignoring the entropy constraints), then the following proposition characterizes the relation between the objective function in (5) and the manifold's intrinsic dimension.

**Proposition 3.2.** *Let $\mathcal{M} \subset \mathbb{R}^D$ be a smooth, compact, Riemannian manifold with intrinsic dimension $dim(\mathcal{M}) < D$. Suppose $\boldsymbol{y}_1, \ldots, \boldsymbol{y}_N \in \mathcal{M}$ are sampled independently from a smooth non-vanishing density $f$ over $\mathcal{M}$, and let $\boldsymbol{W}$ be defined as in (1). Then, for all $i \in \{1, \ldots, N\}$ and sufficiently small $\epsilon_1, \ldots, \epsilon_N \in \mathbb{R}_+$, we have*

$$\sum_{j=1}^{N} W_{i,j} \|\boldsymbol{y}_i - \boldsymbol{y}_j\|^2 \xrightarrow[N \to \infty]{a.s.} \frac{\epsilon_i}{2} \cdot dim(\mathcal{M}) + O(\epsilon_i^2). \quad (6)$$

Hence, for the affinity matrix $\boldsymbol{W}$ from (1) with sufficiently small bandwidth parameters, the objective function in (5) approximates the intrinsic dimension of the manifold multiplied by the sum of bandwidth parameters. This quantity is smaller for manifolds with lower intrinsic dimensions, i.e., simpler manifolds governed by fewer latent variables, or when the bandwidth parameters $\{\epsilon_i\}$ are smaller.

### 3.2. Feature Partitioning and Multi-Graph Learning

Given the dataset $\boldsymbol{y}_1, \ldots, \boldsymbol{y}_N \in \mathbb{R}^D$, we propose to partition the $D$ features into $K$ mutually exclusive subsets, each accompanied by a corresponding $N \times N$ affinity matrix. Intuitively, the data restricted to each subset of features should have a simpler geometric structure than the data across all features combined. To find the feature partitions and their associated graphs, we propose to minimize the sum of graph smoothness scores from (4) over all feature partitions.

Concretely, let $\{\tilde{\boldsymbol{\omega}}^{(1)}, \ldots, \tilde{\boldsymbol{\omega}}^{(K)}\} \in \{0,1\}^D$ be a feasible feature partitioning, where $\tilde{\omega}_d^{(k)} = 1$ if the $d$th feature is used within the $k$th partition, and $\tilde{\omega}_d^{(k)} = 0$ otherwise. The feature partitions cover all features and are mutually exclusive; namely, they satisfy $\sum_{k=1}^{K} \tilde{\omega}_d^{(k)} = 1$ for all $d \in \{1, \ldots, D\}$. The affinity matrices corresponding to the $K$

partitions are denoted by $\tilde{\boldsymbol{W}}^{(1)}, \ldots, \tilde{\boldsymbol{W}}^{(K)} \subset [0,1]^{N \times N}$. We now extend the graph smoothness score defined in (4) to support multiple feature partitions as

$$G(\{\tilde{\boldsymbol{W}}^{(k)}\}_{k=1}^K, \{\tilde{\boldsymbol{\omega}}^{(k)}\}_{k=1}^K, \{\boldsymbol{y}_i\}_{i=1}^N) \qquad (7)$$
$$= \sum_{k=1}^K \sum_{i,j=1}^N \tilde{W}_{i,j}^{(k)} \|\boldsymbol{y}_i - \boldsymbol{y}_j\|_{\tilde{\boldsymbol{\omega}}^{(k)}}^2,$$

recalling that $\|\boldsymbol{v}\|_{\tilde{\boldsymbol{\omega}}^{(k)}}^2 = \sum_{d=1}^D \tilde{\omega}_d^{(k)} v_d^2$ for any $\boldsymbol{v} \in \mathbb{R}^D$. Notably, when all the affinity matrices are the same ($\tilde{\boldsymbol{W}}^{(1)} = \ldots = \tilde{\boldsymbol{W}}^{(K)} = \tilde{\boldsymbol{W}}$), then this score coincides with the score defined in (4), i.e., $G(\{\tilde{\boldsymbol{W}}^{(k)}\}_{k=1}^K, \{\tilde{\boldsymbol{\omega}}^{(k)}\}_{k=1}^K, \{\boldsymbol{y}_i\}_{i=1}^N) = J(\tilde{\boldsymbol{W}}, \{\boldsymbol{y}_i\}_{i=1}^N)$. We define the following optimization problem to determine the feature partitions and corresponding affinity matrices.

**Problem 3.3.**

$$\min_{\left\{\tilde{\boldsymbol{W}}^{(k)}\right\}, \left\{\tilde{\boldsymbol{\omega}}^{(k)}\right\}} G(\{\tilde{\boldsymbol{W}}^{(k)}\}_{k=1}^K, \{\tilde{\boldsymbol{\omega}}^{(k)}\}_{k=1}^K, \{\boldsymbol{y}_i\}_{i=1}^N), \quad (8)$$

*under the constraints* $\sum_{k=1}^K \tilde{\omega}_d^{(k)} = 1$, $\sum_{j=1}^N \tilde{W}_{i,j}^{(k)} = 1$, $\tilde{W}_{i,i}^{(k)} = 0$, *and* $\sum_{j=1}^N \tilde{W}_{i,j}^{(k)} \log\left(\tilde{W}_{i,j}^{(k)}\right) \leq -\log(\alpha)$, *for* $i = 1, \ldots, N$, $k = 1, \ldots, K$, *and* $d = 1, \ldots, D$, *where* $\alpha$ *is a perplexity parameter.*

Next, we characterize the solutions to Problem 3.3.

**Proposition 3.4.** *There exists an optimal partitioning solution* $\{\boldsymbol{\omega}^{(k)}\}_{k=1}^K$ *and corresponding affinity matrices* $\{\boldsymbol{W}^{(k)}\}_{k=1}^K$ *that solve Problem 3.3 and are of the form*

$$W_{i,j}^{(k)} = \begin{cases} \dfrac{\exp\left(-\|\boldsymbol{y}_i - \boldsymbol{y}_j\|_{\boldsymbol{\omega}^{(k)}}^2 / \epsilon_{k,i}\right)}{\sum_{t \neq i} \exp\left(-\|\boldsymbol{y}_i - \boldsymbol{y}_t\|_{\boldsymbol{\omega}^{(k)}}^2 / \epsilon_{k,i}\right)} & if\ i \neq j \\ 0 & else \end{cases} (9)$$

$$\omega_d^{(k)} = \begin{cases} 1 & if\ k = \tilde{k}\ for\ some\ \tilde{k} \in \Omega(d) \\ 0 & else \end{cases}, \quad (10)$$

$$\Omega(d) = \underset{k \in \{1, \ldots, K\}}{\arg\min}\ S(\boldsymbol{W}^{(k)}, d, \{\boldsymbol{y}_i\}_{i=1}^N), \quad (11)$$

*for* $d = 1, \ldots, D$ *and* $i, j = 1, \ldots, N$, *where the bandwidth parameters* $\{\epsilon_{k,i}\} \subset \mathbb{R}_+$ *are the minimum values that satisfy the entropy constraints in Problem 3.3, and* $S$ *is the Laplacian-type score defined in* (3).

We see that the affinity matrix $\boldsymbol{W}^{(k)}$ is simply a row-normalized Gaussian kernel constructed from the features in the $k$th partition, analogously to the traditional construction in (1) using all features. Additionally, the $d$th feature of the data is assigned to the $k$th partition if it is smoother with respect to the affinity matrix $\boldsymbol{W}^{(k)}$ than the other affinity matrices, as measured by the Laplacian-type score in the

right-hand side of (11). Thus, our approach naturally extends the traditional graph construction techniques discussed in Section 3.1 by forming multiple graphs from disjoint feature partitions, which are optimized to minimize the total smoothness of the features across their associated graphs.

To further motivate Problem 3.3, consider data formed by concatenating $K$ feature groups, where the features in each group were sampled independently from a different Riemannian manifold. In this case, under the optimal solution from Proposition 3.4, the objective function in (8) converges to a weighted sum of the manifolds' intrinsic dimensions (see Proposition 3.2). The weights depend on the corresponding bandwidth parameters $\{\epsilon_{k,i}\}$, which are set to enforce the negative entropy constraints. However, for an incorrect partitioning — where features from different manifolds are mixed in each partition — the intrinsic dimension of the data in each partition would be higher. Consequently, the required bandwidth parameters that satisfy the entropy constraints would be larger. Overall, we can interpret the optimization problem as dividing the feature space into partitions whose intrinsic dimensions are as small as possible.

We now consider the task of solving Problem 3.3. A natural strategy is alternating minimization, where the objective function is minimized over the graph parameters while keeping feature partitions fixed, and vice versa. The solution to each step of this alternating minimization is given explicitly by Proposition 3.4. Unfortunately, due to the binary nature of the feature partitions, this procedure is sensitive to local minima. To address this issue, in Appendix C we introduce a regularized variant of the objective function (see Problem C.1) that produces a soft assignment of features instead of hard assignments into partitions (see Proposition C.2). Our proposed algorithm (see Algorithm 1 in Appendix C) solves several instances of the regularized problem sequentially, each with a reduced regularization parameter, initialized with the solution to the previous instance of the regularized problem. In the final step, the regularization parameter reaches zero, thereby minimizing the original unregularized problem. This sequence of solutions to the regularized problems is less likely to get stuck in a local minimum compared to solving the unregularized problem directly; see Appendix C for more details.

## 4. Analysis

In this section, we analyze a variant of the feature partitioning problem (see Problem 3.3) under a data generative model with $K$ partially dependent subsets of features. We investigate the objective function landscape in a high-dimensional asymptotic regime and establish that its minimizer recovers the correct feature partitions. Additionally, numerical results in Appendix E demonstrate that the landscape of this variant closely mirrors that of the original problem. The

proofs for this section are provided in Appendix I.2.

We define a variant of the graph smoothness score as

$$\tilde{G}\left(\{\tilde{\boldsymbol{W}}^{(k)}\}_{k=1}^K, \{\tilde{\boldsymbol{\omega}}^{(k)}\}_{k=1}^K, \{\boldsymbol{y}_i\}_{i=1}^N\right) \quad (12)$$

$$= \sum_{k=1}^K \sum_{i,j=1}^N \tilde{W}_{i,j}^{(k)} \cdot \frac{\|\boldsymbol{y}_i - \boldsymbol{y}_j\|_{\tilde{\boldsymbol{\omega}}^{(k)}}^2}{(1/D)\sum_d \tilde{\omega}_d^{(k)}}.$$

This variant adjusts the Laplacian-based score by normalizing the portion related to each partition based on the average number of features used within that partition. This adjustment accounts for the changes made to the optimization problem, which we define next. As we will show, these modified formulations will retain key properties of the original problem.

Building on this score, we propose to analyze a simplified variant of Problem 3.3, aimed at identifying the feature partitions and their associated affinity matrices. This version adopts a regularized minimization framework that incorporates the negative entropy constraint directly into the objective, enabling a more tractable analysis.

**Problem 4.1.** *Consider the optimization problem defined by*

$$\min_{\left\{\tilde{\boldsymbol{W}}^{(k)}\right\}_k, \left\{\tilde{\boldsymbol{\omega}}^{(k)}\right\}_k} \tilde{G}\left(\{\tilde{\boldsymbol{W}}^{(k)}\}_{k=1}^K, \{\tilde{\boldsymbol{\omega}}\}_{k=1}^K, \{\boldsymbol{y}_i\}_{i=1}^N\right) \quad (13)$$

$$+ \epsilon \sum_{k=1}^K \sum_{i,j=1}^N \tilde{W}_{i,j}^{(k)} \log\left(\tilde{W}_{i,j}^{(k)}\right)$$

*with the following constraints* $\sum_{k=1}^K \tilde{\omega}_d^{(k)} = 1$, $\sum_{j=1}^N \tilde{W}_{i,j}^{(k)} = 1$, $\tilde{W}_{i,i}^{(k)} = 0$, *for* $i = 1, \ldots, N$, $k = 1, \ldots, K$ *and* $d = 1, \ldots, D$.

We characterize the affinity matrices that minimize this objective in the following corollary.

**Corollary 4.2.** *Let* $\{\tilde{\boldsymbol{\omega}}^{(k)}\}$ *be a partitioning that satisfies the constraints in Problem 4.1. Then, graph affinity matrices that minimize* (13) *while fixing the partitioning parameters* $\{\tilde{\boldsymbol{\omega}}^{(k)}\}$ *are*

$$W_{i,j}^{(k)} = \begin{cases} \dfrac{\exp\left(-\dfrac{\|\boldsymbol{y}_i - \boldsymbol{y}_j\|_{\tilde{\boldsymbol{\omega}}^{(k)}}^2}{\epsilon \cdot (1/D)\sum_{d=1}^D \tilde{\omega}_d^{(k)}}\right)}{\sum_{t=1}^N \exp\left(-\dfrac{\|\boldsymbol{y}_i - \boldsymbol{y}_t\|_{\tilde{\boldsymbol{\omega}}^{(k)}}^2}{\epsilon \cdot (1/D)\sum_{d=1}^D \tilde{\omega}_d^{(k)}}\right)} & \text{if } i \neq j \\ 0 & \text{else} \end{cases} \quad (14)$$

*for* $i, j = 1, \ldots, N$ *and* $k = 1, \ldots, K$.

Note that the effective bandwidth parameter of each $\boldsymbol{W}^{(k)}$ adapt according to the number of features in its corresponding partition $\boldsymbol{\omega}^{(k)}$, for $k = 1, \ldots, K$.

Next, we derive the asymptotic value of the objective under a high-dimensional regime. We consider a generative data model in which the observed space is based on subsets of features that exhibit partial dependence. Let $\mathcal{M}_1 \subset \mathbb{R}^{d_1} \ldots, \mathcal{M}_{K+1} \subset \mathbb{R}^{d_{K+1}}$ be latent smooth compact Riemannian manifolds with corresponding smooth, non-vanishing densities $f_1, \ldots, f_{K+1}$. The latent samples $\{\boldsymbol{x}_i^{(s)}\}_{i=1}^N \in \mathcal{M}_s$ are independently sampled according to $f_s$ for $s = 1, \ldots, K+1$. The observed data points, denoted by $\{\boldsymbol{y}_i\}_{i=1}^N \in \mathbb{R}^D$, are constructed by

$$\boldsymbol{y}_i^T = \left((\boldsymbol{y}_i^{(1)})^T, \ldots (\boldsymbol{y}_i^{(K)})^T\right) \in \mathbb{R}^D, \quad (15)$$

$$\boldsymbol{y}_i^{(s)} = \boldsymbol{P}^{(s)}\begin{bmatrix} \boldsymbol{x}_i^{(s)} \\ \boldsymbol{x}_i^{(K+1)} \end{bmatrix} \in \mathbb{R}^{D_s} \qquad s = 1, \ldots, K,$$

for $i = 1, \ldots, N$, where $D = \sum_{s=1}^K D_s$, and the entries of $\boldsymbol{P}^{(s)} \in \mathbb{R}^{D_s \times (d_k + d_{K+1})}$ are independently sampled from $\mathcal{N}(0, 1/D_s)$ for $s = 1, \ldots, K$.

To establish a direct correspondence between each partition $\boldsymbol{\omega}^{(k)}$ and the true $K$ partitions used in the construction of the observation space, we denote $\boldsymbol{\omega}^{(k)} = (\boldsymbol{\omega}^{(k,1)}, \ldots, \boldsymbol{\omega}^{(k,K)})$ for $k = 1, \ldots, K$. We define the relative proportion with respect to each true partition by $\sum_{d=1}^{D_s} \omega_d^{(k,s)}/D_s \to p_s^{(k)} \in (0, 1)$, for any $k, s \in \{1 \ldots, K\}$, where $\sum_{k=1}^K p_s^{(k)} = 1$ for all $s$.

The next theorem characterizes the objective function of Problem 4.1 in a high-dimensional asymptotic regime where $D, N \to \infty$ and $D/\log(N) \to \infty$. We assume that the relative size of each feature subset satisfies $D_s/D \to \beta_s \in (0, 1)$ for any $s \in \{1, \ldots, K\}$, where $\boldsymbol{\beta} \in (0, 1)^K$ and $\sum_{s=1}^K \beta_s = 1$.

**Theorem 4.3.** *There exists* $\bar{\epsilon}(\mathcal{M}, f) \leq 1$ *such that for any* $\epsilon < \bar{\epsilon}$ *and* $p_s^{(k)} \in [\sqrt{\epsilon}, 1 - (K-1)\sqrt{\epsilon}]$, *any partitioning solution* $\{\boldsymbol{\omega}^{(k)}\}$ *(obeying Problem 4.1's constraints) satisfies*

$$\min_{\left\{\tilde{\boldsymbol{W}}^{(k)}\right\}} \frac{1}{\epsilon N}\left(\tilde{G}\left(\left\{\tilde{\boldsymbol{W}}^{(k)}\right\}, \left\{\boldsymbol{\omega}^{(k)}\right\}, \{\boldsymbol{y}_i\}\right)\right) \quad (16)$$

$$+ \epsilon\left(\sum_{k=1}^K \sum_{i,j=1}^N \tilde{W}_{i,j}^{(k)} \log(\tilde{W}_{i,j}^{(k)})\right) + K \log(N-1)$$

$$\overset{a.s.}{\underset{N,D\to\infty}{=}} \sum_{k=1}^K \frac{dim(\mathcal{M}_{K+1})}{2} \log\left(\frac{\sum_{s=1}^K p_s^{(k)}}{\sum_{t=1}^K p_t^{(k)}\beta_t}\right) \quad (17)$$

$$+ \sum_{k,s=1}^K \frac{dim(\mathcal{M}_s)}{2} \log\left(\frac{p_s^{(k)}}{\sum_{t=1}^K p_t^{(k)}\beta_t}\right)$$

$$+ K\sum_{s=1}^{K+1}\left(h_s(f_s) - \frac{dim(\mathcal{M}_s)\log(\pi\epsilon)}{2}\right) + O(\sqrt{\epsilon}),$$

*where* $h_s(f_s) = -\int_{\boldsymbol{z}\in\mathcal{M}_s} f_s(\boldsymbol{z})\log f_s(\boldsymbol{z})d\boldsymbol{z}$ *is the differential entropy of the density* $f_s$ *over* $\mathcal{M}_s$.

Data Composed of Two Independent Substructures

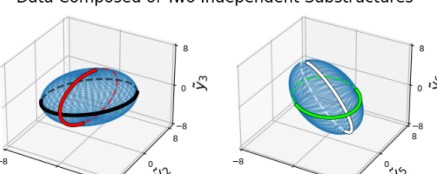

Data Composed of Two Partially Dependent Substructures

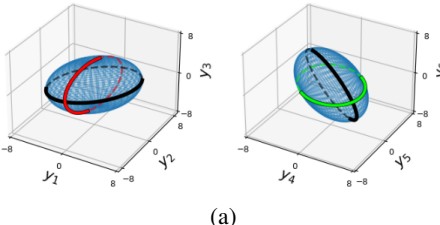

(a)

Absolute Value of Correlation Matrices

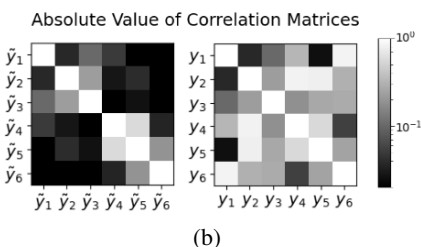

(b)

*Figure 2.* Product of 2-dimensional ellipsoids. (a) Simulated data including $N = 1000$ samples in $\mathbb{R}^6$, where the first three coordinates describe one ellipsoid and the last three describe another, and both parameterized by polar angles depicted by colored ellipses. In the first scenario, the ellipsoids are independent (Top). In contrast, in the second, they are partially dependent (Bottom) since one polar angle is shared between the two ellipsoids (the black ellipse). (b) The absolute value of the feature–feature correlation matrix of the two datasets. For clarity, we slightly abuse notation by using $y_d$, or correspondingly $\tilde{y}_d$, to denote the $d$th coordinate (i.e., feature) of the data.

Evidently, only the first two terms in (17) are affected by the feature partitions, while the rest depend on $\epsilon$, the manifolds' properties, and their densities. In the following theorem, we show that the minimizer of (17) in the case of two partitions ($K = 2$) accurately separates the data features as $\epsilon \to 0$.

**Theorem 4.4.** *Let $K = 2$, and define $f : (0,1)^2 \to \mathbb{R}$ by*

$$f(p_1, p_2) = \sum_{k=1}^{K} \frac{dim(\mathcal{M}_{K+1})}{2} \log \left( \frac{\sum_{s=1}^{K} p_s^{(k)}}{\sum_{t=1}^{K} p_t^{(k)} \beta_t} \right) \quad (18)$$

$$+ \sum_{k,s=1}^{K} \frac{dim(\mathcal{M}_s)}{2} \log \left( \frac{p_s^{(k)}}{\sum_{t=1}^{K} p_t^{(k)} \beta_t} \right),$$

*where $p_1^{(1)} = p_1$ and $p_2^{(1)} = p_2$ and therefore $p_1^{(2)} = 1 - p_1^{(1)}$, $p_2^{(2)} = 1 - p_2^{(2)}$. Then, the limiting minimizer $(p_1^*, p_2^*) = \lim_{\epsilon \to 0} \arg \min_{p_1, p_2 \in [\sqrt{\epsilon}, 1-\sqrt{\epsilon}]^2} f(p_1, p_2)$ is either $(0, 1)$ or $(1, 0)$.*

*Table 1.* Performance of different methods for partitioning the features for the two scenarios shown in Figure 2. The partitioning error is the number of coordinates assigned incorrectly, averaged over 100 randomized experiments, and the standard deviation is shown in parentheses. The correct partition should separate the first three coordinates from the last three, thereby correctly capturing the two ellipsoid structures.

| METHOD\DATA | INDEPENDENT | PARTIALLY DEPENDENT |
|---|---|---|
| SPECTRAL CO-CLUSTERING | 1.85 (0.91) | 1.95 (0.8) |
| SPECTRAL BI-CLUSTERING | 1.99 (0.59) | 2.15 (0.78) |
| K-MEANS (ON FEATURES) | 1.77 (0.55) | 2.22 (0.74) |
| SPECTRAL CLUSTERING (ON FEATURES) | 1.71 (0.94) | 2.2 (0.73) |
| **FP (OURS)** ALGORITHM 1 | **0.** (0.) | **0.** (0.) |

To conclude, we proposed a variant of the feature partitioning problem in Problem 4.1 and analyzed its loss landscape in an asymptotic regime. We considered a data-generating process where the features are composed of $K$ partially dependent feature groups, making the partitioning task nontrivial. Finally, we showed that in this nontrivial case, the loss is minimized when the partitioning solution aligns with the ground truth when $K = 2$. In Appendix E, we show the close resemblance between the loss landscapes of the examined problem and Problem 3.3 using a synthetic dataset.

## 5. Experiments

This section highlights our approach and its advantages through synthetic and real data. In Section 5.1, we illustrate and quantify its effectiveness in a controlled environment using artificial data. Next, in Sections 5.2 and 5.3, we show the applicability of our approach to two real-world high-dimensional biological datasets, yielding enhanced visualizations that are consistent with known biological processes. Finally, in Appendix D, we use video data to demonstrate that our approach can enhance the ability to visualize and correctly analyze complex datasets.

### 5.1. Product of 2-Dimensional Ellipsoids

This experiment demonstrates the effectiveness of our approach in decomposing the data into different substructures in a controlled environment using synthetic datasets. We simulated two datasets in $\mathbb{R}^6$: one with two independent feature subsets and another with partially dependent ones, and the task is to retrieve these subsets ($K = 2$). In the independent case, samples are drawn independently from two rotated $2D$ ellipsoids in $\mathbb{R}^3$, and their coordinates are

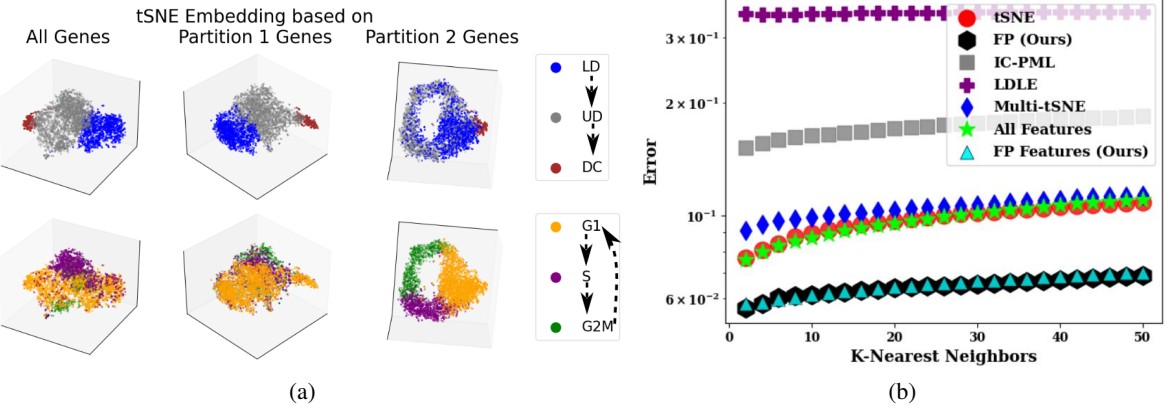

*Figure 3.* Partitioning the genes in scRNA-seq data to discover distinct salient cellular processes. (a) The figure includes the tSNE embeddings using all genes (left) versus genes from partition 1 (middle) and partition 2 (right). Top: Cells colored by cell type, with partition 1 capturing the LD/UD to DC developmental trajectory. Bottom: Cells colored by cell cycle phase, with partition 2 revealing cycling progression. (b) A quantitative comparison of embeddings generated by different algorithms, assessing their correspondence with the two latent processes governing the data via $k$-nearest neighbor error. The results show that our partitioning approach most effectively reveals the underlying structure, with each process captured in a distinct partition. See Appendix F.2 for details on the error metric.

stacked to form vectors in $\mathbb{R}^6$. In the partially dependent case, the generating process is the same, except that one of the polar angles is shared between the ellipsoids. The independent (Top) and partially dependent (Bottom) datasets are visualized in Figure 2(a), and their correlation matrices are shown in Figure 2(b). See Appendix F.1 for further details.

In Table 1, we assess the performance of several methods adapted to partition the features in the two scenarios described above. The recorded error is the number of coordinates (out of six) assigned incorrectly, averaged over 100 randomized experiments. The results demonstrate that traditional clustering or bi-clustering approaches cannot be utilized to solve the problem we consider here. Indeed, while these approaches perform slightly better in the independent case than in the (more challenging) dependent case, they still incur substantial errors in both cases. In contrast, our proposed approach consistently recovers the correct partitions across all experiments.

### 5.2. Dermal scRNA-seq Data

In this experiment, we demonstrate that our approach effectively separates co-occurring biological processes in single-cell RNA-sequencing data, enabling clear visualization of each process, thereby improving scientific discovery capabilities. Specifically, we analyze embryonic dermal cells from mouse skin (Qu et al., 2024), which exhibit two intertwined processes: cell cycle progression (G1, G2M, and S phases) and cell type development (from lower dermal (LD) to upper dermal (UD) to dermal condensate (DC) cells).

The dataset comprises $N = 5572$ cells and $D = 5000$ features, each representing the expression level of a gene

within each cell after standard processing is applied, including variability-based feature selection; see Appendix F.2. The preprocessing is similar to that employed in (Qu et al., 2024). In that study, the authors examined the genes through a gene similarity graph derived from the cells' affinity matrix and their gene profiles. In contrast, our approach partitions the genes according to their graph structure, generating a separate graph for each subset. To motivate the use of our approach, we note that each of these processes is associated with different subsets of genes, many of which are well-characterized (Tirosh et al., 2016).

Using our approach, we partition the genes into two groups ($K = 2$). In Figure 3, we compare the tSNE embedding based on all features with separate tSNE embeddings based on each extracted partition. The embedding based on all genes reveals the cell *type* development (top) but not the cell *cycle phase* progression (bottom). In contrast, the partition-based embeddings reveal both processes: partition 1 captures the cell *type* structure, while partition 2 reveals the cell *cycle* structure. We further validate these findings in Appendix F.2. First, Figure 9 demonstrates similar results using alternative embedding techniques, underscoring the importance of partitioning. Then, in Figure 10, we repeat the task with 10,000 features. While the added genes may introduce variability, our approach still recovers key substructures. Finally, in Figure 11, we repeat the experiment on a different biological dataset with similar characteristics and observe consistent results.

In Figure 3(b), we quantitatively compare the embeddings generated based on our extracted partitions with those produced by alternative techniques. The evaluated approaches

include: 1) tSNE embedding using all features; 2) two tSNE embeddings based on our partitions ('FP'); 3) two embeddings of IC-PML (He et al., 2023); 4) a single embedding of LDLE (Kohli et al., 2021); 5) two embeddings of Multi-tSNE (Van der Maaten & Hinton, 2012); 6) raw data features; and 7) raw data features after partitioning ('FP Features'). For each approach, we assess the correspondence between the structure of the provided embeddings and the latent variables. Specifically, the metric assesses how well the latent variables are reflected within the local neighborhoods of the embeddings for different neighborhood sizes (k-Nearest Neighbors), providing a comprehensive view of the embeddings' quality; see Appendix F.2 for further details. Our approach consistently yields the lowest error, outperforming existing methods when evaluated either on the raw feature partitions or on their tSNE embeddings.

Importantly, the partitions obtained by our approach are consistent with known biological phenomena: partition 1 includes the genes Sox2 and Foxd1, expressed in the DC cell type, and the genes Ptch1 and Lef1, expressed in both UD and DC cell types (Qu et al., 2022). In contrast, partition 2 contains all 86 cell-cycle genes from the Seurat R package that were retained in our dataset after preprocessing (Tirosh et al., 2016). Overall, our approach effectively separates the genes according to the two underlying biological processes.

### 5.3. Liver scRNA-seq Data

In this experiment, we demonstrate our approach on biological data whose features are transformed to enable effective partitioning. Specifically, we consider a single-cell RNA-sequencing liver lobule dataset (Droin et al., 2021), characterized by two independent latent variables, with multiple genes influenced by both of them. Here, the raw features (genes) cannot be partitioned into partially or fully independent subsets, making a suitable transformation necessary.

The dataset comprises $N = 6889$ cells and $D = 2000$ features, each representing the expression level of a gene within each cell after standard processing is applied; see Appendix F.3 for details. The dataset's structure is governed by two latent variables: spatial zonation, associated with the cells' locations along the liver layers (1–8); and the circadian cycle, associated with the time each cell was sampled (ZT 0, 6, 12, and 18) within a 24-hour cycle.

Droin et al. (2021) modeled the expression of each gene across cells based on the cells' liver layer and sampling time, with prior knowledge incorporated into the model. They showed that the two latent variables govern overlapping subsets of genes. To address this challenge, we decorrelated the features using principal component analysis (PCA), a standard preprocessing step in the field (Andrews et al., 2021). We then applied our approach to the transformed data, partitioning the new features into $K = 2$ groups.

In Figure 12, we compare the tSNE embeddings generated based on each group with a standard tSNE embedding based on all features. The standard embedding provides a single visualization where both latent variables are partially visible. Specifically, the circadian cycle is reflected by four clusters corresponding to the four sampling time points, although the cyclic structure is less evident. In contrast, the embedding based on partition 2 reveals both the clusters and the cyclic structure. Additionally, while zonation is partially visible within each cluster in the standard embedding, it is less evident than the clear, progressive pattern seen in partition 1's embedding. The latter aligns with the zonation layers effectively. These results demonstrate the benefit of our approach in revealing distinct latent variables.

## 6. Discussion

This paper presents a novel computational framework to partition the features of a high-dimensional dataset into subsets with simpler underlying structures. Embedding and visualizing the data using the features of each subset can effectively reveal these simple structures, which are obscured in the embedding that uses all the features. We demonstrate the effectiveness of our approach both analytically and empirically using simulated and real-world data, even when the features consist of partially dependent subsets. In Appendix G, we discuss practical considerations, e.g., the selection of the number of feature partitions. Additionally, we include an experiment on a subset of the COIL-20 dataset in Appendix H, demonstrating the effectiveness of our approach in scenarios where the feature separability assumption may not hold.

Our approach addresses some of the criticisms of low-dimensional embeddings raised in (Chari & Pachter, 2023). In particular, it enables more accurate embedding of substructures with low intrinsic dimensions through the proposed partitioning, as demonstrated in our experiments. While there is no guarantee that these substructures can be faithfully visualized in two or three dimensions, the resulting partitions remain valuable for a range of analytical tasks beyond visualization. Thus, by focusing on partitioning the data into simpler structures, our approach is inherently less susceptible to such criticism.

We identify several promising directions for future research. First, developing fully automated methods to determine the optimal number of partitions $K$ would enhance the practicality of our approach. Second, exploring more efficient optimization techniques for Problem 3.3 and deriving convex relaxations could significantly improve computational efficiency and solution quality. Finally, extending the method to account for uninformative or nuisance features presents an important avenue for broadening its applicability.

## Acknowledgements

The research conducted by O.L. is funded by the MOST grant 207892. The research conducted by Y.K. is funded by the National Institutes of Health (R01GM131642, UM1PA051410, R33DA047037, U54AG076043, U54AG079759, U01DA053628, P50CA121974).
We would like to thank Peggy Myung, Ruiqi Li, Rihao Qu, and Junchen Yang for their valuable help with understanding and preprocessing the biological datasets used in this study.

## Impact Statement

This paper presents work whose goal is to advance the field of Machine Learning. There are many potential societal consequences of our work, none of which we feel must be specifically highlighted here.

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

## A. Code Repository

The code for this paper is given in https://github.com/erezpeter/Feature_Partition.git

## B. The Algorithm

---

**Algorithm 1** Feature Partitioning

---

1: **Input:** Data samples $\boldsymbol{y}_1, \ldots, \boldsymbol{y}_N \in \mathbb{R}^D$; Number of partitions $K \in \mathbb{N}$; Number of iterations $T$.
2: Initialize the feature partition vectors $\boldsymbol{\omega}^{(1)}, \ldots, \boldsymbol{\omega}^{(K)}$ according to

$$\omega_d^{(k)} = \frac{\tilde{\omega}_d^{(k)}}{\sum_{s=1}^{K} \tilde{\omega}_d^{(s)}}, \quad \text{for } k = 1, \ldots, K; \ d = 1, \ldots, D,$$

where $\tilde{\boldsymbol{\omega}}^{(1)}, \ldots, \tilde{\boldsymbol{\omega}}^{(K)} \overset{\text{i.i.d.}}{\sim} \text{Uniform}[0,1]^D$.
3: Set $\delta \leftarrow \delta_{\text{init}}$ according to (24).
4: **for** $t = 1$ to $T$ **do**
5:     **if** $t = T$ **then**
6:         Set $\delta \leftarrow 0$.
7:     **end if**
8:     **while** the score $G_{reg}$ from (19) decreases **do**
9:         (a) Update the affinity matrices $\boldsymbol{W}^{(1)}, \ldots, \boldsymbol{W}^{(K)} \in [0,1]^{N \times N}$ according to (9).
10:        (b) Update feature partition vectors $\boldsymbol{\omega}^{(1)}, \ldots, \boldsymbol{\omega}^{(K)} \in [0,1]^D$ according to (21).
11:        (c) Compute the new score $G_{reg}$ according to (19).
12:     **end while**
13:     Update $\delta \leftarrow \delta/2$.
14: **end for**
15: **Return** $\boldsymbol{\omega}^{(1)}, \ldots, \boldsymbol{\omega}^{(K)}$.

---

**Note:** In our simulations, we run this algorithm multiple times with different random initializations and return the solution with the lowest score.

---

## C. Algorithmic Details

This section presents an algorithm for solving the feature partitioning problem described in Problem 3.3. The algorithm solves a sequence of regularized versions of this problem sequentially, gradually reducing the regularization until the problem aligns with the original problem. In this section, we derive the algorithm update formulas, compare it to a naive solution to Problem 3.3, and analyze its computational complexity. The proofs for this section are given in Appendix I.3.

A direct approach for solving Problem 3.3 naturally arises from Proposition 3.4 in the form of an alternating minimization procedure, by minimizing the objective function over the partition parameters while keeping the affinity matrices fixed, and vice-versa. However, such a technique is susceptible to converge to local minima due to the binary nature of the partition parameters, as illustrated in Figure 4 (rightmost column). In order to address this issue, we define a regularized version of the problem that allows soft partitions $\{\boldsymbol{\omega}^{(k)}\} \subset [0,1]^D$. The new objective function uses an entropic regularization term for the weights $\{\boldsymbol{\omega}^{(k)}\}$ with a regularization parameter $\delta \in \mathbb{R}_+$. Specifically, we define the regularized objective function as

$$
\begin{aligned}
G_{reg}(\delta, \{\tilde{\boldsymbol{W}}^{(k)}\}_{k=1}^K, \{\tilde{\boldsymbol{\omega}}^{(k)}\}_{k=1}^K, \{\boldsymbol{y}_i\}_{i=1}^N) \ = \ & G(\{\tilde{\boldsymbol{W}}^{(k)}\}, \{\tilde{\boldsymbol{\omega}}^{(k)}\}, \{\boldsymbol{y}_i\}_{i=1}^N) \\
& + \delta \left( D \log(K) + \sum_{d=1}^{D} \sum_{k=1}^{K} \tilde{\omega}_d^{(k)} \log(\tilde{\omega}_d^{(k)}) \right),
\end{aligned}
\tag{19}
$$

where $G$ is the objective function of the unregularized problem defined in (7).

The new regularized optimization problem is defined as follows.

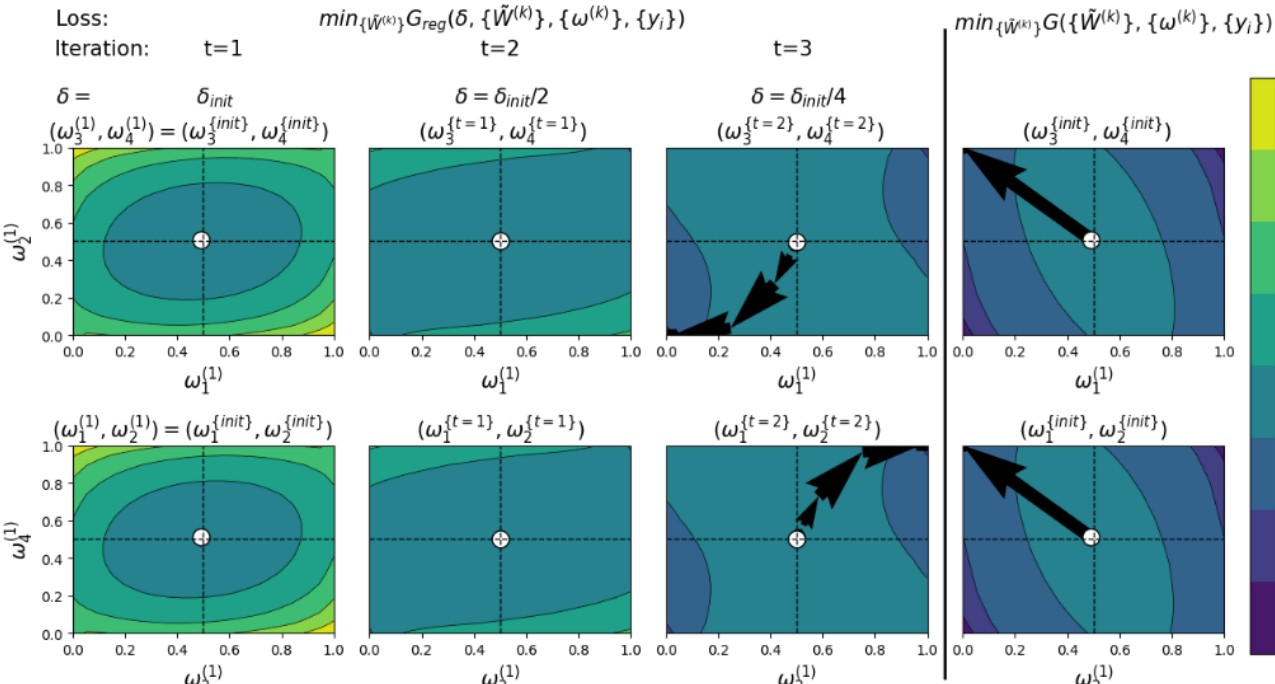

*Figure 4.* An experiment comparing the landscape and iterations of Algorithm 1 (first three columns) with a naive alternating minimization technique based on Proposition 3.4 (rightmost column). On the left, each column shows the loss landscape of our regularized loss in each iteration $t \in \{1, \ldots, 3\}$, where the partitioning updates in steps 5–8 are drawn on top of that. Specifically, in the first row, the loss landscape consists of varying values of the first two partition coordinates, while the last two are constrained to be the initial solution at this step. The bottom row is similar, but considers varying values for the third and fourth coordinates, while constraining the first two. The black arrows depict the partitioning parameter updates applied throughout each algorithm iteration. In the rightmost column, we provide a similar visualization for the alternating minimization based on the unregularized objective function. The data is composed of $N = 400$ samples in $D = 4$ dimensions that consist of two concatenated circles, meaning that the correct solution is $\boldsymbol{\omega}^{(1)} = (0, 0, 1, 1)$. In the regularized case (Left), after the algorithm converges for a specific configuration at $t$, its updated parameter values are assigned to $\boldsymbol{\omega}^{\{t+1\}}$, as described in Algorithm 1.

**Problem C.1.** *Let $\delta \in \mathbb{R}_+$. Consider the following minimization problem*

$$\min_{\left\{\tilde{\boldsymbol{W}}^{(k)}\right\}_{k=1}^{K}, \left\{\tilde{\boldsymbol{\omega}}^{(k)}\right\}_{k=1}^{K}} G_{reg}(\delta, \{\tilde{\boldsymbol{W}}^{(k)}\}_{k=1}^{K}, \{\tilde{\boldsymbol{\omega}}^{(k)}\}_{k=1}^{K}, \{\boldsymbol{y}_i\}_{i=1}^{N}), \tag{20}$$

*with the constraints:* $\sum_{k=1}^{K} \tilde{\omega}_d^{(k)} = 1$, $\sum_{j=1}^{N} \tilde{W}_{i,j}^{(k)} = 1$, $\tilde{W}_{i,i}^{(k)} = 0$ *and* $\sum_{j=1}^{N} \tilde{W}_{i,j}^{(k)} \log \tilde{W}_{i,j}^{(k)} \leq -\log(\alpha)$ *for* $i = 1, \ldots, N$, $k = 1, \ldots, K$, *and* $d = 1, \ldots, D$.

Our proposed algorithm (Algorithm 1) aims to solve the original problem (Problem 3.3) by solving a sequence of the regularized versions of the problem (Problem C.1) with diminishing regularization until the two problems align. Specifically, it begins by solving Problem C.1 for some $\delta = \delta_{init}$, and then uses its solution as the initialization point for solving the regularized problem with reduced $\delta$. This procedure repeats itself with the lowered value of $\delta$. In the last iteration, the regularization parameter $\delta$ is set to zero to match the unregularized problem.

The entropy regularization modifies the loss landscape to penalize hard partitioning solutions and is intended to prevent rapid convergence to poor local minima characterized by hard partitions, as discussed in the context of the unregularized objective at the beginning of the section. Specifically, for any $d \in \{1, \ldots, D\}$, a hard assignment $\{\tilde{\boldsymbol{\omega}}^{(k)}\}_{k=1}^{K} \subset \{0, 1\}^D$ yields zero

entropy, i.e., $\sum_{k=1}^{K} \tilde{\omega}_d^{(k)} \log \tilde{\omega}_d^{(k)} = 0$. In contrast, the regularization favors soft assignments $\{\tilde{\omega}^{(k)}\}_{k=1}^{K} \subset (0,1)^D$, with the uniform assignment $\tilde{\omega}_d^{(k)} = 1/K$ for all $k \in \{1,\ldots,K\}$ and $d \in \{1,\ldots,D\}$ attaining the minimum negative entropy of $-\log(K)$ (See Theorem 2.6.4 in (Cover & Thomas, 1991)).

We begin by describing the problem's solution for each $\delta$. Given that there are two sets of parameters—the feature partitions $\{\boldsymbol{\omega}^{(k)}\}$ and their corresponding affinity matrices $\{\boldsymbol{W}^{(k)}\}$—we propose an alternating minimization approach to solve the regularized problem for each $\delta$. Specifically, the method alternates between minimizing the objective function over the affinity matrices while keeping the partition weights fixed and vice versa. The solution to each step in the alternating minimization procedure is described by the following proposition, whose proof can be found in Appendix I.3.

**Proposition C.2.** *Let $\delta \geq 0$, $\{\boldsymbol{\omega}^{(k)}\}_{k=1}^{K} \subset [0,1]^D$ be a partitioning weights and $\{\boldsymbol{W}^{(k)}\}_{k=1}^{K}, \subset [0,1]^{N \times N}$ be affinity matrices that satisfy the constraints of Problem C.1.*

*Define $\{\boldsymbol{W}^{(k)*}\} \subset [0,1]^{N \times N}$ as in (9) based on $\{\boldsymbol{\omega}^{(k)}\}_{k=1}^{K}$, and $\{\boldsymbol{\omega}^{(k)*}\}_{k=1}^{K}$ by*

$$\omega_d^{(k)*} = \exp\left(-\frac{\sum_{i,j} W_{i,j}^{(k)} \left((\boldsymbol{y}_i)_d - (\boldsymbol{y}_j)_d\right)^2}{\delta}\right) \Big/ \sum_{s=1}^{K} \exp\left(-\frac{\sum_{i,j} W_{i,j}^{(s)} \left((\boldsymbol{y}_i)_d - (\boldsymbol{y}_j)_d\right)^2}{\delta}\right). \tag{21}$$

*Then, we have that*

$$\{\boldsymbol{W}^{(k)*}\} = \underset{\{\tilde{\boldsymbol{W}}^{(k)}\}}{\arg\min} \, G_{reg}(\delta, \{\tilde{\boldsymbol{W}}^{(k)}\}_{k=1}^{K}, \{\boldsymbol{\omega}^{(k)}\}_{k=1}^{K}, \{\boldsymbol{y}_i\}_{i=1}^{N}), \tag{22}$$

$$\{\boldsymbol{\omega}^{(k)*}\} = \underset{\{\tilde{\boldsymbol{\omega}}^{(k)}\}}{\arg\min} \, G_{reg}(\delta, \{\boldsymbol{W}^{(k)}\}_{k=1}^{K}, \{\tilde{\boldsymbol{\omega}}^{(k)}\}_{k=1}^{K}, \{\boldsymbol{y}_i\}_{i=1}^{N}). \tag{23}$$

We now turn to the choice of the initial regularization parameter, $\delta_{\text{init}}$, which determines the starting point of the sequential procedure. In the following proposition, we propose to select $\delta_{\text{init}}$ such that the regularized objective evaluated at the uniform partitioning — i.e. $\omega_d^{(k)} = 1/K$ for all $d \in \{1,\ldots,D\}$ and $k \in \{1,\ldots,K\}$— is guaranteed to be less than or equal to its value at any hard partitioning.

**Proposition C.3.** *Let $\{\overline{\boldsymbol{\omega}}^{(k)}\}_{k=1}^{K} \subset [0,1]^D$ be a soft uniform partitioning, i.e. $\overline{\omega}_d^{(k)} = 1/K$, and let $\{\overline{\boldsymbol{W}}^{(k)}\}_{k=1}^{K}$ be the corresponding affinity matrices from Proposition C.2. Let $\{\boldsymbol{\omega}^{(k)}\}_{k=1}^{K} \subset \{0,1\}^D$ and $\{\boldsymbol{W}^{(k)}\}_{k=1}^{K}$ be the optimal partitioning solution as discussed in Proposition 3.4.*

*Define*

$$\delta_{init} \equiv \frac{G(\{\overline{\boldsymbol{W}}^{(k)}\}_{k=1}^{K}, \{\overline{\boldsymbol{\omega}}^{(k)}\}_{k=1}^{K}, \{\boldsymbol{y}_i\}_{i=1}^{N})}{D \cdot \log(K)}. \tag{24}$$

*Then, for any hard partitioning solution $\{\tilde{\omega}^{(k)}\}_{k=1}^{K} \subset \{0,1\}^D$ and any corresponding affinity matrices $\{\tilde{\boldsymbol{W}}^{(k)}\}_{k=1}^{K}$, we have:*

$$G_{reg}(\delta_{init}, \{\overline{\boldsymbol{W}}^{(k)}\}_{k=1}^{K}, \{\overline{\boldsymbol{\omega}}^{(k)}\}_{k=1}^{K}, \{\boldsymbol{y}_i\}_{i=1}^{N}) \leq G_{reg}(\delta_{init}, \{\tilde{\boldsymbol{W}}^{(k)}\}_{k=1}^{K}, \{\tilde{\omega}^{(k)}\}_{k=1}^{K}, \{\boldsymbol{y}_i\}_{i=1}^{N}) \tag{25}$$

The proof of Proposition C.3 can be found in Appendix I.3.

In Algorithm 1, we outline the steps of our proposed feature partitioning procedure based on Propositions C.2 and C.3. In Figure 4, we illustrate the convergence behavior of our proposed optimization procedure versus a naive alternating minimization of the unregularized objective function (in Problem 3.3). As observed from the rightmost column, the latter quickly converges to an incorrect solution. However, as seen from the leftmost column, the regularization guides the solution towards a uniform partition at $t = 1$. As the regularization parameter $\delta^{\{t\}}$ decreases (second and third columns from the left), the solution gradually shifts towards the correct partition.

**Determining the Bandwidth Parameters.** The expression of the affinity matrices in Proposition C.2 is based on $\{\epsilon_{k,i}\}_{k=1,i=1}^{K,N} \subset [0,\infty)$, which are chosen to be the minimal values that satisfy the entropy constraints ($\sum_{j=1} W_{i,j}^{(k)*} \log W_{i,j}^{(k)*} \leq -\log(\alpha)$ for all $k = 1,\ldots,K$ and $i = 1,\ldots,N$). Our approach for setting the bandwidth

parameters is very similar to the one used in tSNE, albeit adapted for weighted Euclidean distances (appearing in the form of the affinity matrices in Proposition C.2) instead of standard Euclidean distance. Specifically, we use a binary–search–like search to set the bandwidth parameters to satisfy the entropy constraints, as described below.

As the bandwidth parameters are computed independently of each other, we focus on a specific $\epsilon_{k,i}$ for some $i \in \{1, \ldots, N\}$ and $k \in \{1, \ldots, K\}$. The behavior of the entropic function $\sum_{j=1}^{N} W_{i,j}^{(k)*} \log W_{i,j}^{(k)*}$ with respect to $\epsilon_{k,i}$ can be characterized by Lemma I.1 in Appendix I.1 (which is used for the proof of Proposition 3.1 and Lemma I.3), where we use the notation $\Delta_j = \|\boldsymbol{y}_i - \boldsymbol{y}_j\|_{\boldsymbol{\omega}^{(k)}}^2$ for $j = 1, \ldots, N$. Specifically, Lemma I.1 indicates that this entropic function is non-increasing in $\epsilon_{k,i}$ and bounded in the interval $[-\log(N-1), -\log(\tilde{\alpha}_{k,i})]$, where

$$\tilde{\alpha}_{k,i} = |\{j \in \{1, \ldots, N\}/\{i\} \, : \, \|\boldsymbol{y}_i - \boldsymbol{y}_j\|_{\boldsymbol{\omega}^{(k)}} = \min_{t \neq i} \|\boldsymbol{y}_i - \boldsymbol{y}_j\|_{\boldsymbol{\omega}^{(k)}}\}|. \tag{26}$$

Moreover, it establishes that as $\epsilon_{k,i} \to 0$ the entropy converges to $-\log(\tilde{\alpha}_{k,i})$, whereas as $\epsilon_{k,i} \to \infty$ it tends to $-\log(N-1)$. Since the entropy is monotonic and bounded with respect to this parameter, we employ a binary-search-like iterative procedure to efficiently approximate the appropriate value of $\epsilon_{k,i}$ to within a small error, similarly to the one used in tSNE.

**Computational Complexity.** We now analyze the computational complexity of the parameter updates shown in Algorithm 1, which are based on the expressions defined in Proposition C.2. Consider a dataset with $N$ observations in $\mathbb{R}^D$, where the features are partitioned into $K$ subsets. The computational complexity of updating the $K$ partitions and their corresponding affinity matrices using our approach is $O(KN^2D)$, based on the next proposition.

**Proposition C.4.** *Let the data consist of $N$ data points in $\mathbb{R}^D$. Then, the computational complexity of obtaining the partitioning weights $\{\boldsymbol{\omega}^{(k)*}\}_{k=1}^K$ and the affinity matrices $\{\boldsymbol{W}^{(k)*}\}_{k=1}^K$, as defined in Proposition C.2, is $O(KN^2D)$.*

Its proof can be found in Appendix I.3. For reference, the computational cost of the affinity matrix construction in tSNE is $O(N^2D)$. Hence, the computational complexity of our approach incurs an additional factor of $K$, which is typically small. Nonetheless, it may be significantly slower than tSNE due to the iterative procedure we employ for solving our optimization problem.

To enhance scalability, we also describe an implementation that exploits a low-rank approximation of the data, which can considerably reduce the running time for large datasets. Specifically, we approximate the data by truncating its singular value decomposition (SVD) to $S \ll \min(N, D)$ leading components, and then use this compact representation to speed up computations. Subsequently, the computational complexity of updating the $K$ partitions and their corresponding similarity graphs will reduce to $O(K(S^2N^2 + S^2D))$; see the following proposition.

**Proposition C.5.** *Let the data consist of $N$ data points in $\mathbb{R}^D$. Suppose the data is given in the form of a singular value decomposition (SVD) approximation of rank $S \ll N, D$. Then, the computational complexity of obtaining the partitioning weights $\{\boldsymbol{\omega}^{(k)*}\}_{k=1}^K$ and the affinity matrices $\{\boldsymbol{W}^{(k)*}\}_{k=1}^K$, as defined in Proposition C.2, is $O(K(S^2N^2 + S^2D))$.*

Its proof can be found in Appendix I.3.

## D. Additional Experiment Involving Two Concatenated Views of Rotating Figurines

We demonstrate the effectiveness of our approach in disentangling three simultaneously occurring processes, each associated with a distinct rotating object in a shared scene recorded by two separate cameras. Our method enables clearer visualization of each process individually and reveals the correspondence between the two imaging sources, effectively integrating their information.

Specifically, we consider a dynamic scene with three rotating figurines, each spinning at a different angular speed. The rotation angles define the latent parameters governing the scene (Lederman & Talmon, 2018). Two cameras capture this scene simultaneously: both record the shared bulldog figurine, while each also captures a unique, camera-specific figurine. At each time point, the two corresponding image frames—one from each camera—are concatenated horizontally to form a single, wider image. This sequence of concatenated images constitutes the dataset used in our analysis, comprising $N = 5000$ grayscale images with $D = 9600$ pixels each, as illustrated in Figure 5(a). As a result, each individual camera view depends on two latent parameters, while the stacked dataset as a whole is governed by all three.

In (Lederman & Talmon, 2018) and similarly in (Lindenbaum et al., 2015), the views were treated separately and then aligned to extract the shared component. In contrast, our method operates directly on the concatenated dataset, allowing

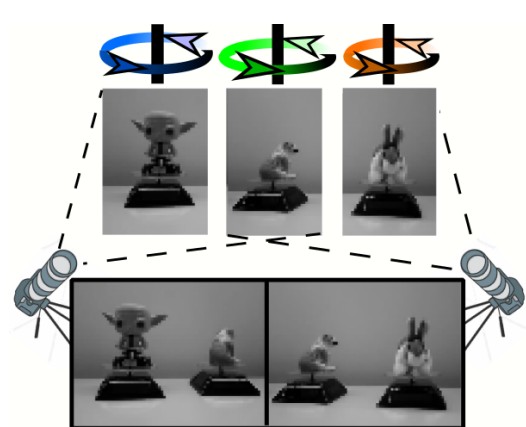

(a) Illustration of the data generating process.

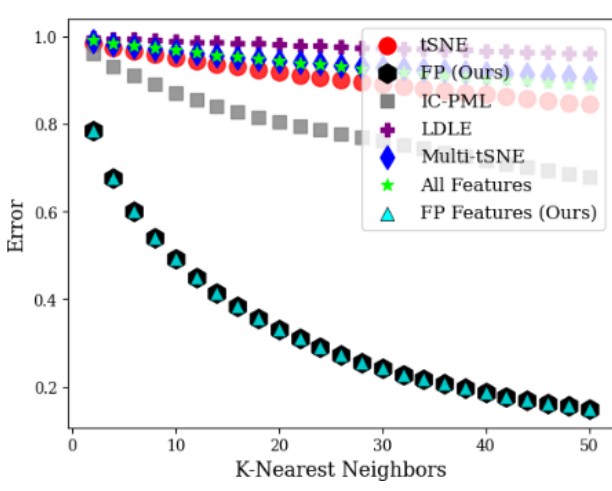

(b) Quantative comparison.

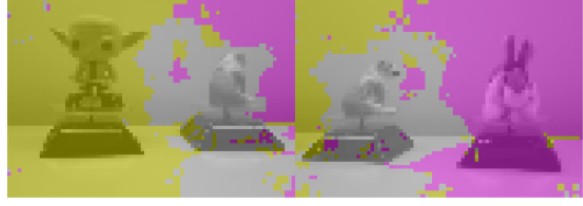

(c) The three extracted partitions using our approach.

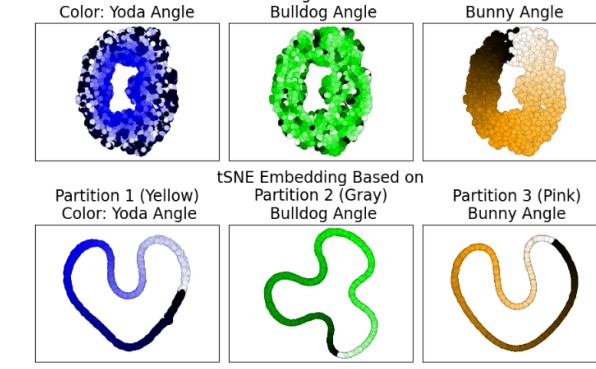

(d) tSNE embeddings.

*Figure 5.* Partitioning and embedding data from two concatenated views of rotating figurines. The dataset consists of $N = 5000$ grayscale images, each with $D = 9600$ pixels, formed by horizontally concatenating synchronized video frames from two different viewing angles (illustrated in Figure 5(a)). (a) Illustration of the data acquisition setup. At each time point, two cameras simultaneously capture the scene, each observing two out of three rotating figurines (Yoda, Bulldog, and Bunny), which are rotating at distinct angular speeds. (b) Quantitative comparison of embeddings produced by various algorithms, evaluated using K-nearest neighbor error against the known latent angles of the figurines. Our method yields the lowest error, indicating that the extracted partitions best reflect the underlying structure. Further evaluation details are provided in Appendix D. (c) The three feature partitions identified by our method (indicated by three different colors). (d) Top row: tSNE embeddings using all pixels, with data points colored by the rotation angle of the Yoda (left), Bulldog (center), and Bunny (right) figurines. Bottom row: tSNE embeddings based on each extracted partition, colored respectively by the angle of the figurine best captured by that partition (left: Yoda, center: Bulldog, right: Bunny), as shown in Figure 5(c).

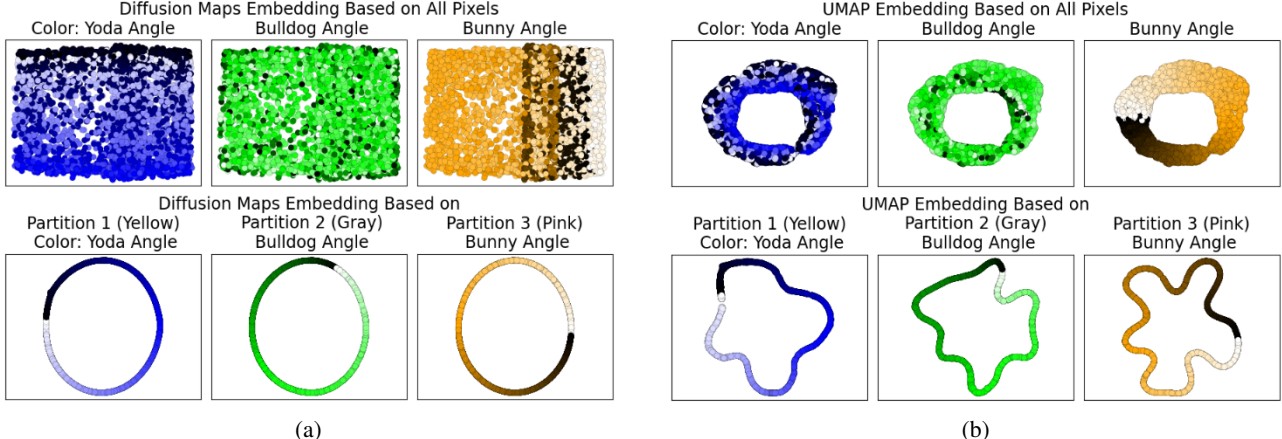

*Figure 6.* Embeddings of two concatenated views of rotating figurines. (a) Top row: Diffusion Maps embedding based on all pixels colored by the angle of the Yoda (top left), Bulldog (top middle), and Bunny (top right) figurines. Bottom row: tSNE embeddings based on the first (bottom left), second (bottom middle), and third extracted partitions (bottom right) as shown in Figure 5(c), with data points colored by the angles of the Yoda, Bulldog, and Bunny figurine, respectively. (b) Embeddings using UMAP analogous to (a).

us to recover the substructure associated with each figurine—including those whose visual footprint spans both camera views. This leads to better visual separation of the underlying processes and reveals the correspondence between the two measurement sources.

We apply our approach to the stacked images and extract $K = 3$ pixel partitions. In Figure 5(d), we compare the tSNE embedding based on all the pixels with the embedding based on each pixel partition. While the embedding based on all pixels reflects only the Bunny figurine's angle, the partition-based embeddings successfully capture the rotational structure originating from all the figurines, with each embedding corresponding to a specific figurine. In Figures 6(a) and 6(b), we further compare the embeddings obtained using UMAP and Diffusion Maps to validate the observed structures.

In Figure 5(b), we quantitatively compare the tSNE embeddings generated from our pixel partitions with those produced by alternative techniques, similarly to the comparison in Section 5.2. The error measure used here differs slightly, as the ground truth latent variables are continuous rotation angles of three figurines, rather than discrete clusters. Hence, to compute an error measure for a given embedding of the dataset, we do the following. For each data point $i$ and for each one of the three rotation angles, we find the $k$ nearest neighbors of point $i$ and the $k$ nearest angles of angle $i$, and compute the relative set difference between the two groups. Specifically, the score is defined as the number of neighbors in the angle-based set that are not present in the embedding-based set, divided by $k$. This provides three scores of angle inconsistencies for each point (corresponding to the angles). Following the same procedure used in Section 5.2, we then average these scores across all data points, producing three error measures that quantify the inconsistency of the embedding with respect to the three figurines' angles. For methods that provide a single embedding of the data, we average these three scores, quantifying how much this embedding is consistent with all the latent processes. For methods that produce three embeddings, we expect each embedding to be consistent with only one of the latent processes. Therefore, in such cases, we assign to each embedding only one of its three scores (without repetition) such that the average of the assigned scores is minimized for the three embeddings.

All methods were applied in a consistent manner with the setup described in Section 5.2, with necessary adjustments to support the comparison across multiple embeddings when applicable. In particular, for algorithms capable of producing multiple embeddings, we generated three—one for each latent process. The results demonstrate that our method significantly outperforms the alternatives in aligning with the underlying rotational structure of the data, highlighting its effectiveness in disentangling independent sources of variation.

We further evaluate our approach by comparing it to traditional clustering methods when applied to the feature space instead of the sample space. Specifically, we extract $K = 3$ partitions using k-means and spectral clustering, treating the values of each feature across all images as a sample point in $\mathbb{R}^N$. We also apply these algorithms with $K = 4$ to provide a more flexible setting that may better capture the structure present in the data. The resulting partitions, shown in Figure 7, highlight that both k-means and spectral clustering fail to isolate each figurine into a distinct partition, whereas our approach

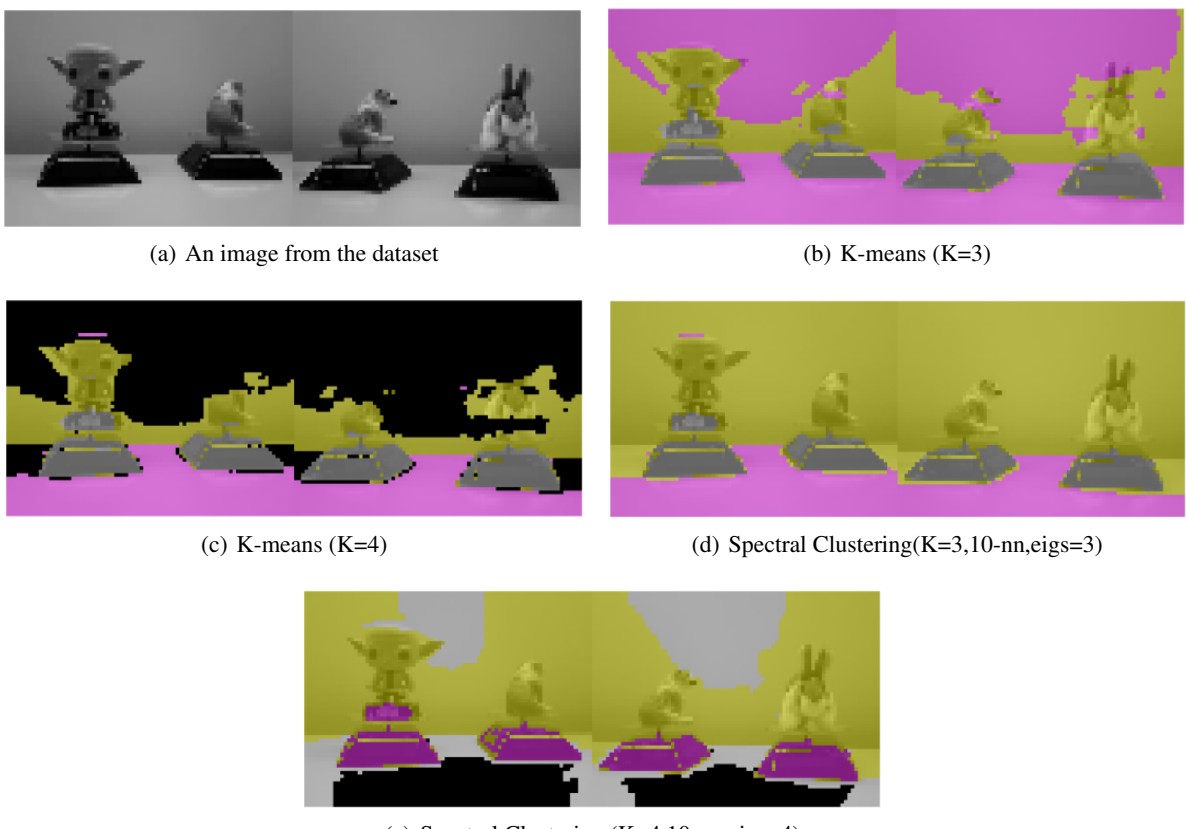

(a) An image from the dataset

(b) K-means (K=3)

(c) K-means (K=4)

(d) Spectral Clustering(K=3,10-nn,eigs=3)

(e) Spectral Clustering (K=4,10-nn,eigs=4)

*Figure 7.* Pixel partitions generated using k-means and spectral clustering by clustering the pixel data while treating the samples as coordinates (i.e., applied to the transposed data matrix). Each color indicates a different partition. Here, $K$ denotes the number of clusters used, 10-nn refers to the use of a 10-nearest neighbors graph, and "eigs" indicates the number of eigenvectors used in spectral clustering.

successfully does so, as illustrated in Figure 5(c).

To quantitatively assess how well each approach partitions the pixels and captures the underlying structure of the data, we evaluate the correspondence between each extracted pixel partition and the true rotation angles of the figurines—the latent variables governing the data. To this end, we define an accuracy measure as follows. For a given partition and figurine, we compute the relative overlap between two sets of 50-nearest neighbors for each image: one based on the true figurine angle and the other based on the pixel values restricted to the partition. The relative intersection score, defined as the size of their intersection divided by 50, reflects how strongly the latent parameter (rotation angle) is expressed in the partition's feature space. Since the figurines rotate independently, an ideal partitioning should align with exactly one figurine's angle. Thus, for each approach, we match each figurine to the partition with the highest relative intersection score and report the average correspondence across all three figurines. Furthermore, we compute the standard deviation of the relative intersection score above, and report its average across all three figurines. We repeat this evaluation using 30-nearest neighbors to provide additional insights into the consistency of the results.

The resulting mean overlap score are shown in Table 2, reflecting the accuracy of each approach in partitioning the pixels into the true subsets associated with each figurine. As evident from the results, our approach substantially outperforms the clustering methods and more effectively isolates the latent factors of variation.

Thus, to conclude, our approach outperforms both clustering and embedding-based methods in capturing the underlying structure of the data. By effectively isolating the latent factors of variation, it provides a more faithful decomposition of the observed measurements, demonstrating clear advantages over traditional techniques.

**Implementation details.** To generate the embedding of the tSNE embedding based on all the features in Figure 5(d) we used perplexity of 40 with 100 simulations. The visualizations using the tSNE algorithm of the data based on each partition

| Method | Agreement of 30-nearest neighbors (std) | Agreement of 50-nearest neighbors (std) |
|---|---|---|
| All features | 8.21 (4.91) | 11.74 (4.75) |
| K- means (K=3) | 18.49 (9.74) | 25.93 (10.61') |
| K-means (K=4) | 18.23 (9.11) | 25.42 (10.31) |
| Spectral Clustering (nn=10,K=3, eigs=3) | 18.28 (8.4) | 24.72 (8.82) |
| Spectral Clustering (nn=10,K=4,eigs= 4) | 18.85 (8.71) | 25.49 (9.19) |
| Spectral Clustering (nn=30,K=3, eigs=3) | 18.12 (8.5) | 24.47 (8.97) |
| Spectral Clustering (nn=30,K=4,eigs=4) | 18.19 (8.5) | 24.53 (8.98) |
| Spectral Clustering (nn=10,K=3,eigs=5) | 18.04 (8.5) | 24.41 (8.98) |
| Spectral Clustering (nn=10,K=4,eigs=5) | 18.63 (8.76) | 25.25 (9.26) |
| Spectral Clustering (nn=10,K=3,eigs=10) | 17.51 (7.94) | 23.8 (8.22) |
| Spectral Clustering (nn=30,K=4,eigs=10) | 18.94 (10.35) | 25.49 (11.51) |
| FP **(Ours)** | **75.93** (12.22) | **85.15** (8.59) |

*Table 2.* Quantitative comparison of different pixel partitions produced by various clustering algorithms, evaluated using $k$-nearest neighbor against the known latent angles of the figurines. Our approach yields the lowest error, indicating that the extracted partitions best reflect the underlying structure. The "All Features" baseline refers to using all pixels to compute the $k$-nearest neighbors without applying any partitioning. Here, $K$ indicates the amount of clusters used within each method, "nn" indicates the amount of nearest neighbors, and "eig" indicates the amount eigenvectors used. For the Feature Partioning method we partitioned the pixels into 3 partitions. Further evaluation details are provided in Appendix D.

uses a perplexity of 110. The high perplexity can be attributed to a known issue with tSNE, where it sometimes distorts circular embeddings, resulting in discontinuities. A common solution to this problem is to increase the perplexity of the embedding.

The Diffusion Maps embeddings, shown in Figure 6(a), were generated using a bandwidth parameter that is the maximal squared Euclidean distance among each data point and its corresponding 10-nearest neighbor. Furthermore, we used a normalization factor of $\alpha = 1$. Finally, the UMAP embeddings based on our extracted partitions in Figure 6(b) were generated using a local neighborhood parameter ($n\_neighbors$) of 80 due to the 1-dimensional structure of the embedding. This is a due to the same issue as discussed above for tSNE. The UMAP embedding based on all the features was constructed with its default parameter 15.

**Details of Quantitative Comparison with Embedding-Based Methods.** The parameters used for each algorithm are identical to the ones used in Section 5.2, with the exception that we generated 3 embeddings for any algorithm that allowed it, for LDLE we used the $\eta_{min}$ parameters of 5 and 10, and for the tSNE of our extracted partitions we used a perplexity of 10. We note that our approach was applied with entropy constraints that correspond to a perplexity of 10.

**Details of Quantitative Comparison with Clustering Methods.** The parameters used for k-means and spectral clustering followed the default settings in `scikit-learn`, unless stated otherwise.

# E. Numerical Comparison of Empirical and Analytical Loss Landscapes

In Figure 8(a), we compare the empirical loss landscape of (16) with its analytical counterpart derived in (17), for the case $K = 2$. The dataset consists of $N = 1000$ samples drawn uniformly from three latent manifolds, $\mathcal{M}_1, \mathcal{M}_2, \mathcal{M}_3 \subset \mathbb{R}^2$, where each manifold forms a unit circle and $\mathcal{M}_3$ serves as the shared component between partitions. The observed feature dimensions are $D_1 = 2500$ and $D_2 = 7500$. As predicted by Theorem 4.3, the minima of the loss function occur near the ground truth partitioning solutions (i.e., $p_1 = 1, p_2 = 0$ or vice versa). In Figure 8(b), we show the empirical landscape of the original loss function defined in Problem 3.3 under the same data configuration. The observed landscape closely resembles that of the simplified objective studied in Section 4, further supporting the validity of the analytical approximation.

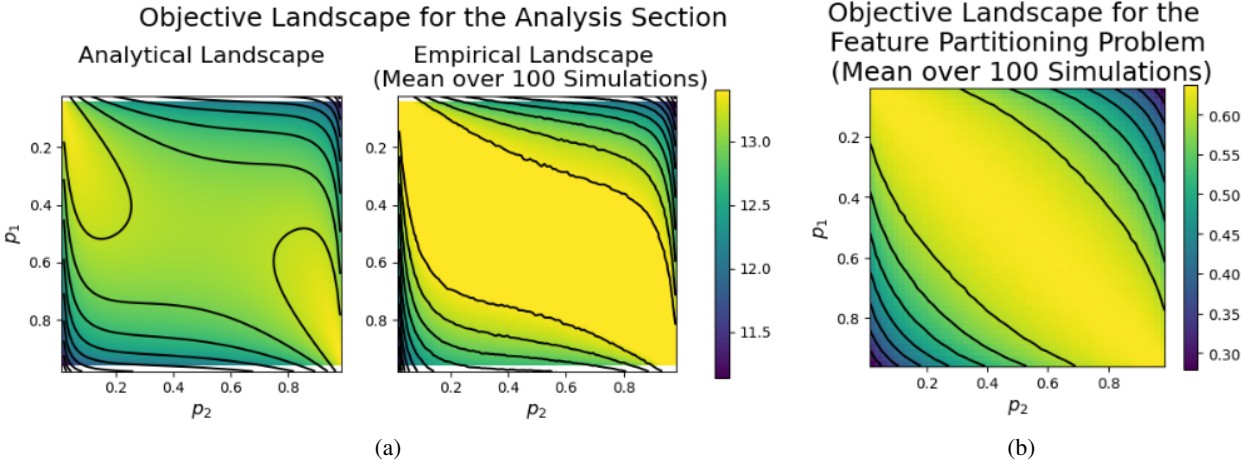

*Figure 8.* An experiment based on Section 4 for $K = 2$ with with $\epsilon = 0.2$. (a) The analytical loss landscape of Problem 4.1 presented in (17) (Left) and the mean empirical loss as defined in (16) (Right). For the empirical case, the $p_1, p_2 \in (0, 1)$ indicate the proportion of features out of the two feature subsets used by $\boldsymbol{\omega}^{(1)}$, while $\boldsymbol{\omega}^{(2)}$ taking the remainder. The color represents the mean value over 100 simulations, based on the affinity matrices defined in Corollary 4.2. (b) We consider a similar mean empirical loss based on Problem 3.3 ($\min_{\{\boldsymbol{W}^{(k)}\}_{k=1}^{K}} G(\{\boldsymbol{W}^{(k)}\}_{k=1}^{K}, \{\boldsymbol{\omega}^{(k)}\}_{k=1}^{K}, \{\boldsymbol{y}_i\}_{i=1}^{N})$), where the affinity matrices are as defined in (9). The perplexity parameter was set to $\alpha = 20$. Additional details can be found in Appendix E.

## F. Experimental Details and Additional Results

### F.1. Details for Section 5.1

In this experiment, we define two sets, $\mathcal{Y}, \tilde{\mathcal{Y}} \subset \mathbb{R}^6$. The two datasets are defined based on a 2-dimensional ellipse defined by

$$\mathcal{A} = \{(8\cos(\theta)\sin(\phi), 6\sin(\theta)\sin(\phi), 4\cos(\theta))^T \quad | \quad \theta \in [0, 2\pi], \, \theta \in [0, \pi]\}, \tag{27}$$

and $\boldsymbol{R}, \boldsymbol{S} \in \mathbb{R}^{3 \times 3}$ be two orthogonal matrices. We now define the two sets by

$$\tilde{\mathcal{Y}} = \left\{ \begin{pmatrix} \boldsymbol{R}\boldsymbol{a} \\ \boldsymbol{S}\boldsymbol{b} \end{pmatrix} \quad | \quad \boldsymbol{a}, \boldsymbol{b} \in \mathcal{A} \right\} \tag{28}$$

$$\mathcal{Y} = \left\{ \begin{pmatrix} \boldsymbol{R}\boldsymbol{a} \\ \boldsymbol{S}\boldsymbol{b} \end{pmatrix} \quad | \quad \boldsymbol{a}, \boldsymbol{b} \in \mathcal{A}, \quad \theta(\boldsymbol{a}) = \theta(\boldsymbol{b}) \right\}, \tag{29}$$

where $\theta(\boldsymbol{a})$ denotes the polar angle $\theta$ corresponding to the data point $\boldsymbol{a}$ in the parametrization of $\mathcal{A}$ in (27).

The set $\tilde{\mathcal{Y}}$ corresponds to the independent case discussed in Section 5.1, where as the first three and last three coordinates are independent of each other. Additionally, $\mathcal{Y}$ represents the partially-dependent case, in which the first and last three coordinates are partially coupled through a shared polar angle. In both cases, the samples used in the experiment— denoted by $\tilde{\boldsymbol{y}}_1, \dots, \tilde{\boldsymbol{y}}_N \in \tilde{\mathcal{Y}}$ and $\boldsymbol{y}_1, \dots, \boldsymbol{y}_N \in \mathcal{Y}$— were drawn independently and uniformly from their respective sets.

This experiment evaluates the ability of our approach and several baseline algorithms to partition the coordinates into the true feature subsets defining the data—namely, the first three coordinates and the last three. The comparison includes Spectral Co-Clustering (Dhillon, 2001), Spectral Bi-Clustering (Kluger et al., 2003), k-means (Lloyd, 1982) and spectral clustering (Ng et al., 2001). We evaluated Spectral Co-Clustering and Bi-Clustering using their feature clustering. Additionally, we applied k-means and spectral clustering on the features, treating samples as coordinates (i.e., applied on the transposed data), and clustering them into $K = 2$ groups. Additionally, for spectral clustering, we used two embedding dimensions. The output of each of the clustering algorithms will be considered as a pair of binary indicator vectors $\tilde{\boldsymbol{\omega}}^{(1)}, \tilde{\boldsymbol{\omega}}^{(2)} \in \{0, 1\}^6$. Specifically, the vector $\tilde{\boldsymbol{\omega}}^{(1)} \in \{0, 1\}^6$ takes the value 1 for all coordinates assigned to the first cluster and 0 otherwise. The complementary vector $\tilde{\boldsymbol{\omega}}^{(2)}$ is defined analogously, indicating membership in the second cluster.

Finally, we define the error measure used to quantify the discrepancy between the estimated partitions $\tilde{\boldsymbol{\omega}}^{(1)}, \tilde{\boldsymbol{\omega}}^{(2)} \in \{0, 1\}^6$ and the true partitions $\boldsymbol{\omega}^{(1)}, \boldsymbol{\omega}^{(2)} \in \{0, 1\}^6$. As discussed above, the true partitions are given by $\boldsymbol{\omega}^{(1)} = (1, 1, 1, 0, 0, 0)$,

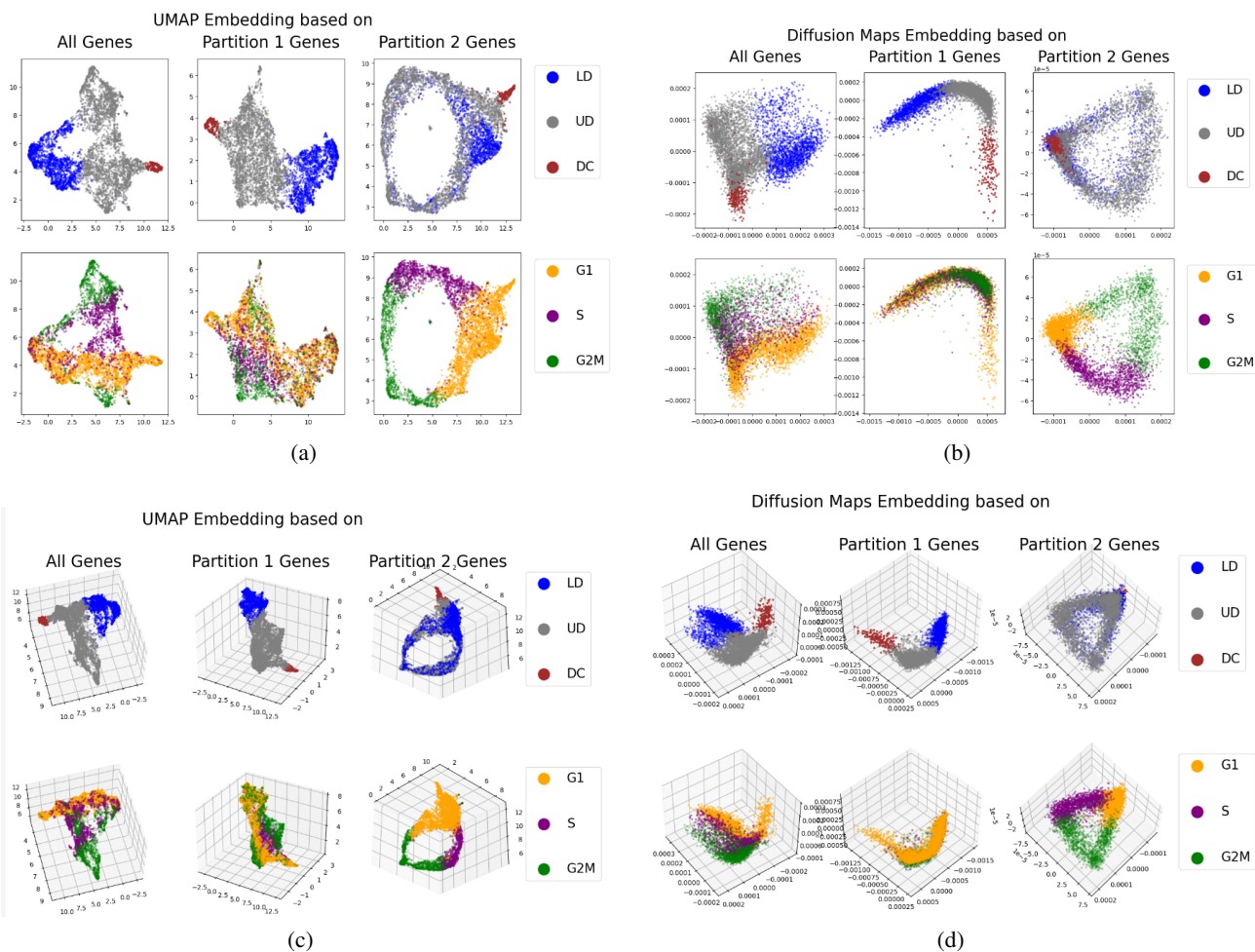

*Figure 9.* Additional embeddings of the scRNA-seq data from Section 5.2, highlighting the advantages of our approach to discover distinct salient cellular processes. Plots (a) and (c) show the UMAP embeddings of the data in two and three dimensions, respectively, using either all genes or only the genes from partitions 1 and 2. Similarly, plots (b) and (d) present the corresponding Diffusion Maps embeddings. Within each plot, the top row includes the cells' embeddings colored by their cell type, with partition 1 revealing the LD/UD to DC developmental trajectory. At the bottom, the cells are colored by the cell type, with partition 2 revealing the cycling progression.

with $\boldsymbol{\omega}^{(2)}$ being the complimentary vector. The error is defined as

$$d(\{\boldsymbol{\omega}^{(k)}\}_{k=1}^K, \{\tilde{\boldsymbol{\omega}}^{(k)}\}_{k=1}^K) = \min_{\tilde{k} \in \{1,2\}} \|\boldsymbol{\omega}^{(1)} - \tilde{\boldsymbol{\omega}}^{(\tilde{k})}\|_1, \tag{30}$$

where $\|\boldsymbol{a}\|_1 = \sum_{d=1}^6 |a_d|$ denotes the $L_1$ norm. Only $\boldsymbol{\omega}^{(1)}$ is considered, as $\boldsymbol{\omega}^{(2)}$ is its complementary partition. This error can be interpreted as a non-normalized Jaccard distance between the true feature partition and the estimated one, up to a permutation of the ordering of partitions.

### F.2. Details and Additional Results for Section 5.2

This section provides additional information and extended results related to Section 5.2. We begin by describing the preprocessing applied to the dataset, followed by additional visualizations using the extracted feature partitions. We then repeat the experiment using a larger gene set and a related dataset to further demonstrate the applicability of our method. Finally, we provide details about the quantitative evaluation done in the main text.

**Dataset Preprocessing.** In Section 5.2, we applied our approach to the dataset used in (Qu et al., 2024), specifically the wild-type cells within the E14.5 WLS experiment. The data included a count data matrix comprised of $5,572$ cells along

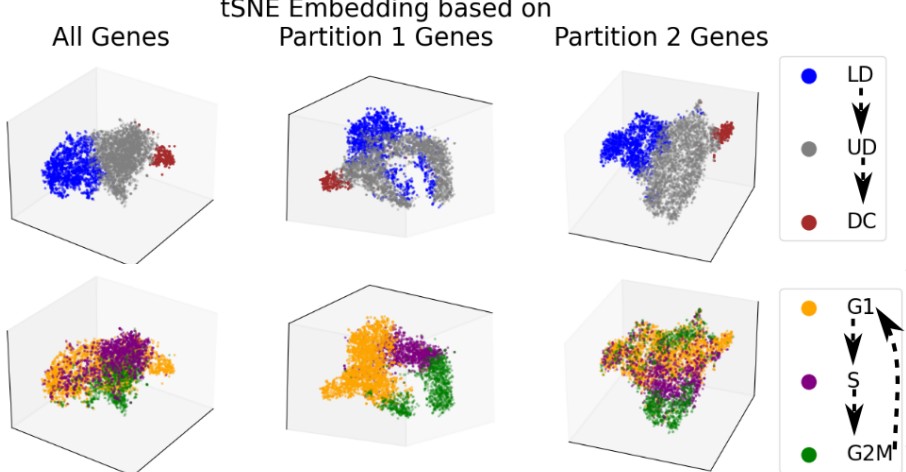

*Figure 10.* Embeddings obtained by repeating the experiment of Section 5.2 using 10000 genes (instead of 5000 in Section 5.2). The embeddings in the upper row are colored according the cell type, while the bottom row (containing the same embeddings) is colored according to the cell cycle state. The embedding based on all the genes and the embedding from partition 2 reflect the cell type development, while the embedding based on partition 1 uniquely reflects the cell cycle development. We can see that despite the additional variation due to the extra genes, the substructures remains clearly visible.

with their $32,285$ gene expressions. We employed the normalization scheme from the Seurat package, where the gene counts in each cell were divided by the total number of genes within that cell and then scaled by $10,000$. Each entry was subsequently log-normalized using the transformation $\log(1 + x)$. We then retained the $5,000$ most highly variable genes. Finally, we used the z-score transformation on each gene across the samples and reduced the rank of the data matrix to $50$ by truncating the singular value decomposition (SVD). As for our approach, we applied Algorithm 1 on the data using a perplexity parameter of $40$ and $100$ simulations.

For comparison, Qu et al. (2024) used only the top $2,000$ highly variable genes and retained the top 30 principal components, which corresponds to keeping the leading 30 components in the SVD. Thus, our configuration involves a larger set of genes and a higher-rank representation, retaining more variability in the data.

We evaluated the alignment between partition 2 and cell-cycle activity using Seurat's updated G2M and S phase gene sets (cc.genes.updated.2019 in Seurat 5.3.0) (Tirosh et al., 2016), which are commonly used in single-cell analysis. Since these lists use human gene names, we mapped them to mouse counterparts with gprofiler2 (v0.2.3), yielding 95 genes. Then, we removed the genes that were filtered during the data preprocessing stage, resulting in 86 remaining genes. All 86 genes were assigned to partition 2, indicating that the cell-cycle signal governs this partition.

**Additional Results.** The effectiveness of our feature partitioning is further demonstrated through additional embeddings of the dataset, computed using all genes as well as each extracted partition; see Figure 9. As in Figure 3, the embedding based on all genes reveals only the cell *type*, while each partition reveals only one of the biological processes: cell *type* (by partition 1) and cell *cycle* (by partition 2). It is evident that the partitioning of features has a vital effect on visualization techniques in general, and is not specific to tSNE.

We next repeat the original experiment in a more challenging setting using $10,000$ of the most variable genes to show the robustness of our approach. By increasing the number of genes, we allow noisy features into our dataset, making the partition task more complex. We applied our approach again with $K = 2$, and in Figure 10 we compare the tSNE embedding based on all genes, with the embeddings based on each of the extracted partitions. Evidently, the two partitions still extract the two underlying structures that correspond to the cell *type* (by partition 2) and cell *cycle* (by partition 1). Furthermore, we validated these partitions using known gene markers for each biological process, as described in the final paragraph of Section 5.2. Specifically, the gene markers associated with cell type were found in partition 2. For the cell-cycle genes, we used Seurat's updated G2M and S phase gene sets, originally comprising 95 genes. After removing the genes that were filtered during the data's preprocessing stage, only 93 genes remained, and they were all assigned to partition 1. This indicates that the genes were partitioned effectively according to the biological underlying processes.

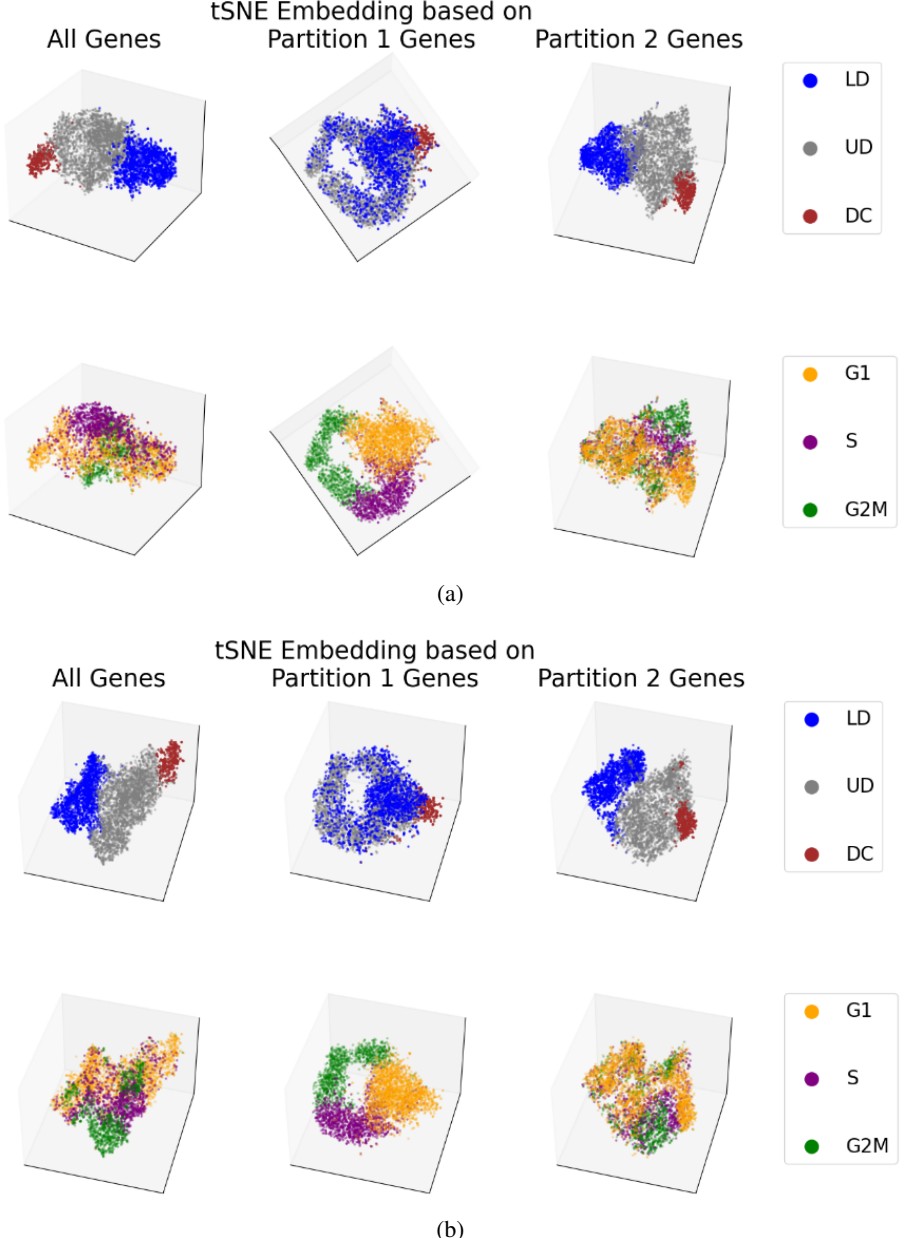

*Figure 11.* A similar experiment to the one done in Section 5.2 on a different dataset. Plot (a) shows the embedding of $N = 5598$ cells characterized by their $D = 5000$ gene values, while plot (b) presents the corresponding embedding obtained using $D = 10000$ genes. The upper row within each plot indicates each cell's type, while the bottom row shows each cell's cell cycle state. As shown, the embeddings based on all the genes and partition 2 genes are inductive to the cell type development (top), while the embedding based on partition 1 genes reflects the cell cycle development (bottom).

We repeat the experiment on a similar dataset to further validate our results on data with similar characteristics. Specifically, we consider the wild-type cells within the E14.5 SMOM experiment (Qu et al., 2024) that consists of a count data matrix with $5,598$ cells along with their $32,286$ genes expressions. We apply the same preprocessing pipeline as in the previous experiment, including the selection of highly variable genes. As before, we evaluate two configurations: one using the top $5,000$ most variable genes and another using the top $10,000$.

In Figure 11, we show the results for both configurations. We compare the tSNE embeddings based on all the genes with the embeddings based on each extracted partition. As shown in Figure 11, the embedding based on all the genes reveals only the structure related to the cell type development, while obscuring the cell-cycle phase. On the other hand, the embeddings from our two partitions reveal both the cell-cycle phase and the cell type development (in partition 1 and 2, respectively).

Furthermore, we validated these partitions using known gene markers for each biological process, as described in the final paragraph of Section 5.2. Specifically, the gene markers associated with cell type were found in partition 2. For the cell-cycle genes, we used Seurat's updated G2M and S phase gene sets, originally comprising 95 genes. After removing the genes that were filtered during the data's preprocessing stage, only 63 out of the 95 genes remained in the $5,000$-gene dataset and $81$ out of the 95 in the $10,000$-gene dataset. In both cases, all these genes were assigned to partition 1. This indicates that the genes were partitioned effectively according to the biological underlying processes.

**Quantitative Comparison.** The quantitative comparison shown in Figure 3(b) evaluates how well the embeddings generated based on our extracted partitions reveal the two underlying processes of the data compared to other methods. To define an error measure for a given embedding of the dataset, we proceed as follows. For each point (cell) i and each of the two types of labels (cell phase and type), we find the k nearest neighbors of point i and compute the proportion of nearest neighbors whose label differs from the label of point i. This provides two scores (of label inconsistencies) for each point, corresponding to the two processes. We then average these scores across all data points, producing two error measures: one quantifying the inconsistency of the embedding with respect to the cell phase and the other with the cell type. For methods that provide a single embedding of the data, we average these two scores, quantifying how much this embedding is consistent with both latent processes. For methods that produce two embeddings, we expect each embedding to be consistent with only one of the latent processes. Therefore, in such cases, we assign to each embedding only one of its two scores (without repetition) such that the average of the assigned scores is minimized for the two embeddings.

Next, we describe the parameters used by the different embedding algorithms in the quantitative comparison. It is important to note that the error measured for IC-PML, Multi-tSNE, and LDLE for each specific $K$-nearest neighbor corresponds to the smallest error observed across a wide range of hyperparameter values. The specific hyperparameter settings explored are detailed below:

- IC-PML (960 parameter configurations):

  - Number of manifolds: 2
  - Total eigenvectors: 50
  - Number of eigenvectors used from each manifold: $[1, 2, 3, 4, 5]$.
  - Epsilon parameters: the maximal distance between each point and its $k$-th nearest neighbor, where the considered $k$ included $5, 10, 20, 40$.
  - Similarity criterion coefficient: $[0.01, 0.03, 0.05, 0.1, 0.3, 0.5]$
  - Eigenvalue criterion coefficient: $[0.01, 0.03, 0.05, 0.1, 0.3, 0.5, 1, 2]$

- LDLE (32 parameter configurations):

  - Embedding dimensions: 2 and 3.
  - $\eta_{\min}$: 10 and 20.
  - k_tune: $5, 10, 20$ and $40$
  - torn: We considered the torn and non-torn data embedding options.

- Multi-tSNE (8 parameter configurations):

  - Number of maps: 2
  - Embedding dimension: 2 and 3.
  - Perplexity: $5, 10, 20$ and $40$.

- tSNE based on all features (8 parameter configurations):

  - Embedding dimension: 2 and 3.
  - Perplexity: $5, 10, 20$ and $40$.

- Feature Partition (Ours):

  - Perplexity: 40
  - Simulations: 100

- tSNE based on the Feature Partition (Ours):

- – Perplexity for Feature Partition: 40
- – Simulations for Feature Partition: 100
- – Perplexity for tSNE: 40
- – Embedding dimension: 3

### F.3. Data Preparation and Comparative Visualizations for Section 5.3

In Section 5.3, we applied our approach to the dataset used in (Droin et al., 2021). The data included a count data matrix comprised of 6889 cells along with their 14812 gene expressions. We employed the normalization scheme from the Seurat package, where the gene counts in each cells were divided by the total number of genes within that cell and then scaled by 10000. Each entry was subsequently log-normalized using the transformation $\log(1 + x)$. We then retained 2000 of the most highly variable genes. Finally, we used the z-score transformation across the samples and applied principal components analysis (PCA) to reduce the data to dimension 20. As for our method, we applied Algorithm 1 using a perplexity of 40 and 100 simulations. All tSNE embeddings used a perplexity of 40.

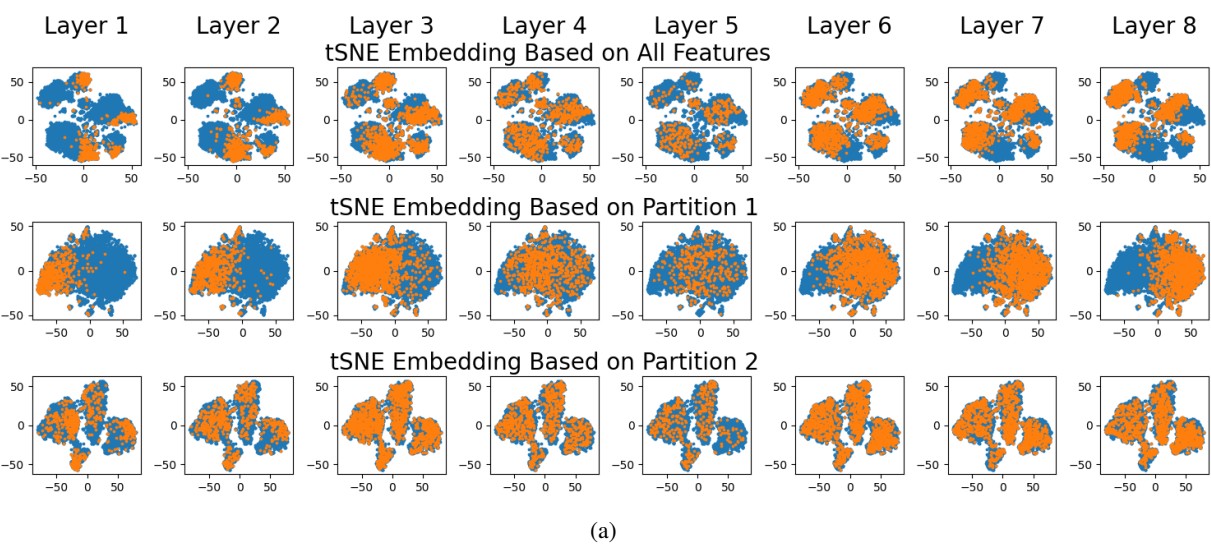

(a)

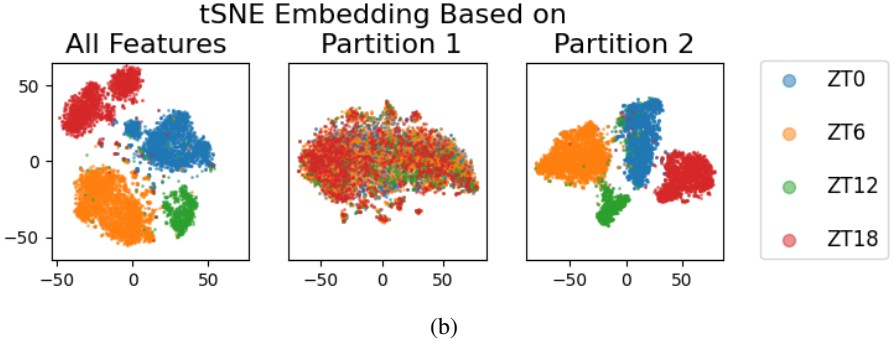

(b)

*Figure 12.* tSNE embeddings generated using all the features and using the features from our proposed partitioning, applied to the liver dataset described in Appendix F.3. (a) Embedded data points (blue) are overlaid with the labels of specific liver layers (orange). (b) Embedded data points are colored according to their circadian clock phase. Our method effectively disentangles the two underlying factors: partition 1 captures the spatial liver layer structure while partition 2 captures the circadian phase. In contrast, the tSNE embedding using all features reflects a complex mixture of both variables, making it difficult to interpret the embedding.

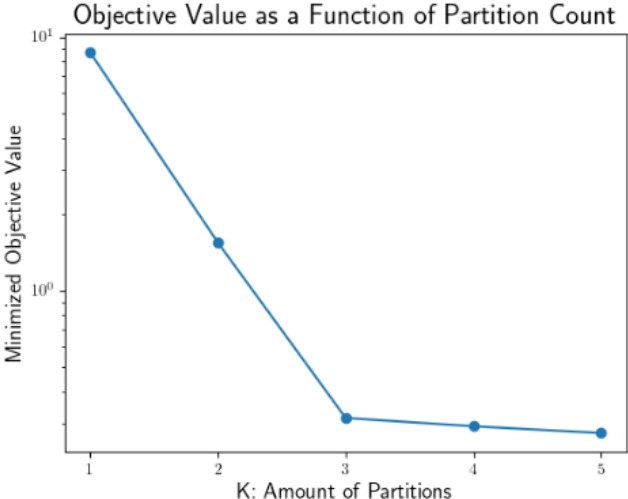

*Figure 13.* The minimal objective value achieved by our algorithm (defined in (8)), normalized by the number of samples $N$, as a function of the number of partitions $K$. Results are shown for the dataset of three rotating figurines used in Appendix D.

## G. Choosing the Number of Partitions ($K$) and Validating the Usefulness of Our Partitioning

The selection of the number of partitions ($K$) is a practical consideration that directly affects the outcome of our algorithm and therefore the visualizations generated based on it. A sub-optimal choice of $K$ can result in visualizations that are either overly complex or redundant. In this section, we discuss how to determine $K$ and discuss the implications of selecting a sub-optimal number of partitions. Finally, we propose a procedure to verify whether the data is indeed composed of subsets of features that are partially dependent as our approach assumes.

**Determine $K$ for $K \geq 2$.** In order to determine $K$ we propose to inspect the behavior of the smoothness score from our approach, defined in (8), as a function of the number of partitions $K$. As long as $K$ increases but remains lower than the true number of partitions, we expect to see a rapid decay of the score. Then, when $K$ reaches the true number of partitions, we expect to see an 'elbow', after which the score saturates or decays very slowly. This phenomenon is reminiscent of similar situations, such as manually selecting the number of clusters in k-means or the number of components in PCA. We demonstrate this behavior of the smoothness score in Figure 13.

Next, we discuss the expected impact of over-selecting or under-selecting $K$ on the resulting representation and embedding. If the number of partitions is lower than the number of true partitions in the data, we expect the partitions to separate the features into super-groups that correspond to the most distinct geometric structures, even if some of these groups could be further subdivided. In this case, each group of features describes a geometry that is simpler than that of the original data, i.e., it has a lower intrinsic dimension. Hence, the outcome of our procedure in this case still improves the ability to embed and visualize the data in a low-dimensional space, albeit sub-optimally — where the embedding dimension may need to be higher than it would be if the optimal choice of $K$ was used.

If the number of partitions is slightly larger than the true number of partitions in the data, we expect that one or more of the true partitions will be arbitrarily subdivided. This will introduce redundancy into the representation, where some feature subsets will exhibit similar geometric structures. The user should take into account this possibility and inspect the resulting embeddings for potential redundancy. Nonetheless, the partitioning is still beneficial in this case, since the data for each partition can be easily embedded and visualized in a low-dimensional space. If the number of partitions is grossly overestimated, then in addition to redundancy, some of the feature subsets may not reliably represent the underlying geometry of any of the true feature subsets.

**Determine Whether $K = 1$.** When the data cannot be partitioned into features with simpler structures, we expect that applying our algorithm to partition the features into two or more groups will result in partitions with similar or nearly identical underlying structures. Consequently, the graphs that correspond to the partitions will also exhibit similar characteristics, leading to embeddings (such as tSNE) that closely resemble each other.

To determine if the features can be partitioned into meaningful subsets, we propose a type of a permutation test. Specifically, we propose to compare the smoothness score produced by our method from the original data to the analogous score obtained from manipulated versions of the data. The manipulated versions are obtained by applying a random orthogonal transformation to the features, in which case the data does not satisfy our underlying assumption by design. The different steps of this procedure and their justification are detailed below:

1. Apply our procedure with $K = 2$ to the given data matrix and store the associated score, defined in (8).

2. Generate multiple modified versions of the dataset by applying random orthogonal transformation to the features of the data. Each orthogonal transformation randomly mixes all the features in the dataset. If distinct feature partitions existed in the original data, they will be completely mixed after the transformation. Hence, our underlying assumption does not hold for these manipulated datasets.

3. Apply our procedure with $K = 2$ to each transformed dataset and store the associated scores, defined in (8).

4. Compare the score from step 1 to the distribution of the scores from step 3. If the score from step 1 is smaller than a chosen percentile of the scores from step 3 (e.g., 0.01), we conclude that the original data contains feature partitions with significantly distinct structures, suggesting that our assumption holds, at least to some extent.

## H. Experiment on the COIL-20 Dataset

We conducted an additional experiment using a subset of the COIL-20 dataset (Nene et al., 1996), consisting of $128 \times 128$ grayscale images of three distinct but visually similar cars captured at varying azimuths. While this dataset may not explicitly satisfy our approach's theoretical assumptions, it provides an interesting test case for our approach. For this experiment, we used all images from the car objects $2, 5, 18$, which resemble one another, resulting in 216 images in total. The images were flattened to vectors and the resulting dataset was approximated using its 20 leading SVD components to reduce its variability.

We applied our algorithm with $K = 2$ partitions and two different perplexity parameters: 10 and 20. The resulting tSNE visualizations can be seen in Figure 14. Evidently, the standard tSNE visualization that use all pixels (left) reveals a closed loop structure corresponding to the object's azimuth. Notably, partition 1 (middle) produces a more coherent representation of the azimuth, demonstrating also enhanced stability across different perplexity values. Interestingly, partition 2 (right) effectively separates the images based on car identity, suggesting our method naturally discovered semantically meaningful features. The actual pixel partitions are shown in Figure 14(c). Specifically, partition 1 identified the outer pixels that typically capture the front and back of the car—features that correspond more closely to the azimuth of the object rather to its identity, as reflected in Figures 14(a) and 14(b). In contrast, partition 2 focused on the rooftop region, which corresponds more directly to the identity of each specific car object.

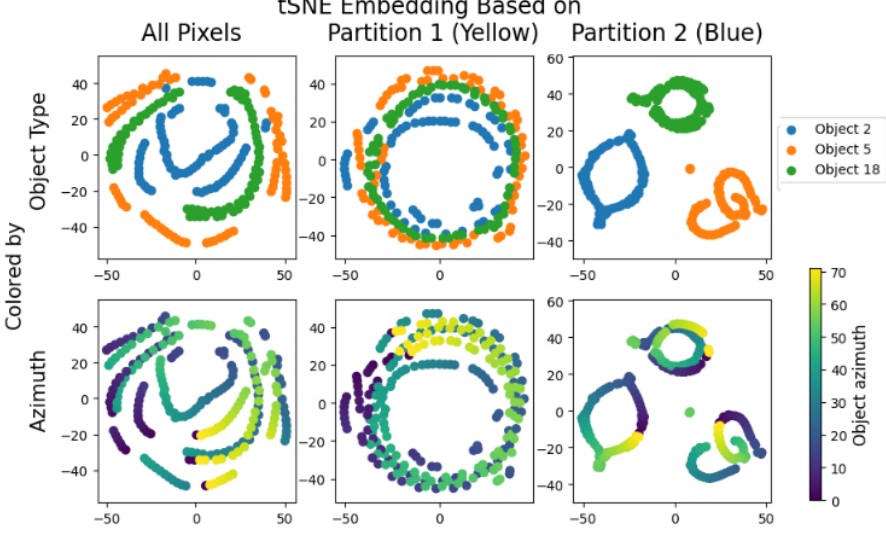

(a) tSNE embeddings with perplexity parameter of 10

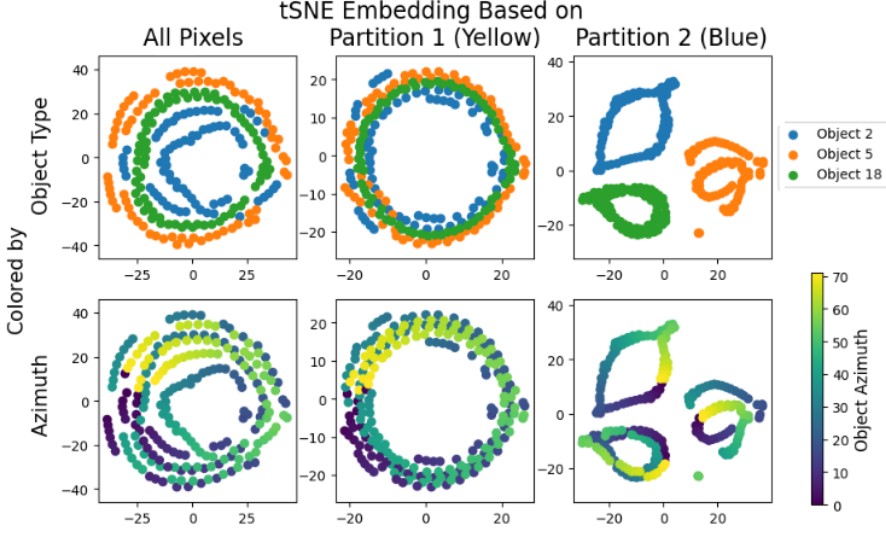

(b) tSNE embeddings with perplexity parameter of 20

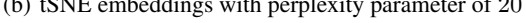

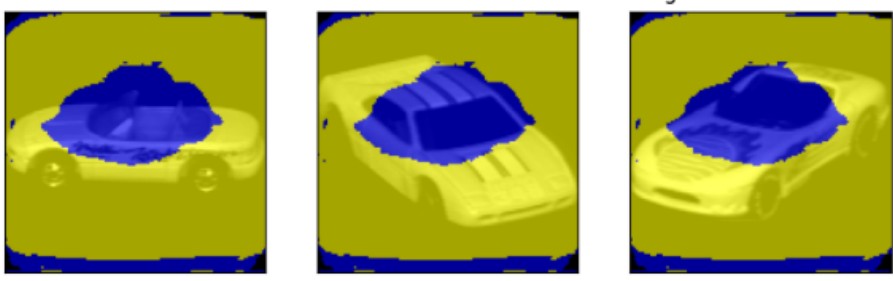

(c) Extracted partitions overlaid in blue and yellow on three sample images.

*Figure 14.* The experiment detailed in Appendix H. The dataset includes $N = 216$ gray-scale images with $D = 16,384$ pixels from COIL-20. Each image contains a single rotated object, and the considered objects are three types of cars. (a) tSNE embedding of the data with a perplexity parameter of 10, (b) tSNE embedding of the data with a perplexity parameter of 20. (c) Three sample images from the dataset, overlaid with the partitions extracted by Algorithm 1, shown in yellow (partition 1) and blue (partition 2).

# I. Proofs

*Proofs overview.*

This section is organized into three main parts: Appendix I.1 contains the proofs of the results in Section 3, Appendix I.2 presents the proofs of the results in Section 4, and Appendix I.3 includes the proofs of the results in Appendix C.

Below is a brief outline of the results and proofs presented in Appendix I.1:

1. **Lemma I.1**. This is an auxiliary lemma used in Proposition 3.1 and Lemma I.3 that characterizes the entropy function of a discrete distribution. Specifically, it considers a normalized exponential distribution that is based on the pairwise distances within the data.

2. **The proof of Proposition 3.1**. The proof begins by showing that the problem is convex. Then, it derives an optimal solution to the optimization problem under some assumptions of the data by using the Karush-Kuhn-Tucker (KKT) conditions for convex optimization problems using Lemma I.1. Finally, we derive an optimal solution that admits the same form as the previous one when the assumptions do not hold.

3. **Lemma I.2**. This is an auxiliary lemma used in Proposition 3.2 that approximates specific integrals over a manifold in terms of the manifold's properties and density. The lemma uses results from (Singer, 2006).

4. **The proof of Proposition 3.2**. The proof begins by asymptotically approximating the function using results from (Hein et al., 2005). Then, it derives its expression using Lemma I.2 in terms of the manifold's characteristics and density. Finally, a further simplification of the expression is done by referring to results from (Osher et al., 2017).

5. **Lemma I.3**. This is an auxiliary lemma used in Proposition 3.4 that characterizes the optimal graphs in a Problem 3.3 with fixed partitions. The proof begins by showing that the problem is convex. Then, it derives an optimal solution to the optimization problem under some assumptions of the data by using the Karush-Kuhn-Tucker (KKT) conditions for convex optimization problems using Lemma I.1. Finally, we derive an optimal solution that admits the same form as the previous one when the assumptions do not hold.

6. **Lemma I.4**. This is an auxiliary lemma used in Proposition 3.4 that characterizes the optimal partitions in a Problem 3.3 with fixed graphs.

7. **The proof of Proposition 3.4**. The proof divides the problem into two sub-problems, fixing either the feature partitions or their associated graphs in each case. Then it combines the results from Lemmas I.3 and I.4 to fully characterize the problem's optimal solutions.

Similarly, below is an outline of the results and proofs presented in Appendix I.2:

1. **The proof of Corollary 4.2.** The proof begins by proving that the optimization problem is convex. Then, it derives an optimal solution to the optimization problem under some assumptions of the data by using the Karush-Kuhn-Tucker (KKT) conditions for convex optimization problems using Lemma I.1. Finally, we derive an optimal solution that admits the same form as the previous one when the assumptions do not hold.

2. **Lemma I.5**. This is an auxiliary lemma used in Lemma I.6 and in Lemma I.7 to generate an upper bound for a given event by separating it into multiple s events.

3. **Lemma I.6**. This is an auxiliary lemma used in Theorem 4.3 that characterizes a specific sum in terms of the data setup parameters and partition parameters defined in Section 4.

4. **Lemma I.7.** This is an auxiliary lemma used in Theorem 4.3 that characterizes a specific sum via an integral, in terms of the data setup parameters and partition parameters as defined in Section 4.

5. **Lemma I.8**. This is an auxiliary lemma used in Lemma I.9 that approximates specific integrals over a manifold in terms of the manifold's properties and density and some predefined parameters. Its proof uses results from Lemma I.2.

6. **Lemma I.9.** This is an auxiliary lemma used in Theorem 4.3 that approximates specific integrals over a manifold in terms of the manifold's properties, density and some predefined parameters. Each of the integrals under consideration includes a nested integral, with the inner integral being addressed in Lemma I.8.

7. **The proof of Theorem 4.3.** The proof begins by solving the minimization problem over the graph parameters derived in Corollary 4.2. It then asymptotically approximates the solution as the dimension and number of samples tends to infinity, utilizing Lemmas I.6 and I.7. Next, it approximates this asymptotic value by applying Lemma I.9 to express it in terms of the manifold's properties, density, and partition parameters.

8. **Lemma I.10.** This auxiliary lemma, used in Theorem 4.4, considers the case of two partitions. It demonstrates that the analytical form derived in Theorem 4.3 is minimized when the partitioning is close to the true feature partitioning under a constraint on the possible partitioning solutions.

9. **The proof of Theorem 4.4.** The proof builds on Lemma I.10 by considering a sequence of problems and demonstrates that the problem is minimized when the true partitioning solution is used.

Finally, below we outline the results and proofs presented in Appendix I.3:

1. **Lemma I.11.** This is an auxiliary lemma used in Proposition C.2 that characterizes the optimal graphs in a Problem C.1 with fixed partitions. The proof begins by showing that the problem is convex. Then, it derives an optimal solution to the optimization problem under some assumptions of the data by using the Karush-Kuhn-Tucker(KKT) conditions for convex optimization problems using Lemma I.1. Finally, we derive an optimal solution that admits the same form as the previous one when the assumptions do not hold.

2. **The proof of Proposition C.2.** The proof divides the problem into two sub-problems, fixing either the feature partitions or their associated graphs in each case. Then it combines the results from Lemma I.3 from Section Appendix I.1 and Lemma I.11 to achieve a full characterization of optimal solutions to the problem.

3. **The proof of Proposition C.3.** The proof begins by evaluating the objective of Problem C.1 under a specific regularization parameter, using the soft uniform partitioning and its corresponding affinity matrix as defined in Proposition C.2. It then upper bounds this value by the objective achieved by any hard partitioning solution.

4. **The proof of Proposition C.4.** The proof analyzes the computational complexity of each update rule proposed in Proposition C.2.

5. **The proof of Proposition C.5.** The proof analyzes the computational complexity of each update rule proposed in Proposition C.2 when the data is given the form of singular value approximation. It starts by rewriting each term as a multiplication of matrices and then derives the computational complexity by considering each multiplication within.

## I.1. Proofs of Section 3.1 and Section 3.2

**Lemma I.1.** *Let $i \in \{1, \ldots, N\}$, and let $\Delta_1, \ldots, \Delta_N \in [0, \infty)$ be some distance function between the $i$-th element and all the other elements. Define the function $f : [0, \infty) \to \mathbb{R}$ by*

$$f(\beta) = \sum_{j \in \{1,\ldots,N\}/\{i\}} \frac{\exp(-\Delta_j \beta)}{\sum_{t \in \{1,\ldots,N\}/\{i\}} \exp(-\Delta_t \beta)} \log \frac{\exp(-\Delta_j \beta)}{\sum_{s \in \{1,\ldots,N\}/\{i\}} \exp(-\Delta_s \beta)}. \tag{31}$$

*Then $f$ is a non-decreasing function and satisfies*

$$\lim_{\beta \to 0} f(\beta) = -\log(N-1) \qquad and \qquad \lim_{\beta \to \infty} f(\beta) = -\log(\tilde{\alpha}_N), \tag{32}$$

*where $\tilde{\alpha}_i = |\{j \in \{1, \ldots, N-1\} \mid \Delta_j = \min_{s \in \{1,\ldots,N\}/\{i\}} \Delta_s\}$. Furthermore, for any $j \in \{1, \ldots, N\}/\{i\}$ we get that*

$$\lim_{\beta \to \infty} \frac{\exp(-\Delta_j \beta)}{\sum_{t \in \{1,\ldots,N\}/\{i\}} \exp(-\Delta_t \beta)} \log \frac{\exp(-\Delta_j \beta)}{\sum_{s \in \{1,\ldots,N\}/\{i\}} \exp(-\Delta_s \beta)} \tag{33}$$

$$= \begin{cases} \frac{1}{\tilde{\alpha}_N} & if \Delta_j = \tilde{\alpha}_N \\ 0 & else \end{cases}.$$

*Proof.* We indicate that this proof is not new, and is shown here for completeness. Without loss of generality the proof will assume $i = N$. Therefore, $\tilde{\alpha}_N$ that will be used throughout the proof will be related to the $i$-th element.

The proof will begin by bounding the first derivative of $f(\beta)$ between zero and some positive constant for all $\beta \in [0, \infty)$, and therefore show that it is non-decreasing. Then, to bound the image of $f$ it will derive the value of $f$ at the boundaries of the domain.

We begin by deriving the bounds of the derivative of $f$. To do so, we rewrite $f$ by

$$f(\beta) = \sum_{j=1}^{N-1} \left( \frac{\exp(-\Delta_j \beta)}{\sum_{t=1}^{N-1} \exp(-\Delta_t \beta)} \cdot \log \frac{\exp(-\Delta_j \beta)}{\sum_{s=1}^{N} \exp(-\Delta_s \beta)} \right) \tag{34}$$

$$= \left( \sum_{j=1}^{N-1} \frac{\exp(-\Delta_j \beta)}{\sum_{t=1}^{N-1} \exp(-\Delta_t \beta)} \cdot \log(\exp(-\Delta_j \beta)) \right) \tag{35}$$

$$- \left( \sum_{j=1}^{N-1} \frac{\exp(-\Delta_j \beta)}{\sum_{t=1}^{N-1} \exp(-\Delta_t \beta)} \cdot \log(\sum_{s=1}^{N-1} \exp(-\Delta_s \beta)) \right)$$

$$= \left( \sum_{j=1}^{N-1} \frac{\exp(-\Delta_j \beta)}{\sum_{t=1}^{N-1} \exp(-\Delta_t \beta)} \log(\exp(-\Delta_j \beta)) \right) \tag{36}$$

$$- \left( \frac{\sum_{j=1}^{N-1} \exp(-\Delta_j \beta)}{\sum_{t=1}^{N-1} \exp(-\Delta_t \beta)} \right) \log(\sum_{s=1}^{N-1} \exp(-\Delta_s \beta))$$

$$= -\beta \left( \sum_{j=1}^{N-1} \frac{\exp(-\Delta_j \beta)}{\sum_{t=1}^{N-1} \exp(-\Delta_t \beta)} \cdot \Delta_j \right) - \log(\sum_{s=1}^{N-1} \exp(-\Delta_s \beta)). \tag{37}$$

The derivative of $f$ is non negative as

$$\frac{d}{d\beta} f(\beta) \tag{38}$$

$$= - \left( \sum_{j=1}^{N-1} \frac{\exp(-\Delta_j \beta)}{\sum_{t=1}^{N-1} \exp(-\Delta_t \beta)} \cdot \Delta_j \right) \tag{39}$$

$$- \beta \sum_{j=1}^{N-1} \Delta_j \cdot \frac{\exp(-\Delta_j \beta)(-\Delta_j) \sum_{r=1}^{N-1} \exp(-\Delta_r \beta) - \exp(-\Delta_j \beta) \left( \sum_{s=1}^{N-1} \exp(-\Delta_s \beta)(-\Delta_s) \right)}{(\sum_{t=1}^{N-1} \exp(-\Delta_t \beta))^2}$$

$$- \frac{\sum_{r=1}^{N-1} \exp(-\Delta_r \beta)(-\Delta_r)}{\sum_{s=1}^{N-1} \exp(-\Delta_s \beta)}$$

$$= - \left( \sum_{j=1}^{N-1} \frac{\exp(-\Delta_j \beta)}{\sum_{t=1}^{N-1} \exp(-\Delta_t \beta)} \cdot \Delta_j \right) \tag{40}$$

$$- \beta \sum_{j=1}^{N-1} \Delta_j \exp(-\Delta_j \beta) \left( -\frac{\Delta_j}{\sum_{t=1}^{N-1} \exp(-\Delta_t \beta)} + \frac{\sum_{s=1}^{N-1} \exp(-\Delta_s \beta) \Delta_s}{(\sum_{t=1}^{N-1} \exp(-\Delta_t \beta))^2} \right)$$

$$+ \frac{\sum_{r=1}^{N-1} \exp(-\Delta_r \beta) \Delta_r}{\sum_{s=1}^{N-1} \exp(-\Delta_s \beta)}$$

$$= \beta \sum_{j=1}^{N-1} \Delta_j \exp(-\Delta_j \beta) \left( \frac{\Delta_j}{\sum_{t=1}^{N-1} \exp(-\Delta_t \beta)} - \frac{\sum_{s=1}^{N-1} \exp(-\Delta_s \beta) \Delta_s}{(\sum_{t=1}^{N-1} \exp(-\Delta_t \beta))^2} \right) \tag{41}$$

$$= \beta \left( \sum_{j=1}^{N-1} \Delta_j^2 \cdot \frac{\exp(-\Delta_j \beta)}{\sum_{t=1}^{N-1} \exp(-\Delta_t \beta)} \right) - \beta \left( \sum_{j=1}^{N-1} \sum_{s=1}^{N-1} \Delta_j \Delta_s \cdot \frac{\exp(-\Delta_j \beta) \exp(-\Delta_s \beta)}{(\sum_{t=1}^{N-1} \exp(-\Delta_t \beta))^2} \right) \tag{42}$$

$$= \beta \left( \sum_{j=1}^{N-1} \Delta_j^2 \frac{\exp(-\Delta_j \beta)}{\sum_{t=1}^{N-1} \exp(-\Delta_t \beta)} \right) - \beta \left( \sum_{j=1}^{N-1} \Delta_j \frac{\exp(-\Delta_j \beta)}{\sum_{t=1}^{N-1} \exp(-\Delta_t \beta)} \right)^2 \tag{43}$$

$$= \beta \cdot \sum_{j=1}^{N-1} \frac{\exp(-\Delta_j \beta)}{\sum_{t=1}^{N-1} \exp(-\Delta_t \beta)} \cdot \left( \Delta_j - \left( \sum_{r=1}^{N-1} \Delta_r \frac{\exp(-\Delta_r \beta)}{\sum_{s=1}^{N-1} \exp(-\Delta_s \beta)} \right) \right)^2 \tag{44}$$

$$\geq 0. \tag{45}$$

where in (43) we use the variance decomposition into moments ($Var(\|\boldsymbol{X}\|^2) = E[\|\boldsymbol{X}\|^4] - \mathbb{E}[\|\boldsymbol{X}\|^2]^2$ for some random variable $\boldsymbol{X}$) and in (44) we use the fact that all the elements are non-negative.

The derivative of $f$ can be bounded from above as well as will derived below. Based on (44), we can see that

$$\frac{d}{d\beta} f(\beta) \quad = \quad \beta \cdot \sum_{j=1}^{N-1} \frac{\exp(-\Delta_j \beta)}{\sum_{t=1}^{N-1} \exp(-\Delta_t \beta)} \cdot \left( \Delta_j - \left( \sum_{r=1}^{N-1} \Delta_r \frac{\exp(-\Delta_r \beta)}{\sum_{s=1}^{N-1} \exp(-\Delta_s \beta)} \right) \right)^2 \tag{46}$$

$$= \quad \beta \left( \sum_{j=1}^{N-1} \frac{\exp(-\Delta_j \beta)}{\sum_{t=1}^{N-1} \exp(-\Delta_t \beta)} \Delta_j^2 - \left( \sum_{j=1}^{N-1} \frac{\exp(-\Delta_j \beta)}{\sum_{t=1}^{N-1} \exp(-\Delta_t \beta)} \Delta_j \right)^2 \right) \tag{47}$$

$$\leq \quad \beta \sum_{j=1}^{N-1} \frac{\exp(-\Delta_j \beta)}{\sum_{t=1}^{N-1} \exp(-\Delta_t \beta)} \Delta_j^2 \tag{48}$$

$$\leq \quad \max_{j \in \{1, \ldots N-1\}} \beta \Delta_j^2. \tag{49}$$

Since the derivative of $f$ is nonnegative, its image is bounded by the values it achieves on the boundary of its domain. To find these values, we examine each normalized exponential $j \in \{1, \ldots, N-1\}$ within $f$

$$\frac{\exp(-\beta \Delta_j)}{\sum_{t=1}^{N-1} \exp(-\beta \Delta_t)} = \frac{1}{1 + \sum_{t \in \{1, \ldots, N-1\}/\{j\}} \exp((\Delta_j - \Delta_t)\beta)} \in [0, 1]. \tag{50}$$

Its limiting value when $\beta$ tends to zero is

$$\lim_{\beta \to 0_+} \frac{\exp(-\Delta_j \beta)}{\sum_{t=1}^{N-1} \exp(-\Delta_t \beta)} \quad = \quad \lim_{\beta \to 0_+} \frac{1}{\sum_{t=1}^{N-1} \exp((\Delta_j - \Delta_t)\beta)} \tag{51}$$

$$= \quad \frac{1}{N-1}. \tag{52}$$

To derive its value when $\beta \to \infty$, we define $I_{\min} = \{j \in \{1, \ldots, N-1\} : \Delta_j = \min_{s \in \{1, \ldots, N-1\}} \Delta_s\}$. In this limiting case each term converges to

$$\lim_{\beta \to \infty} \frac{\exp(-\Delta_j \beta)}{\sum_{t=1}^{N-1} \exp(-\Delta_t \beta)} \tag{53}$$

$$= \quad \lim_{\beta \to \infty} \frac{1}{\sum_{t=1}^{N-1} \exp((\Delta_j - \Delta_t)\beta)} \tag{54}$$

$$= \quad \lim_{\beta \to \infty} \frac{1}{\sum_{t \in I_{\min}} \exp((\Delta_j - \Delta_t)\beta) + \sum_{\tilde{t} \notin I_{\min}} \exp((\Delta_j - \Delta_{\tilde{t}})\beta)} \tag{55}$$

$$= \quad \begin{cases} \frac{1}{|I_{\min}| + (N-1-|T|) \cdot 0} & \text{if } j \in I_{\min} \\ 0 & else \end{cases} \tag{56}$$

$$= \quad \begin{cases} \frac{1}{|I_{\min}|} & \text{if } j \in I_{\min} \\ 0 & \text{else} \end{cases}, \tag{57}$$

as $\Delta_j - \Delta_t > 0$ for any $j \notin I_{\min}$ and $t \in I_{\min}$.

To connect this result with $f$, we define $h : [0, 1] \to (-\infty, 0]$ by $h(a) = a \log(a)$ for all $a \in [0, 1]$. By composing it with any of the normalized exponentials from above we get that

$$\lim_{\beta \to 0_+} \frac{\exp(-\Delta_j\beta)}{\sum_{t=1}^{N-1} \exp(-\Delta_t\beta)} \log \frac{\exp(-\Delta_j\beta)}{\sum_{s=1}^{N-1} \exp(-\Delta_s\beta)} = \frac{1}{N} \log \frac{1}{N} \qquad j \in \{1, \ldots, N-1\} \tag{58}$$

$$\lim_{\beta \to \infty} \frac{\exp(-\Delta_j\beta)}{\sum_{t=1}^{N-1} \exp(-\Delta_t\beta)} \log \frac{\exp(-\Delta_j\beta)}{\sum_{s=1}^{N-1} \exp(-\Delta_s\beta)} = \frac{1}{I_{\min}} \log \frac{1}{I_{\min}} \qquad j \in I_{\min}, \tag{59}$$

where we apply the property that the limit of a product of two functions is the product of each function's limit, given that both exists. Furthermore, when $\beta$ tends to infinity, the limiting value of the term for any $j \notin I_{\min}$ we converge to

$$\lim_{\beta \to \infty} \frac{\exp(-\Delta_j\beta)}{\sum_{t=1}^{N-1} \exp(-\Delta_t\beta)} \log \frac{\exp(-\Delta_j\beta)}{\sum_{s=1}^{N-1} \exp(-\Delta_s\beta)} \tag{60}$$

$$= \lim_{\beta \to \infty} -\frac{\log \sum_{s=1}^{N-1} \exp((\Delta_j - \Delta_s)\beta)}{\sum_{t=1}^{N-1} \exp((\Delta_j - \Delta_t)\beta)} \tag{61}$$

$$= \lim_{\beta \to \infty} -\frac{\sum_{l=1}^{N-1} \exp((\Delta_j - \Delta_l)\beta) \cdot (\Delta_j - \Delta_l)}{\sum_{s=1}^{N-1} \exp((\Delta_j - \Delta_s)\beta)} \cdot \frac{1}{\sum_{t=1}^{N-1} \exp((\Delta_j - \Delta_t)\beta) \cdot (\Delta_j - \Delta_t)} \tag{62}$$

$$= \lim_{\beta \to \infty} -\frac{1}{\sum_{s=1}^{N-1} \exp((\Delta_j - \Delta_s)\beta)} \tag{63}$$

$$= \lim_{\beta \to \infty} -\frac{\exp(-\Delta_j\beta)}{\sum_{s=1}^{N-1} \exp(-\Delta_s\beta)} \tag{64}$$

$$= 0 \tag{65}$$

where we use the L'Hopital's rule and the result from (53).

Now, we can rewrite $f$ as a sum of the components defined above by

$$f(\beta) = \sum_{j=1}^{N-1} h\left(\frac{\exp(-\beta\Delta_j)}{\sum_{t=1}^{N-1} \exp(-\beta\Delta_t)}\right), \tag{66}$$

and derive it value at the boundary.

$$\lim_{\beta \to 0_+} f(\beta) = \lim_{\beta \to 0_+} \sum_{j=1}^{N-1} h\left(\frac{\exp(-\beta\Delta_j)}{\sum_{t=1}^{N-1} \exp(-\beta\Delta_t)}\right) \tag{67}$$

$$= \sum_{j=1}^{N-1} \lim_{\beta \to 0_+} h\left(\frac{\exp(-\beta\Delta_j)}{\sum_{t=1}^{N-1} \exp(-\beta\Delta_t)}\right) \tag{68}$$

$$= \log\left(\frac{1}{N-1}\right) \tag{69}$$

$$\lim_{\beta \to \infty} f(\beta) = \lim_{\beta \to \infty} \sum_{j=1}^{N-1} h\left(\frac{\exp(-\beta\Delta_j)}{\sum_{t=1}^{N-1} \exp(-\beta\Delta_t)}\right) \tag{70}$$

$$= \sum_{j=1}^{N-1} \lim_{\beta \to \infty} h\left(\frac{\exp(-\beta\Delta_j)}{\sum_{t=1}^{N-1} \exp(-\beta\Delta_t)}\right) \tag{71}$$

$$= \log\left(\frac{1}{|I_{\min}|}\right), \tag{72}$$

where we apply the property that the limit of a sum of functions is the sum of each function's limit, given that they exists. □

**Proposition 3.1** The matrix $\boldsymbol{W} \in [0, 1]^{N \times N}$ defined in (1) is a solution to

$$\underset{\tilde{\boldsymbol{W}} \in [0,1]^{N \times N}}{\arg\min} \quad J(\tilde{\boldsymbol{W}}, \{\boldsymbol{y}_1, \ldots, \boldsymbol{y}_N\}), \tag{5}$$

subject to the constraints $\tilde{W}_{i,i} = 0$, $\sum_{j=1}^{N} \tilde{W}_{i,j} = 1$ and $\sum_{j=1}^{N} \tilde{W}_{i,j} \log \tilde{W}_{i,j} \leq -\log(\alpha)$ for all $i \in \{1, \ldots, N\}$, where $\epsilon_1, \ldots, \epsilon_N \in \mathbb{R}_+$ from (1) are the minimum values that satisfy the entropy constraint.

*Proof of Proposition 3.1.* The proof will demonstrate that the optimal solution can be derived from solving $N$ subproblems, each corresponding to a row of the graph solution. We will then use convexity conditions to derive the optimal solution under certain assumptions on the data. Finally, we will show that if these assumptions do not hold, the solution takes a similar form, but with bandwidth parameters approaching zero.

We begin by showing that the problem can be solved using $N$ subproblems separately. For each $i \in \{1, \ldots, N\}$ we define a subproblem related to the $i - th$ row of the graph matrix

$$\underset{\tilde{W}_{i,1}, \ldots, \tilde{W}_{i,N} \in [0,1]}{\arg\min} J_i(\{\tilde{W}_{i,j}\}_{j=1}^{N}, \{\boldsymbol{y}_j\}_{j=1}^{N}) \tag{73}$$

subject to the constraints $\tilde{W}_{i,i} = 0$, $\sum_{j=1}^{N} \tilde{W}_{i,j} = 1$ and $\sum_{j=1}^{N} \tilde{W}_{i,j} \log \tilde{W}_{i,j} \leq -\log(\alpha)$, where

$$J_i(\{\tilde{W}_{i,j}\}_{j=1}^{N}, \{\boldsymbol{y}_j\}_{j=1}^{N}) = \sum_{j=1}^{N} \tilde{W}_{i,j} \|\boldsymbol{y}_i - \boldsymbol{y}_j\|^2. \tag{74}$$

Based on the next derivation, we can see that the optimal solution should be optimal for each of these sub-problems

$$\underset{\tilde{\boldsymbol{W}} \in [0,1]^{N \times N}}{\min} J(\tilde{\boldsymbol{W}}, \{\boldsymbol{y}_j\}_{j=1}^{N}) \tag{75}$$

$$= \underset{\tilde{\boldsymbol{W}} \in [0,1]^{N \times N}}{\min} \sum_{i,j=1}^{N} \tilde{W}_{i,j} \|\boldsymbol{y}_i - \boldsymbol{y}_j\|^2 \tag{76}$$

$$= \sum_{i=1}^{N} \underset{\tilde{\boldsymbol{W}}_{i,:} \in [0,1]^{N}}{\min} \sum_{j=1}^{N} \tilde{W}_{i,j} \|\boldsymbol{y}_i - \boldsymbol{y}_j\|^2 \tag{77}$$

$$= \sum_{i=1}^{N} \underset{\tilde{\boldsymbol{W}}_{i,:} \in [0,1]^{N}}{\min} J_i(\{\tilde{W}_{i,j}\}_{j=1}^{N}, \{\boldsymbol{y}_j\}_{j=1}^{N}), \tag{78}$$

where $\tilde{\boldsymbol{W}}_{i,:}^{(k)}$ denotes the $i$-th row of $\tilde{\boldsymbol{W}}^{(k)}$, and the domain of each minimization problem above contains the constraints that $\tilde{W}_{i,i} = 0$, $\sum_{j=1}^{N} \tilde{W}_{i,j} = 1$ and $\sum_{j=1}^{N} \tilde{W}_{i,j} \log \tilde{W}_{i,j} \leq -\log(\alpha)$ for all $i \in \{1, \ldots, N\}$.

Without loss of generality, we are going to find the optimal $W_{i,1}, \ldots, W_{i,N}$ that minimize the $J_i$ for some $i \in \{1, \ldots, N\}$. Specifically, we assume $W_{i,i} = 0$ based on the equality to zero constraint, and omit this constraint. To find the optimal solution, we begin by building on the Karush-Kuhn-Tucker (KKT) Theorem (Corollary 28.3.1 in (Rockafellar, 1970)). We note that this solution will depend on certain conditions; therefore, after this derivation, we will provide an alternative solution that attains a similar form when these conditions are not met.

The theorem assumes that the objective and the inequality constraint is convex and that the equality constraint is an affine function. Furthermore, it assumes that there exists a solution within the domain that satisfies the inequality constraint in a strict manner (Slater's conditions). We begin with the latter assumption, we define a valid solution by $\tilde{W}_{i,j} = 1/(N - 1)$ for any $j \neq i$ and $\tilde{W}_{i,i} = 0$, as it sums up to ones. The entropy constraint is strictly satisfied $\sum_{j=1}^{N} \tilde{W}_{i,j} \log \tilde{W}_{i,j} = -\log(N - 1) < -\log(\alpha)$, and therefore Slater's conditions are satisfied.

The objective $J_i$ is an affine function and therefore is convex, and the equality constraint $\sum_{j=1}^{N} \tilde{W}_{i,j} = 1$ is indeed affine. Finally, we will show that the constraint of the entropy inequality is convex by showing that its Hessian is a positive semi-definite matrix (Theorem 4.5 in (Rockafellar, 1970)). Specifically, the Hessian elements are

$$\frac{d^2}{d^2 \tilde{W}_{i,j}} \sum_{t=1}^{N} \tilde{W}_{i,t} \log(\tilde{W}_{i,t}) = \frac{d}{d\tilde{W}_{i,j}} \left(1 + \log(\tilde{W}_{i,j})\right) \tag{79}$$

$$= \frac{1}{\tilde{W}_{i,j}} \tag{80}$$

$$\geq \quad 1. \tag{81}$$

$$\frac{d}{d\tilde{W}_{i,r}} \frac{d}{d\tilde{W}_{i,j}} \sum_{t=1}^{N} \tilde{W}_{i,t} \log(\tilde{W}_{i,t}) \quad = \quad \frac{d}{d\tilde{W}_{i,r}} \left( 1 + \log(\tilde{W}_{i,j}) \right) \tag{82}$$

$$= \quad 0 \tag{83}$$

for all $j, r \in \{1, \ldots, N\}/\{i\}$ where $j \neq r$. As the Hessian is a diagonal matrix with positive values we can conclude that it is a positive definite matrix.

The KKT theorem states that a solution satisfying the KKT conditions—including stationarity, primal feasibility, dual feasibility, and complementary slackness—is an optimal solution to the problem, provided Slater's condition holds. To derive such a solution we need to first define the Lagrangian of the minimization problem by

$$\tilde{J}_i(\{\tilde{W}_{i,j}\}_{j=1}^N, \{\boldsymbol{y}_j\}_{j=1}^N) \quad \equiv \quad \sum_{j=1}^{N} \tilde{W}_{i,j} \|\boldsymbol{y}_i - \boldsymbol{y}_j\|^2 \tag{84}$$

$$+ \quad \epsilon_i \left( \log(\alpha) + \sum_{j=1}^{N} \tilde{W}_{i,j} \log(\tilde{W}_{i,j}) \right) \tag{85}$$

$$+ \quad \mu_i \left( \sum_{j=1}^{N} \tilde{W}_{i,j} - 1 \right) \tag{86}$$

where $\epsilon_i \geq 0$ and $\mu_i \in \mathbb{R}$.

A solution that satisfies the stationary condition should attain for all $j \neq i$

$$0 \quad = \quad \frac{d\tilde{J}_i(\{W_{i,j}\}_{j=1}^N, \{\boldsymbol{y}_j\}_{j=1}^N)}{dW_{i,j}} \tag{87}$$

$$= \quad \|\boldsymbol{y}_i - \boldsymbol{y}_j\|^2 + \epsilon_i \left( 1 + \log(W_{i,j}) \right) + \mu_i \tag{88}$$

$$W_{i,j} \quad = \quad \exp\left( -\frac{\|\boldsymbol{y}_i - \boldsymbol{y}_j\|^2 + \epsilon_i + \mu_i}{\epsilon_i} \right). \tag{89}$$

The primal feasibility condition on the on the equality constraint induces-

$$1 \quad = \quad \sum_{j=1}^{N} W_{i,j} \tag{90}$$

$$= \quad \sum_{j \neq i} \exp\left( -\frac{\|\boldsymbol{y}_i - \boldsymbol{y}_j\|^2 + \epsilon_i + \mu_i}{\epsilon_i} \right) \tag{91}$$

$$\exp\left( \frac{\mu_i}{\epsilon_i} \right) \quad = \quad \sum_{j \neq i} \exp\left( -\frac{\|\boldsymbol{y}_i - \boldsymbol{y}_j\|^2 + \epsilon_i}{\epsilon_i} \right). \tag{92}$$

By pushing it back into (89) we get that for all $j \neq i$

$$W_{i,j} \quad = \quad \frac{\exp\left( -\frac{\|\boldsymbol{y}_i - \boldsymbol{y}_j\|^2 + \epsilon_i}{\epsilon_i} \right)}{\sum_{t \neq i} \exp\left( -\frac{\|\boldsymbol{y}_i - \boldsymbol{y}_t\|^2 + \epsilon_i}{\epsilon_i} \right)} \tag{93}$$

$$= \quad \frac{\exp\left( -\frac{\|\boldsymbol{y}_i - \boldsymbol{y}_j\|^2}{\epsilon_i} \right)}{\sum_{t \neq i} \exp\left( -\frac{\|\boldsymbol{y}_i - \boldsymbol{y}_t\|^2}{\epsilon_i} \right)}. \tag{94}$$

Furthermore, to satisfy the primal feasibility and complementary slackness of the entropy constraint we define any $\epsilon_i$ by

$$\epsilon_i \quad s.t. \quad \sum_j W_{i,j} \log W_{i,j} = -\log(\alpha). \tag{95}$$

By incorporating the results from Lemma I.1 with $\Delta_j \equiv \|\boldsymbol{y}_i - \boldsymbol{y}_j\|^2$ for any $j \in \{1, \dots, N-1\}$, we get that the function $\sum_j W_{i,j} \log W_{i,j}$ is a non-increasing function in $\epsilon_i$. Additionally, according to the lemma, this function is bounded by $[-\log(N-1), -\log(\tilde{\alpha}_i)]$, where $\tilde{\alpha}_i \equiv |\{j \in \{1, \dots, N\}/\{i\} \ : \ \|\boldsymbol{y}_j - \boldsymbol{y}_i\| = \min_{t \neq i} \|\boldsymbol{y}_t - \boldsymbol{y}_i\|\}|$. These boundary values correspond to the limit of the function as $\epsilon_i \to \infty$ and $\epsilon_i \to 0$, respectively. Therefore a solution $\epsilon_i$ exists only if $\tilde{\alpha}_i \leq \alpha$.

Therefore, based on the Karush-Kuhn-Tucker Theorem if $\tilde{\alpha}_i \leq \alpha$ then an optimal assignment of $W_{i,1} \dots, W_{i,N}$ is as shown in (94), with a $\epsilon_i$ that satisfies (95). To address the optimal assignment when $\tilde{\alpha}_i > \alpha$ we use a direct approach that does not build on the KKT conditions. In this regime, any assignment of $\epsilon_i$ should satisfy the inequality entropy constraint, as the entropy function is bounded from above by $-\log \tilde{\alpha}_i$ (Lemma I.1). Building on the suggested solution of $W_{i,1}, \dots, W_{i,N}$, the assignment of $\epsilon_i$ that tends to 0 will result in the following solution

$$W_{i,j} = \begin{cases} 0 & \text{if } j = i \\ \frac{1}{\tilde{\alpha}_i} & \text{if } j \neq i \quad \text{and} \quad \|\boldsymbol{y}_j - \boldsymbol{y}_i\| = \min_{t \neq i} \|\boldsymbol{y}_t - \boldsymbol{y}_i\| \\ 0 & else \end{cases} . \tag{96}$$

based on Lemma I.1. This solution is valid as it satisfies the entropy constraint as $\sum_j W_{i,j} \log W_{i,j} = -\log(\tilde{\alpha}_i) \leq -\log(\alpha)$, it sums up to one, and $W_{i,i} = 0$. Now, we show that it achieves the minimal values among all solutions

$$\sum_{j=1}^{N} W_{i,j} \|\boldsymbol{y}_i - \boldsymbol{y}_j\|^2 \tag{97}$$

$$= \min_{t \neq i} \|\boldsymbol{y}_t - \boldsymbol{y}_i\|^2 \tag{98}$$

$$\leq \min_{\tilde{W}_{i,1}, \dots, \tilde{W}_{i,N} \in [0,1]: \sum_j \tilde{W}_{i,j} = 1, \tilde{W}_{i,i} = 0} \sum_{j=1}^{N} \tilde{W}_{i,j} \|\boldsymbol{y}_i - \boldsymbol{y}_j\|^2 \tag{99}$$

$$\leq \min_{\tilde{W}_{i,1}, \dots, \tilde{W}_{i,N} \in [0,1]: \sum_j \tilde{W}_{i,j} = 1, \tilde{W}_{i,i} = 0, \ \sum_j \tilde{W}_{i,j} \log \tilde{W}_{i,j} \leq -\log(\alpha)} \sum_{j=1}^{N} \tilde{W}_{i,j} \|\boldsymbol{y}_i - \boldsymbol{y}_j\|^2. \tag{100}$$

Hence, we can conclude the proof. $\qquad \square$

**Lemma I.2.** *Let $\mathcal{M} \subset \mathbb{R}^D$ be a smooth compact Riemannian manifold and let $g : \mathcal{M} \to \mathbb{R}_+$ be some smooth positive function over it. Let $\Delta_{\mathcal{M}} : \mathcal{M} \to \mathbb{R}$ be the Laplace-Beltrami operator over $\mathcal{M}$. Then, there exists $\tilde{\epsilon}(\mathcal{M}, g)$ so that for any $\epsilon < \tilde{\epsilon}(\mathcal{M}, g)$ and $\boldsymbol{x} \in \mathcal{M}$*

$$\frac{1}{C} \int_{\boldsymbol{x} \in \mathcal{M}} \exp \left( -\frac{\|\boldsymbol{x} - \boldsymbol{y}\|^2}{2\epsilon} \right) g(\boldsymbol{x}) d\boldsymbol{x} = g(\boldsymbol{y}) + \epsilon/2 (E(\boldsymbol{y})g(\boldsymbol{y}) + \Delta_{\mathcal{M}} g(\boldsymbol{y})) + O(\epsilon^2) \tag{101}$$

$$= g(\boldsymbol{y})(1 + O(\epsilon)) \tag{102}$$

$$\frac{1}{C} \int_{\boldsymbol{x} \in \mathcal{M}} \exp \left( -\frac{\|\boldsymbol{x} - \boldsymbol{y}\|^2}{2\epsilon} \right) g(\boldsymbol{x})(x_d - y_d)^2 d\boldsymbol{x} = \epsilon g(\boldsymbol{y}) \|\nabla_{\mathcal{M}} y_d\|^2 + O(\epsilon^2) \tag{103}$$

$$= \epsilon g(\boldsymbol{y}) \|\nabla_{\mathcal{M}} y_d\|^2 (1 + O(\epsilon)) \tag{104}$$

*for all $d \in \{1, \dots, D\}$, where $E(\boldsymbol{x})$ is a smooth scalar function of the curvature of $\mathcal{M}$ at $\boldsymbol{x} \in \mathcal{M}$ and $C = (2\pi\epsilon)^{dim(\mathcal{M})/2}$.*

*Proof.* The proof of (101) is given in (Singer, 2006) (see Eq. 2.11). Now we begin proving (103). As $\boldsymbol{x}$ tends to $\boldsymbol{y}$

$$\Delta_{\mathcal{M}} g(\boldsymbol{x})(x_d - y_d)^2 |_{\boldsymbol{x} = \boldsymbol{y}} \tag{105}$$

$$= g(\boldsymbol{x})(x_d - y_d) \Delta_{\mathcal{M}}(x_d - y_d) + (x_d - y_d) \Delta_{\mathcal{M}} g(\boldsymbol{x})(x_d - y_d) \tag{106}$$
$$+ 2 \langle \nabla_{\mathcal{M}} g(\boldsymbol{x})(x_d - y_d), \nabla_{\mathcal{M}}(x_d - y_d) \rangle |_{\boldsymbol{x} = \boldsymbol{y}}$$

$$= \quad g(\boldsymbol{x})(x_d - y_d)\Delta_{\mathcal{M}}x_d + (x_d - y_d)\Delta_{\mathcal{M}}g(\boldsymbol{x})(x_d - y_d) \tag{107}$$
$$+2(x_d - y_d)\langle\nabla_{\mathcal{M}}g(\boldsymbol{x}), \nabla_{\mathcal{M}}x_d\rangle + 2g(\boldsymbol{x})\|\nabla_{\mathcal{M}}x_d\|^2\big|_{\boldsymbol{x}=\boldsymbol{y}}$$

$$= \quad 2g(\boldsymbol{x})\|\nabla_{\mathcal{M}}x_d\|^2, \tag{108}$$

where we use the identity $\Delta_{\mathcal{M}}\rho \cdot h = h\Delta_{\mathcal{M}}(\rho) + \rho\Delta_{\mathcal{M}}(h) + 2\langle\nabla_{\mathcal{M}}\rho, \nabla_{\mathcal{M}}h\rangle$ for smooth functions $\rho, h : \mathcal{M} \to \mathbb{R}$, as both $f$ and the restriction of the manifold onto a specific dimension are smooth with respect to them manifold $\mathcal{M}$. Furthermore, we use the fact that by combining it with the fact that the manifold is compact, we get that the Laplace-Beltrami operator and the gradients over $x_d, g(\boldsymbol{x}), g(\boldsymbol{x}) \cdot x_d$ are bounded.

Now, by combining it with (101) we get that for sufficiently small $\epsilon$

$$\frac{1}{C}\int_{\boldsymbol{x}\in\mathcal{M}}\exp\left(-\frac{\|\boldsymbol{x}-\boldsymbol{y}\|^2}{2\epsilon}\right)g(\boldsymbol{x})(x_d-y_d)^2d\boldsymbol{x} = \epsilon/2(E(\boldsymbol{y})\cdot 0 + 2g(\boldsymbol{y})\|\nabla_{\mathcal{M}}y_d\|^2) + O(\epsilon^2) \tag{109}$$

$$= \epsilon g(\boldsymbol{y})\|\nabla_{\mathcal{M}}y_d\|^2 + O(\epsilon^2), \tag{110}$$

based on the characteristics of $E$. $\qquad\square$

**Proposition 3.2** Let $\mathcal{M} \subset \mathbb{R}^D$ be a smooth, compact, Riemannian manifold with intrinsic dimension $dim(\mathcal{M}) < D$. Suppose $\boldsymbol{y}_1, \ldots, \boldsymbol{y}_N \in \mathcal{M}$ are sampled independently from a smooth non-vanishing density $f$ over $\mathcal{M}$, and let $\boldsymbol{W}$ be defined as in (1). Then, for all $i \in \{1, \ldots, N\}$ and sufficiently small $\epsilon_1, \ldots, \epsilon_N \in \mathbb{R}_+$, we have

$$\sum_{j=1}^{N}W_{i,j}\|\boldsymbol{y}_i - \boldsymbol{y}_j\|^2 \xrightarrow[N\to\infty]{a.s.} \frac{\epsilon_i}{2}\cdot dim(\mathcal{M}) + O(\epsilon_i^2). \tag{6}$$

*Proof of Proposition 3.2.* For simplicity denote $\boldsymbol{x} = \boldsymbol{y}_i$. Let's examine the inquired term in (6)

$$\sum_{j=1}^{N}W_{i,j}\|\boldsymbol{x} - \boldsymbol{y}_j\|^2 \tag{111}$$

$$= \sum_{j\in\{1,\ldots,N\}/\{i\}}\frac{\exp(-\|\boldsymbol{x}-\boldsymbol{y}_j\|^2/\epsilon_i)}{\sum_{t\in\{1,\ldots,N\}/\{i\}}\exp(-\|\boldsymbol{x}-\boldsymbol{y}_t\|^2/\epsilon_i)}\cdot\|\boldsymbol{x}-\boldsymbol{y}_j\|^2 \tag{112}$$

$$\xrightarrow[N\to\infty]{a.s.} \int_{\boldsymbol{y}\in\mathcal{M}}\frac{\exp(-\|\boldsymbol{x}-\boldsymbol{y}\|^2/\epsilon_i)\|\boldsymbol{x}-\boldsymbol{y}\|^2 f(\boldsymbol{y})d\boldsymbol{y}}{\int_{\boldsymbol{z}\in\mathcal{M}}\exp(-\|\boldsymbol{x}-\boldsymbol{z}\|^2/\epsilon_i)f(\boldsymbol{z})d\boldsymbol{z}}, \tag{113}$$

where the derivation made in the last line is based on Lemma 2 in (Hein et al., 2005). Specifically by considering $A_{0,\epsilon_i,n-1}$ and the samples $\{\boldsymbol{y}_j\}_{j\in I_i}$ where $I_i = \{1, \ldots, N\}/\{i\}$. To be explicit, in the context of the derivation, this Lemma states that: Let $\boldsymbol{x} \in \mathcal{M}$ and $g$ be a continuous function on $\mathcal{M}$. Then, there exists a constant $C \geq 1$ so that for any $\epsilon, \delta \in (0, 1/C)$ such that

$$Pr\left(\left|\frac{\sum_{j\in I}\exp(-\|\boldsymbol{x}-\boldsymbol{y}_j\|^2/\epsilon)g(\boldsymbol{y}_j)}{\sum_{j\in I}\exp(-\|\boldsymbol{x}-\boldsymbol{y}_j\|^2/\epsilon)} - \frac{E_{\boldsymbol{y}\in\mathcal{M}}[\exp(-\|\boldsymbol{x}-\boldsymbol{y}\|^2/\epsilon)g(\boldsymbol{y})]}{E_{\boldsymbol{z}\in\mathcal{M}}[\exp(-\|\boldsymbol{x}-\boldsymbol{z}\|^2/\epsilon)]}\right| \geq \delta\right) \tag{114}$$

$$\leq CN\cdot\exp(-N\epsilon^{dim(\mathcal{M})}\delta^2/C). \tag{115}$$

Then, we obtain the almost-sure convergence via the Borel-Cantelli lemma as the sum below is finite for any $\delta \in (0, 1)$, and therefore applies for $\delta \geq 1$ as well .

$$\sum_{N=10}^{\infty}Pr\left(\exists i\in\{1,\ldots,N\}:\left|\frac{\sum_{j\in I_i}\exp(-\|\boldsymbol{y}_i-\boldsymbol{y}_j\|^2/\epsilon)g(\boldsymbol{y}_j)}{\sum_{j\in I_i}\exp(-\|\boldsymbol{y}_i-\boldsymbol{y}_j\|^2/\epsilon)} - \frac{E_{\boldsymbol{y}\in\mathcal{M}}[\exp(-\|\boldsymbol{y}_i-\boldsymbol{y}\|^2/\epsilon)g(\boldsymbol{y})]}{E_{\boldsymbol{z}\in\mathcal{M}}[\exp(-\|\boldsymbol{y}_i-\boldsymbol{z}\|^2/\epsilon)]}\right| \geq \delta\right) \tag{116}$$

$$\leq \sum_{N=10}^{\infty}\sum_{i=1}^{N}Pr\left(\left|\frac{\sum_{j\in I}\exp(-\|\boldsymbol{x}-\boldsymbol{y}_j\|^2/\epsilon)g(\boldsymbol{y}_j)}{\sum_{j\in I}\exp(-\|\boldsymbol{x}-\boldsymbol{y}_j\|^2/\epsilon)} - \frac{E_{\boldsymbol{y}\in\mathcal{M}}[\exp(-\|\boldsymbol{x}-\boldsymbol{y}\|^2/\epsilon)g(\boldsymbol{y})]}{E_{\boldsymbol{z}\in\mathcal{M}}[\exp(-\|\boldsymbol{x}-\boldsymbol{z}\|^2/\epsilon)]}\right| \geq \delta\right) \tag{117}$$

$$\leq \sum_{N=10}^{\infty}CN^2\cdot\exp(-N\epsilon^{dim(\mathcal{M})}\delta^2/C) \tag{118}$$

$$< \infty, \tag{119}$$

where we used Boole's inequality, and the sum's finiteness follows from its terms' exponential decay.

Next, we decompose the Euclidean distance and apply Lemma I.2 to approximate the integral presented in (113):

$$\int_{\boldsymbol{y} \in \mathcal{M}} \frac{\exp(-\|\boldsymbol{x} - \boldsymbol{y}\|^2/\epsilon_i)\|\boldsymbol{x} - \boldsymbol{y}\|^2 f(\boldsymbol{y})d\boldsymbol{y}}{\int_{\boldsymbol{z} \in \mathcal{M}} \exp(-\|\boldsymbol{x} - \boldsymbol{z}\|^2/\epsilon_i)f(\boldsymbol{z})dz} = \sum_{d=1}^{D} \int_{\boldsymbol{y} \in \mathcal{M}} \frac{\exp(-\|\boldsymbol{x} - \boldsymbol{y}\|^2/\epsilon_i)(x_d - y_d)^2 f(\boldsymbol{y})d\boldsymbol{y}}{\int_{\boldsymbol{z} \in \mathcal{M}} \exp(-\|\boldsymbol{x} - \boldsymbol{z}\|^2/\epsilon_i)f(\boldsymbol{z})dz} \tag{120}$$

$$= \sum_{d=1}^{D} \frac{(\epsilon_i/2)f(\boldsymbol{x})\|\nabla_{\mathcal{M}} x_d\|^2(1 + O(\epsilon_i))}{f(\boldsymbol{x})(1 + O(\epsilon_i))} \tag{121}$$

$$= \sum_{d=1}^{D}(\epsilon_i/2)\|\nabla_{\mathcal{M}} x_d\|^2(1 + O(\epsilon_i)) \cdot \frac{1}{1 + O(\epsilon_i)} \tag{122}$$

$$= \sum_{d=1}^{D}(\epsilon_i/2)\|\nabla_{\mathcal{M}} x_d\|^2(1 + O(\epsilon_i)) \cdot (1 + O(\epsilon_i)) \tag{123}$$

$$= \left(\frac{\epsilon_i}{2} \sum_{d=1}^{D} \|\nabla_{\mathcal{M}} x_d\|^2\right) + O(\epsilon_i^2), \tag{124}$$

where we use the first-order approximation of $1/(1 + a) = 1 - a + O(a^2) = 1 + O(a)$ for sufficiently small $a > 0$. Finally, by using proposition 3.1 from (Osher et al., 2017) we get

$$= \frac{\epsilon_i}{2} \cdot \dim(\mathcal{M}) + O\left(\epsilon^2\right), \tag{125}$$

and thus, the proof is complete. $\qquad\square$

**Lemma I.3.** *Let $\alpha \in (1, N - 1)$ and let $\{\boldsymbol{\omega}^{(k)}\}_{k=1}^{K}, \subset [0,1]^D$ be a partition solution as defined in Problem 3.3. Define a set of $K$ affinity matrices $\{\boldsymbol{W}^{(k)}\} \subset [0,1]^{N \times N}$ by*

$$W_{i,j}^{(k)} = \begin{cases} \dfrac{\exp\left(-\dfrac{\|\boldsymbol{y}_i - \boldsymbol{y}_j\|_{\boldsymbol{\omega}^{(k)}}^2}{\epsilon_{k,i}}\right)}{\sum_{t=1}^{N} \exp\left(-\dfrac{\|\boldsymbol{y}_i - \boldsymbol{y}_t\|_{\boldsymbol{\omega}^{(k)}}^2}{\epsilon_{k,i}}\right)} & \text{for } j \neq i \\ 0 & \text{else} \end{cases} \tag{126}$$

*for all $i, j = 1, \ldots, N$, and $K = 1, \ldots, K$ where $\epsilon_{k,i}$ attains the minimum value that satisfies*

$$\sum_{j=1}^{N} W_{i,j}^{(k)} \log W_{i,j}^{(k)} \leq -\log(\alpha). \tag{127}$$

*Then, a minimizer of the function $G$ (defined in (7)) over matrices that satisfy the constraints in Problem 3.3 is*

$$\{\boldsymbol{W}^{(k)}\} = \underset{\{\tilde{\boldsymbol{W}}^{(k)}\}}{\arg\min} G(\{\tilde{\boldsymbol{W}}^{(k)}\}_{k=1}^{K}, \{\boldsymbol{\omega}^{(k)}\}_{k=1}^{K}, \{\boldsymbol{y}_1, \ldots, \boldsymbol{y}_N\}). \tag{128}$$

*Proof.* The proof will demonstrate that the optimal solution can be derived from solving $NK$ subproblems, each corresponding to a row of each graph solution. We will then use convexity conditions to derive the optimal solution under certain assumptions on the data. Finally, we will show that if these assumptions do not hold, the solution takes a similar form, but with bandwidth parameters approaching zero.

We begin by showing that the problem can be solved using $NK$ subproblems separately. For each $i \in \{1, \ldots, N\}$ and $k \in \{1, \ldots, K\}$ we define a subproblem related to the $i - th$ row of the $k$-th graph matrix by

$$\underset{\tilde{W}_{i,1}^{(k)}, \ldots, \tilde{W}_{i,N}^{(k)} \in [0,1]}{\arg\min} G_{k,i}(\{\tilde{W}_{i,j}\}_{j=1}^{N}, \boldsymbol{\omega}^{(k)}, \{\boldsymbol{y}_j\}_{j=1}^{N}) \tag{129}$$

subject to the constraints $\tilde{W}_{i,i}^{(k)} = 0$, $\sum_{j=1}^{N} \tilde{W}_{i,j}^{(k)} = 1$ and $\sum_{j=1}^{N} \tilde{W}_{i,j}^{(k)} \log \tilde{W}_{i,j}^{(k)} \leq -\log(\alpha)$, where

$$G_{k,i}(\{\tilde{W}_{i,j}^{(k)}\}_{j=1}^{N}, \boldsymbol{\omega}^{(k)}, \{\boldsymbol{y}_j\}_{j=1}^{N}) = \sum_{j=1}^{N} \tilde{W}_{i,j}^{(k)} \|\boldsymbol{y}_i - \boldsymbol{y}_j\|^2. \tag{130}$$

Based on the next derivation, we can see that the optimal solution should be optimal for each of these sub-problems

$$\min_{\{\tilde{\boldsymbol{W}}^{(k)}\} \subset [0,1]^{N \times N}} G(\{\tilde{\boldsymbol{W}}^{(k)}\}_{k=1}^{K}, \{\boldsymbol{\omega}^{(k)}\}_{k=1}^{K}, \{\boldsymbol{y}_j\}_{j=1}^{N}) \tag{131}$$

$$= \min_{\{\tilde{\boldsymbol{W}}^{(k)}\} \subset [0,1]^{N \times N}} \sum_{k=1}^{K} \sum_{i,j=1}^{N} \tilde{W}_{i,j}^{(k)} \|\boldsymbol{y}_i - \boldsymbol{y}_j\|^2 \tag{132}$$

$$= \sum_{k=1}^{K} \sum_{i=1}^{N} \min_{\tilde{\boldsymbol{W}}_{i,:}^{(k)} \in [0,1]^{N}} \sum_{j=1}^{N} \tilde{W}_{i,j}^{(k)} \|\boldsymbol{y}_i - \boldsymbol{y}_j\|^2 \tag{133}$$

$$= \sum_{k=1}^{K} \sum_{i=1}^{N} \min_{\tilde{\boldsymbol{W}}_{i,:}^{(k)} \in [0,1]^{N}} G_{k,i}(\{\tilde{W}_{i,j}^{(k)}\}_{j=1}^{N}, \boldsymbol{\omega}^{(k)}, \{\boldsymbol{y}_j\}_{j=1}^{N}). \tag{134}$$

where $\boldsymbol{W}_{i,:}^{(k)}$ denotes the $i$-th row of $\boldsymbol{W}^{(k)}$, and the domain of each minimization problem above contains the constraints shown in Problem 3.3.

Without loss of generality, we are going to find the optimal $W_{i,1}^{(k)}, \ldots, W_{i,N}^{(k)}$ that minimize the $G_{k,i}$ for some $i \in \{1, \ldots, N\}$ and $k \in \{1, \ldots, K\}$. Specifically, we assume $W_{i,i}^{(k)} = 0$ based on the equality to zero constraint, and omit this constraint. To find the optimal solution, we begin by building on the Karush-Kuhn-Tucker (KKT) Theorem (Corollary 28.3.1 in (Rockafellar, 1970)). We note that this solution will depend on certain conditions; therefore, after this derivation, we will provide an alternative solution that attains a similar form when these conditions are not met.

The theorem assumes that the objective and the inequality constraint is convex and that the equality constraint is an affine function. Furthermore, it assumes that there exists a solution within the domain that satisfies the inequality constraint in a strict manner (Slater's conditions). We begin with the latter assumption, by defining a valid solution $\tilde{W}_{i,j}^{(k)} = 1/(N-1)$ for any $j \neq i$ and $\tilde{W}_{i,i}^{(k)} = 0$, as its rows sums up to ones. The entropy constraint is strictly satisfied $\sum_{j=1}^{N} \tilde{W}_{i,j}^{(k)} \log \tilde{W}_{i,j}^{(k)} = -\log(N-1) < -\log(\alpha)$, and therefore Slater's conditions are satisfied.

The objective $G_{k,i}$ is an affine function and therefore is convex, and the equality constraint $\sum_{j=1}^{N} \tilde{W}_{i,j}^{(k)} = 1$ is indeed affine. Finally, we will show that the constraint of the entropy inequality is convex by showing that its Hessian is a positive semi-definite matrix (Theorem 4.5 in (Rockafellar, 1970)). Specifically, the Hessian elements are

$$\frac{d^2}{d^2 \tilde{W}_{i,j}^{(k)}} \sum_{t=1}^{N} \tilde{W}_{i,t}^{(k)} \log(\tilde{W}_{i,t}^{(k)}) = \frac{d}{d\tilde{W}_{i,j}^{(k)}} \left(1 + \log(\tilde{W}_{i,j}^{(k)})\right) \tag{135}$$

$$= \frac{1}{\tilde{W}_{i,j}^{(k)}} \tag{136}$$

$$\geq 1. \tag{137}$$

$$\frac{d}{d\tilde{W}_{i,r}^{(k)}} \frac{d}{d\tilde{W}_{i,j}^{(k)}} \sum_{t=1}^{N} \tilde{W}_{i,t}^{(k)} \log(\tilde{W}_{i,t}^{(k)}) = \frac{d}{d\tilde{W}_{i,r}^{(k)}} \left(1 + \log(\tilde{W}_{i,j}^{(k)})\right) \tag{138}$$

$$= 0 \tag{139}$$

for all $j, r \in \{1, \ldots, N\}/\{i\}$ where $j \neq r$. As the Hessian is a diagonal matrix with positive values we can conclude that it is a positive definite matrix.

The theorem suggests that a solution that satisfies the KKT conditions, including stationary, primal feasibility, dual feasibility, and complementary slackness, is an optimal solution for the problem. To satisfy its conditions we need to first define the

Lagrangian of the minimization problem by

$$L_{k,i}(\{\tilde{W}_{i,j}^{(k)}\}_{j=1}^N, \boldsymbol{\omega}^{(k)}, \{\boldsymbol{y}_j\}_{j=1}^N) \equiv \sum_{j=1}^N \tilde{W}_{i,j}^{(k)} \|\boldsymbol{y}_i - \boldsymbol{y}_j\|_{\boldsymbol{\omega}^{(k)}}^2 \tag{140}$$

$$+ \epsilon_{k,i} \left( \log(\alpha) + \sum_{j=1}^N \tilde{W}_{i,j}^{(k)} \log(\tilde{W}_{i,j}^{(k)}) \right)$$

$$+ \mu_{k,i} \left( \sum_{j=1}^N \tilde{W}_{i,j}^{(k)} - 1 \right)$$

where $\epsilon_{k,i} \geq 0$ and $\mu_{k,i} \in \mathbb{R}$.

A solution that satisfies the stationary condition should attain for all $j \neq i$

$$0 = \frac{dL_{k,i}(\{\tilde{W}_{i,j}^{(k)}\}_{j=1}^N, \boldsymbol{\omega}^{(k)}, \{\boldsymbol{y}_j\}_{j=1}^N)}{dW_{i,j}^{(k)}} \tag{141}$$

$$= \|\boldsymbol{y}_i - \boldsymbol{y}_j\|_{\boldsymbol{\omega}^{(k)}}^2 + \epsilon_{k,i} \left( 1 + \log(W_{i,j}^{(k)}) \right) + \mu_{k,i} \tag{142}$$

$$W_{i,j}^{(k)} = \exp \left( -\frac{\|\boldsymbol{y}_i - \boldsymbol{y}_j\|_{\boldsymbol{\omega}^{(k)}}^2 + \epsilon_{k,i} + \mu_{k,i}}{\epsilon_{k,i}} \right). \tag{143}$$

The primal feasibility condition on the on the equality constraint induces-

$$1 = \sum_{j=1}^N W_{i,j}^{(k)} \tag{144}$$

$$= \sum_{j\neq i} \exp \left( -\frac{\|\boldsymbol{y}_i - \boldsymbol{y}_j\|_{\boldsymbol{\omega}^{(k)}}^2 + \epsilon_{k,i} + \mu_{k,i}}{\epsilon_{k,i}} \right) \tag{145}$$

$$\exp \left( \frac{\mu_{k,i}}{\epsilon_{k,i}} \right) = \sum_{j\neq i} \exp \left( -\frac{\|\boldsymbol{y}_i - \boldsymbol{y}_j\|_{\boldsymbol{\omega}^{(k)}}^2 + \epsilon_{k,i}}{\epsilon_{k,i}} \right). \tag{146}$$

By pushing it back into (143) we get that for all $j \neq i$

$$W_{i,j}^{(k)} = \frac{\exp \left( -\frac{\|\boldsymbol{y}_i - \boldsymbol{y}_j\|_{\boldsymbol{\omega}^{(k)}}^2 + \epsilon_{k,i}}{\epsilon_{k,i}} \right)}{\sum_{t\neq i} \exp \left( -\frac{\|\boldsymbol{y}_i - \boldsymbol{y}_t\|_{\boldsymbol{\omega}^{(k)}}^2 + \epsilon_{k,i}}{\epsilon_{k,i}} \right)} \tag{147}$$

$$= \frac{\exp \left( -\frac{\|\boldsymbol{y}_i - \boldsymbol{y}_j\|_{\boldsymbol{\omega}^{(k)}}^2}{\epsilon_{k,i}} \right)}{\sum_{t\neq i} \exp \left( -\frac{\|\boldsymbol{y}_i - \boldsymbol{y}_t\|_{\boldsymbol{\omega}^{(k)}}^2}{\epsilon_{k,i}} \right)}. \tag{148}$$

Furthermore, to satisfy the primal feasibility and complementary slackness of the entropy constraint we define any $\epsilon_{k,i}$ by

$$\epsilon_{k,i} \quad s.t. \quad \sum_{j=1}^N W_{i,j}^{(k)} \log W_{i,j}^{(k)} = -\log(\alpha). \tag{149}$$

By incorporating the results from Lemma I.1 with $\Delta_j \equiv \|\boldsymbol{y}_i - \boldsymbol{y}_j\|_{\boldsymbol{\omega}^{(k)}}^2$ for any $j \in \{1, \ldots, N-1\}$, we get that the function $\sum_j W_{i,j}^{(k)} \log W_{i,j}^{(k)}$ is a non-increasing function in $\epsilon_{k,i}$. Additionally, according to the lemma, this function is bounded by

$[-\log(N-1), -\log(\tilde{\alpha}_{k,i})]$, where $\tilde{\alpha}_{k,i} \equiv |\{j \in \{1,\ldots,N\}/\{i\} : \|\boldsymbol{y}_j - \boldsymbol{y}_i\|_{\boldsymbol{\omega}^{(k)}} = \min_{t\neq i} \|\boldsymbol{y}_t - \boldsymbol{y}_i\|_{\boldsymbol{\omega}^{(k)}}\}|$. These boundary values correspond to the limit of the function as $\epsilon_i \to \infty$ and $\epsilon_i \to 0$, respectively. Therefore a solution $\epsilon_i$ exists only if $\tilde{\alpha}_i \leq \alpha$.

Therefore, based on the Karush-Kuhn-Tucker Theorem if $\tilde{\alpha}_i \leq \alpha$ then the optimal assignment of $W_{i,1}\ldots,W_{i,N}$ is as shown in (94), with a $\epsilon_i$ that satisfies (95). To address the optimal assignment when $\tilde{\alpha}_i > \alpha$ we use a direct approach that does not build on the KKT conditions. In this regime, any assignment of $\epsilon_i$ should satisfy the inequality entropy constraint, as the entropy function is bounded from above by $-\log \tilde{\alpha}_i$ (Lemma I.1). Building on the suggested solution of $W_{i,1},\ldots,W_{i,N}$, the assignment of $\epsilon_i$ that tends to 0 will result in the following solution

$$W_{i,j}^{(k)} = \begin{cases} 0 & \text{if } j = i \\ \frac{1}{\tilde{\alpha}_i} & \text{if } j \neq i \quad \text{and} \quad \|\boldsymbol{y}_j - \boldsymbol{y}_i\|_{\boldsymbol{\omega}^{(k)}} = \min_{t\neq i} \|\boldsymbol{y}_t - \boldsymbol{y}_i\|_{\boldsymbol{\omega}^{(k)}} \\ 0 & else \end{cases} \tag{150}$$

based on Lemma I.1. This solution is valid as it satisfies the entropy constraint as $\sum_j W_{i,j}^{(k)} \log W_{i,j}^{(k)} = -\log(\tilde{\alpha}_{k,i}) \leq -\log(\alpha)$, it sums up to one, and $W_{i,i}^{(k)} = 0$. Now, we show that it achieves the minimal values among all solutions

$$\sum_{j=1}^{N} W_{i,j}^{(k)} \|\boldsymbol{y}_i - \boldsymbol{y}_j\|_{\boldsymbol{\omega}^{(k)}}^2 \tag{151}$$

$$= \min_{t\neq i} \|\boldsymbol{y}_t - \boldsymbol{y}_i\|_{\boldsymbol{\omega}^{(k)}}^2 \tag{152}$$

$$\leq \min_{\tilde{W}_{i,1}^{(k)},\ldots,\tilde{W}_{i,N}^{(k)} \in [0,1]:\sum_j \tilde{W}_{i,j}^{(k)}=1, \tilde{W}_{i,i}^{(k)}=0} \sum_{j=1}^{N} \tilde{W}_{i,j}^{(k)} \|\boldsymbol{y}_i - \boldsymbol{y}_j\|_{\boldsymbol{\omega}^{(k)}}^2 \tag{153}$$

$$\leq \min_{\tilde{W}_{i,1}^{(k)},\ldots,\tilde{W}_{i,N}^{(k)} \in [0,1]:\sum_j \tilde{W}_{i,j}^{(k)}=1, \tilde{W}_{i,i}^{(k)}=0, \sum_j \tilde{W}_{i,j}^{(k)} \log \tilde{W}_{i,j}^{(k)} \leq -\log(\alpha)} \sum_{j=1}^{N} \tilde{W}_{i,j}^{(k)} \|\boldsymbol{y}_i - \boldsymbol{y}_j\|_{\boldsymbol{\omega}^{(k)}}^2. \tag{154}$$

Hence, we can conclude the proof. $\qquad\square$

**Lemma I.4.** *Let* $\{\boldsymbol{W}^{(k)}\} \in [0,1]^{N\times N}$ *be affinity matrices under the constraints in Problem 3.3. Define a partitioning solution* $\{\boldsymbol{\omega}^{(k)}\} \subset \{0,1\}^D$ *by*

$$\omega_d^{(k)} = \begin{cases} 1 & \text{if } k = \tilde{k} \text{ for some } \tilde{k} \in \Omega(d) \\ 0 & else \end{cases}, \tag{155}$$

*for* $d = 1,\ldots,D$, *where* $\Omega(d) = \arg\min_{k\in\{1,\ldots,K\}} S(\boldsymbol{W}^{(k)}, d)$ *and* $S$ *is defined in* (3).

*Then, a minimizer of the objective function in Problem 3.3 while fixing the graph parameters satisfy its constraints is*

$$\{\boldsymbol{\omega}^{(k)}\} = \arg\min_{\{\tilde{\boldsymbol{\omega}}^{(k)}\}} G(\{\boldsymbol{W}^{(k)}\}_{k=1}^K, \{\tilde{\boldsymbol{\omega}}^{(k)}\}_{k=1}^K, \{\boldsymbol{y}_1,\ldots,\boldsymbol{y}_N\}). \tag{156}$$

*Proof.* The minimization problem is

$$\arg\min_{\{\tilde{\boldsymbol{\omega}}^{(k)}\}} \sum_{d=1}^D \sum_{k=1}^K \tilde{\omega}_d^{(k)} \left( \sum_{i=1}^N \sum_{j=1}^N W_{i,j}^{(k)} ((\boldsymbol{y}_i)_d - (\boldsymbol{y}_j)_d)^2 \right). \tag{157}$$

As the constraints are $\sum_k \omega_d^{(k)} = 1$ for any $d = 1,\ldots,D$, the problem can be decomposed into $D$ independent problems. Meaning that for any $d \in \{1,\ldots,D\}$ we need to solve

$$\arg\min_{\{\tilde{\omega}_d^{(k)}\}} \sum_{k=1}^K \tilde{\omega}_d^{(k)} \left( \sum_{i=1}^N \sum_{j=1}^N W_{i,j}^{(k)} ((\boldsymbol{y}_i)_d - (\boldsymbol{y}_j)_d)^2 \right). \tag{158}$$

Hence, the solution to this problem is

$$\omega_d^{(k)} = \begin{cases} 1 & \text{if } k = \tilde{k} \text{ for some } \tilde{k} \in I(d) \\ 0 & else \end{cases} \tag{159}$$

for all $k \in \{1, \ldots, K\}$ and $d \in \{1, \ldots, D\}$, where

$$\Omega(d) = \underset{k \in \{1,\ldots,K\}}{\arg\min} S(\boldsymbol{W}^{(k)}, d). \tag{160}$$

$\square$

**Proposition 3.4** There exists an optimal partitioning solution $\{\boldsymbol{\omega}^{(k)}\}_{k=1}^{K}$ and corresponding affinity matrices $\{\boldsymbol{W}^{(k)}\}_{k=1}^{K}$ that solve Problem 3.3 and are of the form

$$W_{i,j}^{(k)} = \begin{cases} \exp\left(-\|\boldsymbol{y}_i - \boldsymbol{y}_j\|_{\boldsymbol{\omega}^{(k)}}^2 / \epsilon_{k,i}\right) / D_{i,i}^{(k)} & \text{if } i \neq j \\ 0 & else \end{cases} \text{,} \tag{9}$$

$$\omega_d^{(k)} = \begin{cases} 1 & \text{if } k = \tilde{k} \text{ for some } \tilde{k} \in \Omega(d) \\ 0 & else \end{cases} \text{,} \tag{10}$$

$$\Omega(d) = \underset{k \in \{1,\ldots,K\}}{\arg\min} \sum_{i,j=1}^{N} S(\boldsymbol{W}^{(k)}, d), \tag{11}$$

for $d = 1, \ldots, D$ and $i, j = 1, \ldots, N$, where the bandwidth parameters $\{\epsilon_{k,i}\} \subset \mathbb{R}_+$ are the minimum values that satisfy the entropy constraints in Problem 3.3, and $S$ is the Laplacian-type score defined in (3).

*Proof of Proposition 3.4.* The proposition aims to characterize optimal parameters of Problem 3.3. We begin by defining two sub-problems that are related to it, each focusing on minimizing one set of parameters while keeping the other set fixed:

$$\{\boldsymbol{W}^{(k)}\} = \underset{\{\tilde{\boldsymbol{W}}^{(k)}\}_k}{\arg\min} G(\{\tilde{\boldsymbol{W}}^{(k)}\}_{k=1}^{K}, \{\tilde{\boldsymbol{\omega}}^{(k)}\}_{k=1}^{K}, \{\boldsymbol{y}_1, \ldots, \boldsymbol{y}_N\}) \tag{161}$$

$$\{\boldsymbol{\omega}^{(k)}\} = \underset{\{\tilde{\boldsymbol{\omega}}^{(k)}\}_k}{\arg\min} G(\{\tilde{\boldsymbol{W}}^{(k)}\}_{k=1}^{K}, \{\tilde{\boldsymbol{\omega}}^{(k)}\}_{k=1}^{K}, \{\boldsymbol{y}_1, \ldots, \boldsymbol{y}_N\}). \tag{162}$$

where the parameters are limited to the constraints stated in Problem 3.3.

These two sub-problems are considered in Lemmas I.3 and I.4. Specifically, in Lemma I.3 the optimal graph matrices are derived in the form of

$$W_{i,j}^{(k)} = \begin{cases} \dfrac{\exp\left(-\dfrac{\|\boldsymbol{y}_i - \boldsymbol{y}_j\|_{\tilde{\boldsymbol{\omega}}^{(k)}}^2}{\epsilon_{k,i}}\right)}{\sum_{t=1}^{N} \exp\left(-\dfrac{\|\boldsymbol{y}_i - \boldsymbol{y}_t\|_{\tilde{\boldsymbol{\omega}}^{(k)}}^2}{\epsilon_{k,i}}\right)} & \text{for } j \neq i \\ 0 & else \end{cases} \text{,} \tag{163}$$

$$\tag{164}$$

for $i, j = 1, \ldots, N$ and $k = 1, \ldots, K$, with $\epsilon_{k,i}$ that attains the minimal value that satisfies

$$\sum_{j=1}^{N} W_{i,j}^{(k)} \log W_{i,j}^{(k)} \leq -\log(\alpha). \tag{165}$$

On the other hand, in Lemma I.4 an optimal partitioning parameters are derived in the form of

$$\omega_d^{(k)} = \begin{cases} 1 & \text{if } k = \tilde{k} \text{ for some } \tilde{k} \in \Omega(d) \\ 0 & else \end{cases} \tag{166}$$

for $d = 1, \ldots, D$ and $k = 1, \ldots, K$, where $\Omega(d) = \arg\min_{k \in \{1,\ldots,K\}} S(\tilde{\boldsymbol{W}}^{(k)}, d)$ and $S$ is defined in (3).

Therefore there exists parameters of this form that minimizes Problem 3.3. $\square$

## I.2. Proofs of Section 4

**Corollary 4.2** Let $\{\tilde{\boldsymbol{\omega}}^{(k)}\}$ be a partitioning that satisfies the constraints in Problem 4.1. Then, graph affinity matrices that minimize (13) while fixing the partitioning parameters $\{\tilde{\boldsymbol{\omega}}^{(k)}\}$ are

$$W_{i,j}^{(k)} = \begin{cases} \exp\left(-\frac{\|\boldsymbol{y}_i - \boldsymbol{y}_j\|_{\tilde{\boldsymbol{\omega}}^{(k)}}^2}{\epsilon \cdot (1/D) \sum_{d=1}^{D} \tilde{\omega}_d^{(k)}}\right) / \sum_{t=1}^{N} \exp\left(-\frac{\|\boldsymbol{y}_i - \boldsymbol{y}_t\|_{\tilde{\boldsymbol{\omega}}^{(k)}}^2}{\epsilon \cdot (1/D) \sum_{d=1}^{D} \tilde{\omega}_d^{(k)}}\right) & \text{if } i \neq j \\ 0 & \text{else} \end{cases} \tag{14}$$

for $i, j = 1, \ldots, N$ and $k = 1, \ldots, K$.

*The Proof of Corollary 4.2.* The proof will demonstrate that the optimal solution can be derived from solving $NK$ subproblems, each corresponding to a row of each graph solution. We will then use convexity conditions to derive the optimal solution.

For simplicity we denote the following term distance term

$$\gamma(\boldsymbol{y}_i, \boldsymbol{y}_j; \boldsymbol{\omega}^{(k)}) \equiv \frac{\|\boldsymbol{y}_i - \boldsymbol{y}_j\|_{\boldsymbol{\omega}^{(k)}}^2}{(1/D) \sum_{d=1}^{D} \omega_d^{(k)}}. \tag{167}$$

This will be used throughout the proof.

We begin by showing that the problem can be solved using $NK$ subproblems separately. For each $i \in \{1, \ldots, N\}$ and $k \in \{1, \ldots, K\}$ we define a subproblem related to the $i - th$ row of the $k$-th graph matrix by

$$\underset{\tilde{W}_{i,1}^{(k)}, \ldots, \tilde{W}_{i,N}^{(k)} \in [0,1]}{\arg\min} \tilde{G}_{k,i}(\tilde{W}_{i,1}, \ldots, \tilde{W}_{i,N}, \{\boldsymbol{y}_1, \ldots, \boldsymbol{y}_N\}) \tag{168}$$

subject to the constraints $\tilde{W}_{i,i}^{(k)} = 0$ and $\sum_{j=1}^{N} \tilde{W}_{i,j}^{(k)} = 1$, where

$$\tilde{G}_{k,i}(\tilde{W}_{i,1}^{(k)}, \ldots, \tilde{W}_{i,N}^{(k)}), \{\boldsymbol{y}_1, \ldots, \boldsymbol{y}_N\}) = \sum_{j=1}^{N} \tilde{W}_{i,j}^{(k)} \gamma(\boldsymbol{y}_i, \boldsymbol{y}_j; \boldsymbol{\omega}^{(k)}) + \epsilon \sum_{j=1}^{N} \tilde{W}_{i,j}^{(k)} \log\left(\tilde{W}_{i,j}^{(k)}\right). \tag{169}$$

Based on the next derivation, we can see that the optimal solution should be optimal for each of these sub-problems

$$\underset{\{\tilde{\boldsymbol{W}}^{(k)}\}_{k=1}^{K} \subset [0,1]^{N \times N}}{\min} \tilde{G}(\tilde{W}_{i,1}, \ldots, \tilde{W}_{i,N}, \{\boldsymbol{y}_1, \ldots, \boldsymbol{y}_N\}) + \epsilon \sum_{k=1}^{K} \sum_{i,j=1}^{N} \tilde{W}_{i,j}^{(k)} \log\left(\tilde{W}_{i,j}^{(k)}\right) \tag{170}$$

$$\tag{171}$$

$$= \underset{\{\tilde{\boldsymbol{W}}^{(k)}\}_{k=1}^{K} \subset [0,1]^{N \times N}}{\min} \sum_{k=1}^{K} \sum_{i,j=1}^{N} \tilde{W}_{i,j}^{(k)} \gamma(\boldsymbol{y}_i, \boldsymbol{y}_j; \boldsymbol{\omega}^{(k)}) + \epsilon \sum_{k=1}^{K} \sum_{i,j=1}^{N} \tilde{W}_{i,j}^{(k)} \log\left(\tilde{W}_{i,j}^{(k)}\right) \tag{172}$$

$$= \sum_{k=1}^{K} \sum_{i=1}^{N} \underset{\tilde{\boldsymbol{W}}_{i,:} \in [0,1]^{N}}{\min} \sum_{j=1}^{N} \tilde{W}_{i,j}^{(k)} \gamma(\boldsymbol{y}_i, \boldsymbol{y}_j; \boldsymbol{\omega}^{(k)}) + \epsilon \sum_{j=1}^{N} \tilde{W}_{i,j}^{(k)} \log\left(\tilde{W}_{i,j}^{(k)}\right) \tag{173}$$

$$= \sum_{k=1}^{K} \sum_{i=1}^{N} \underset{\tilde{\boldsymbol{W}}_{i,:} \in [0,1]^{N}}{\min} \tilde{G}_{k,i}(\tilde{W}_{i,1}, \ldots, \tilde{W}_{i,N}, \{\boldsymbol{y}_1, \ldots, \boldsymbol{y}_N\}), \tag{174}$$

where $\tilde{\boldsymbol{W}}_{i,:}^{(k)}$ denotes the $i$-th row of $\tilde{\boldsymbol{W}}^{(k)}$, and the domain of each minimization problem above includes $W_{i,i}^{(k)} = 0$ and $\sum_{j=1}^{N} W_{i,j}^{(k)} = 1$ for all $k \in \{1, \ldots, K\}$ and $i \in \{1, \ldots, N\}$.

Without loss of generality, we are going to find the optimal $W_{i,1}^{(k)}, \ldots, W_{i,N}^{(k)}$ that minimize the $\tilde{G}_{k,i}$ for some $i \in \{1, \ldots, N\}$ and $k \in \{1, \ldots, K\}$. Specifically, we assume $W_{i,i}^{(k)} = 0$ based on the equality to zero constraint, and omit this constraint. To find the optimal solution, we begin by building on the Karush-Kuhn-Tucker (KKT) Theorem (Corollary 28.3.1 in

(Rockafellar, 1970)). We note that this solution will depend on certain conditions; therefore, after this derivation, we will provide an alternative solution that attains a similar form when these conditions are not met.

The theorem assumes that the objective is convex and that the equality constraint is an affine function. Furthermore, it assumes that there exists a solution within the domain that satisfies the inequality constraints in a strict manner (Slater's conditions). We begin with the latter assumption, as there are no inequality constraints the Slater's conditions are satisfied.

The objective $\tilde{G}_{k,i}$ is a sum of a linear function and an entropy function. Based on Theorem 4.5 in (Rockafellar, 1970), by showing that the Hessian of the entropy function is positive semi-definite we can deduce that it is convex. Specifically, the Hessian elements are

$$\frac{d^2}{d^2 \tilde{W}_{i,j}^{(k)}} \sum_{t=1}^{N} \tilde{W}_{i,t}^{(k)} \log(\tilde{W}_{i,t}^{(k)}) = \frac{d}{d\tilde{W}_{i,j}^{(k)}} \left(1 + \log(\tilde{W}_{i,j}^{(k)})\right) \tag{175}$$

$$= \frac{1}{\tilde{W}_{i,j}^{(k)}} \tag{176}$$

$$\geq 1. \tag{177}$$

$$\frac{d}{d\tilde{W}_{i,r}^{(k)}} \frac{d}{d\tilde{W}_{i,j}^{(k)}} \sum_{t=1}^{N} \tilde{W}_{i,t}^{(k)} \log(\tilde{W}_{i,t}^{(k)}) = \frac{d}{d\tilde{W}_{i,r}^{(k)}} \left(1 + \log(\tilde{W}_{i,j}^{(k)})\right) \tag{178}$$

$$= 0 \tag{179}$$

for all $j, r \in \{1, \ldots, N\}/\{i\}$ where $j \neq r$. As the Hessian is a diagonal matrix with positive values we can conclude that it is a positive definite matrix. Now, as the objective is the sum of two linear functions it is convex as well based on Theorem 5.2 in (Rockafellar, 1970) and the convexity of linear functions.

The KKT theorem states that a solution satisfying the KKT conditions—including stationarity, primal feasibility, dual feasibility, and complementary slackness—is an optimal solution to the problem, provided Slater's condition holds. To derive such a solution we need to first define the Lagrangian of the minimization problem by

$$\tilde{L}_{k,i}(\{\tilde{\boldsymbol{W}}^{(k)}\}_{k=1}^{K}, \{\boldsymbol{\omega}^{(k)}\}_{k=1}^{K}) \equiv \sum_{j=1}^{N} \tilde{W}_{i,j}^{(k)} \gamma(\boldsymbol{y}_i, \boldsymbol{y}_j; \boldsymbol{\omega}^{(k)}) \tag{180}$$

$$+ \ \epsilon \sum_{j=1}^{N} \tilde{W}_{i,j}^{(k)} \log(\tilde{W}_{i,j}^{(k)})$$

$$+ \ \mu_{k,i} \left( \sum_{j=1}^{N} \tilde{W}_{i,j}^{(k)} - 1 \right)$$

where $\epsilon_{k,i} \geq 0$ and $\mu_{k,i} \in \mathbb{R}$.

A solution that satisfies the stationary condition should attain for all $j \neq i$:

$$0 = \frac{d\tilde{L}_{k,i}(\{\boldsymbol{W}^{(k)}\}, \{\boldsymbol{\omega}^{(k)}\})}{dW_{i,j}^{(k)}} \tag{181}$$

$$= \gamma(\boldsymbol{y}_i, \boldsymbol{y}_j; \boldsymbol{\omega}^{(k)}) + \epsilon \left(1 + \log(W_{i,j}^{(k)})\right) + \mu_{k,i} \tag{182}$$

$$W_{i,j}^{(k)} = \exp\left(-\frac{\gamma(\boldsymbol{y}_i, \boldsymbol{y}_j; \boldsymbol{\omega}^{(k)}) + \epsilon + \mu_{k,i}}{\epsilon}\right). \tag{183}$$

The primal feasibility condition on the on the equality constraint induces-

$$1 = \sum_{j=1}^{N} W_{i,j}^{(k)} \tag{184}$$

$$= \sum_{j \neq i} \exp\left(-\frac{\gamma(\boldsymbol{y}_i, \boldsymbol{y}_j; \boldsymbol{\omega}^{(k)}) + \epsilon + \mu_{k,i}}{\epsilon}\right) \tag{185}$$

$$\exp\left(\frac{\mu_{k,i}}{\epsilon}\right) = \sum_{j \neq i} \exp\left(-\frac{\gamma(\boldsymbol{y}_i, \boldsymbol{y}_j; \boldsymbol{\omega}^{(k)}) + \epsilon}{\epsilon}\right). \tag{186}$$

By pushing it back into (183) we get that for all $j \neq i$ and $k \in \{1, \ldots, N\}$

$$W_{i,j}^{(k)} = \frac{\exp\left(-\frac{\gamma(\boldsymbol{y}_i, \boldsymbol{y}_j; \boldsymbol{\omega}^{(k)}) + \epsilon}{\epsilon}\right)}{\sum_{t \neq i} \exp\left(-\frac{\gamma(\boldsymbol{y}_i, \boldsymbol{y}_y; \boldsymbol{\omega}^{(k)}) + \epsilon}{\epsilon}\right)} \tag{187}$$

$$= \frac{\exp\left(-\frac{\gamma(\boldsymbol{y}_i, \boldsymbol{y}_j; \boldsymbol{\omega}^{(k)})}{\epsilon}\right)}{\sum_{t \neq i} \exp\left(-\frac{\gamma(\boldsymbol{y}_i, \boldsymbol{y}_t; \boldsymbol{\omega}^{(k)})}{\epsilon}\right)}. \tag{188}$$

Therefore we can conclude the proof with the derived optimal solution form. $\square$

**Lemma I.5.** *Let $X_1, \ldots, X_N$ be random variables. Then for any $\gamma > 0$*

$$Pr(|\sum_{i=1}^{N} X_i| \geq \gamma) \leq \sum_{i=1}^{N} Pr(|X_i| \geq \gamma/N) \tag{189}$$

*Proof.* This lemma is not new and is shown to simplify other lemmas. We begin with the upper inequality

$$Pr(|\sum_{i=1}^{N} X_i| \geq \gamma) \leq \quad Pr(\sum_{i=1}^{N} |X_i| \geq \gamma) \leq \quad Pr(\exists i : |X_i| \geq \gamma/N) \leq \quad \sum_{i=1}^{N} Pr(|X_i| \geq \gamma/N), \tag{190}$$

where the left inequality follows from the triangle inequality, the middle from set inclusion, and the right from Boole's inequality. $\square$

**Lemma I.6.** *Assume the configuration described in Section 4, and let $\{\boldsymbol{\omega}^{(k)}\} \subset \{0,1\}^D$ be a partitioning solution that satisfies its conditions, and $\epsilon \in (0,1)$. Then,*

$$(1/N) \sum_{s=1}^{K} \sum_{i=1}^{N} \log\left((1/(N-1)) \sum_{j=1; j \neq i}^{N} \exp\left(-\frac{\|\boldsymbol{y}_i - \boldsymbol{y}_j\|_{\boldsymbol{\omega}^{(k)}}^2}{\epsilon(1/D) \sum_{t=1}^{D_s} \sum_{d=1}^{D_s} \omega_d^{(k,t)}}\right)\right) \tag{191}$$

$$-(1/N) \sum_{s=1}^{K} \sum_{i=1}^{N} \log\left((1/(N-1)) \sum_{j=1; j \neq i}^{N} \exp\left(-\sum_{s=1}^{K} \frac{p_k^{(s)}\left(\|\boldsymbol{x}_i^{(s)} - \boldsymbol{x}_j^{(s)}\|^2 + \|\boldsymbol{x}_i^{(K+1)} - \boldsymbol{x}_j^{(K+1)}\|^2\right)}{\epsilon \sum_{t=1}^{K} \beta_t p_t^{(k)}}\right)\right)$$

$$\xrightarrow[D,N \to \infty]{a.s.} 0 \tag{192}$$

*Proof.* The proof will begin by first showing the uniform almost surely convergence of the terms inside the exponentials, then it will derive the above convergence. We denote the following terms that will be used throughout the proof

$$\tilde{x}_i^{(s)} = \begin{pmatrix} \boldsymbol{x}_i^{(s)} \\ \boldsymbol{x}_i^{(K+1)} \end{pmatrix} \qquad for\ i = 1, \ldots, N\ and\ s = 1, \ldots, K \tag{193}$$

$$\beta_{\min} = \min_{s \in \{1, \ldots, K\}} \beta_s \tag{194}$$

$$C = \max(1, \max_{\boldsymbol{z}, \boldsymbol{v} \in \mathcal{M}} \|\boldsymbol{z} - \boldsymbol{v}\|^2). \tag{195}$$

where $\mathcal{M} = \mathcal{M}_1 \times \ldots \times \mathcal{M}_{K+1}$ denotes the domain of the latent space.

We want to show the uniform almost sure convergence of the terms inside the exponentials in (191) by showing that the following quantity almost surely is bounded from above by zero.

$$\max_{\substack{i,j\in\{1,\ldots,N\},k\in\{1,\ldots,K\}\\j\neq i}} \left| \frac{\|\boldsymbol{y}_i - \boldsymbol{y}_j\|^2_{\boldsymbol{\omega}^{(k)}}}{\epsilon(1/D)\sum_{t=1}^{D_s}\sum_{d=1}^{D_s}\omega_d^{(k,t)}} - \sum_{s=1}^{K} \frac{p_k^{(s)}\left(\|\boldsymbol{x}_i^{(s)} - \boldsymbol{x}_j^{(s)}\|^2 + \|\boldsymbol{x}_i^{(K+1)} - \boldsymbol{x}_j^{(K+1)}\|^2\right)}{\epsilon\sum_{t=1}^{K}\beta_t p_t^{(k)}} \right| \quad (196)$$

$$= \max_{\substack{i,j\in\{1,\ldots,N\},k\in\{1,\ldots,K\}\\j\neq i}} \left| \sum_{s=1}^{K} \left( \frac{\sum_{d=1}^{D_s}\omega_d^{(k,s)}(\boldsymbol{P}^{(s)}(\tilde{\boldsymbol{x}}_i^{(s)} - \tilde{\boldsymbol{x}}_j^{(s)}))_d^2}{\epsilon(1/D)\sum_{t=1}^{D_s}\sum_{d=1}^{D_s}\omega_d^{(k,t)}} - \frac{p_k^{(s)}\|\tilde{\boldsymbol{x}}_i^{(s)} - \tilde{\boldsymbol{x}}_j^{(s)}\|^2}{\epsilon\sum_{t=1}^{K}\beta_t p_t^{(k)}} \right) \right| \quad (197)$$

$$\leq \max_{\substack{i,j\in\{1,\ldots,N\},k\in\{1,\ldots,K\}\\j\neq i}} \left| \sum_{s=1}^{K} \left( \frac{\sum_{d=1}^{D_s}\omega_d^{(k,s)}(\boldsymbol{P}^{(s)}(\tilde{\boldsymbol{x}}_i^{(s)} - \tilde{\boldsymbol{x}}_j^{(s)}))_d^2}{\epsilon(1/D)\sum_{t=1}^{D_s}\sum_{d=1}^{D_s}\omega_d^{(k,t)}} - \frac{(1/D_s)\sum_{d=1}^{D_s}\omega_d^{(k,s)}\|\tilde{\boldsymbol{x}}_i^{(s)} - \tilde{\boldsymbol{x}}_j^{(s)}\|^2}{\epsilon(1/D)\sum_{t=1}^{D_s}\sum_{d=1}^{D_s}\omega_d^{(k,t)}} \right) \right| \quad (198)$$

$$+ \max_{\substack{i,j\in\{1,\ldots,N\},k\in\{1,\ldots,K\}\\j\neq i}} \left| \sum_{s=1}^{K} \left( \frac{(1/D_s)\sum_{d=1}^{D_s}\omega_d^{(k,s)}\|\tilde{\boldsymbol{x}}_i^{(s)} - \tilde{\boldsymbol{x}}_j^{(s)}\|^2}{\epsilon(1/D)\sum_{t=1}^{D_s}\sum_{d=1}^{D_s}\omega_d^{(k,t)}} - \frac{p_k^{(s)}\|\tilde{\boldsymbol{x}}_i^{(s)} - \tilde{\boldsymbol{x}}_j^{(s)}\|^2}{\epsilon\sum_{t=1}^{K}\beta_t p_t^{(k)}} \right) \right|,$$

where the first derivation introduces the problem configuration shown in (15) and the notation in (193). The second derivation uses the triangle inequality and the sub-additivity property of the maximum function. Below, we will examine these terms and then combine their results to derive the convergence.

Next, we introduce key properties of the configuration, as outlined in Section 4. In particular, the configuration governs the limiting behavior of the weighting and dimensionality parameters: $(1/D_s)\sum_{d=1}^{D_s}\omega_d^{(k,s)} \to p_s^{(k)}$ and $D_s/D \to \beta_s$ for any $s,k \in \{1,\ldots,K\}$. Consequently, for any $\delta_0 \leq \min(\beta_{\min}, \sqrt{\epsilon}/2)$ there exists $N_0, D_0$ such that for any $N \geq N_0$, and correspondingly $D \geq D_0$, we have

$$|D_s/D - \beta_s| \leq \delta_0 \qquad and \qquad |(1/D_s)\sum_{d=1}^{D_s}\omega_d^{(k,s)} - p_s^{(k)}| \leq \delta_0. \quad (199)$$

Hence, we can derive the following lower bounds for all $s,k \in \{1,\ldots,K\}$ by

$$D_s/D \geq \beta_s - \delta_0 \geq \beta_s - \beta_s/2 = \beta_s/2 > 0 \quad (200)$$

$$(1/D_s)\sum_{d=1}^{D_s}\omega_d^{(k,s)} \geq p_s^{(k)} - \delta_0 \geq \sqrt{\epsilon} - \sqrt{\epsilon}/2 = \sqrt{\epsilon}/2 > 0 \quad (201)$$

$$(1/D)\sum_{d=1}^{D_s}\omega_d^{(k,s)} = (D_s/D)\left((1/D_s)\sum_{d=1}^{D_s}\omega_d^{(k,s)}\right) \geq (\beta_s/2)(\sqrt{\epsilon}/2) > 0. \quad (202)$$

We begin by analyzing the first term in (198), focusing on the individual differences that appear within the summation over $s$. After examining these components, we combine the results to establish bounds on the full sum over $s$. The denominators in these expressions can be uniformly lower bounded using (202) by

$$(1/D)\sum_{s=1}^{K}\sum_{d=1}^{D_s}\omega_d^{(k,s)} \geq \sum_{s=1}^{K}\beta_s\sqrt{\epsilon}/4 = \sqrt{\epsilon}/4 > 0. \quad (203)$$

since $\sum_{s=1}^{K}\beta_s = 1$. Hence, all of the summed elements within the first term are well-defined.

Next, we consider the numerators of the terms appearing in the summation over $s$, focusing on bounding the difference between the corresponding numerators. Specifically, for any $t \in (0,1)$, we bound the probability that this difference deviates from zero by

$$Pr\left[\exists i \neq j \in [N], k,s \in [K]: \left|\sum_{d=1}^{D_s}\omega_d^{(k,s)}(\boldsymbol{P}^{(s)}(\tilde{\boldsymbol{x}}_i - \tilde{\boldsymbol{x}}_j))_d^2 - \|\tilde{\boldsymbol{x}}_i - \tilde{\boldsymbol{x}}_j\|^2 \cdot \frac{\sum_{d=1}^{D_s}\omega_d^{(k,s)}}{D_s}\right| \geq t\right] \quad (204)$$

$$=Pr\left[\exists i\neq j\in[N],k,s\in[K]:\left|\left(\frac{\sum_{d=1}^{D_s}\omega_d^{(k,s)}(\sqrt{D_s}\boldsymbol{P}^{(s)}(\tilde{\boldsymbol{x}}_i-\tilde{\boldsymbol{x}}_j))_d^2}{\sum_{d=1}^{D_s}\omega_d^{(k,s)}}\right)-\|\tilde{\boldsymbol{x}}_i-\tilde{\boldsymbol{x}}_j\|^2\right|\geq t\cdot\frac{D_s}{\sum_{d=1}^{D_s}\omega_d^{(k,s)}}\right]\quad(205)$$

$$\leq Pr\left[\exists i\neq j\in[N],k,s\in[K]:\left|\left(\frac{1}{\sum_{d=1}^{D_s}\omega_d^{(k,s)}}\sum_{d=1}^{D_s}\omega_d^{(k,s)}(\sqrt{D_s}\boldsymbol{P}^{(s)}(\tilde{\boldsymbol{x}}_i-\tilde{\boldsymbol{x}}_j))_d^2\right)-\|\tilde{\boldsymbol{x}}_i-\tilde{\boldsymbol{x}}_j\|^2\right|\geq t\right]\quad(206)$$

$$\leq Pr\left[\exists i\neq j\in[N],k,s\in[K]:\right.\quad(207)$$

$$\left.\left|\left(\frac{1}{\sum_{d=1}^{D_s}\omega_d^{(k,s)}}\sum_{d=1}^{D_s}\omega_d^{(k,s)}(\sqrt{D_s}\boldsymbol{P}^{(s)}(\tilde{\boldsymbol{x}}_i-\tilde{\boldsymbol{x}}_j))_d^2\right)-\|\tilde{\boldsymbol{x}}_i-\tilde{\boldsymbol{x}}_j\|^2\right|\geq\frac{t\|\tilde{\boldsymbol{x}}_i-\tilde{\boldsymbol{x}}_j\|^2}{C}\right]\quad(208)$$

$$\leq Pr\left[\exists i\neq j\in[N],k,s\in[K]:\left|\left(\frac{1}{\sum_{d=1}^{D_s}\omega_d^{(k,s)}}\sum_{d=1}^{D_s}\omega_d^{(k,s)}(\sqrt{D_s}\boldsymbol{P}^{(s)}(\tilde{\boldsymbol{x}}_i-\tilde{\boldsymbol{x}}_j)/\|\tilde{\boldsymbol{x}}_i-\tilde{\boldsymbol{x}}_j\|)_d^2\right)-1\right|\geq\frac{t}{C}\right]\quad(209)$$

$$\leq\sum_{\substack{i,j=1\\j\neq i}}^{N}\sum_{s,k=1}^{K}Pr\left[\left|\left(\frac{1}{\sum_{d=1}^{D_s}\omega_d^{(k,s)}}\sum_{d=1}^{D_s}\omega_d^{(k,s)}(\sqrt{D_s}\boldsymbol{P}^{(s)}(\tilde{\boldsymbol{x}}_i-\tilde{\boldsymbol{x}}_j)/\|\tilde{\boldsymbol{x}}_i-\tilde{\boldsymbol{x}}_j\|)_d^2\right)-1\right|\geq\frac{t}{C}\right],\quad(210)$$

where $[N]=\{1,\ldots,N\}$ and $[K]=\{1,\ldots,K\}$. In the first derivation we divide by $(1/D_s)\sum_{d=1}^{D_s}\omega_d^{(k,s)}$, which is within $(0,1]$ based on its definition in Section 4 and (203). The second and third derivations upper bound the probabilities by replacing the threshold with a smaller value, following the same reasoning as before, and the definition of $C$ in (195). Next, we invoke the independence of the samples and the assumption of a continuous sampling distribution, which implies that the probability of any two data points being identical is zero. We then apply Boole's inequality, which decomposes the events into multiple simpler events.

We can bound each of these probabilities by applying Example 2.11 from (Wainwright, 2019), which has two conditions that must hold independently for each quadruple $(i,j,s,k)$. First, the entries of $\sqrt{D_s}\boldsymbol{P}^{(s)}(\tilde{\boldsymbol{x}}_i-\tilde{\boldsymbol{x}}_j)/\|\tilde{\boldsymbol{x}}_i-\tilde{\boldsymbol{x}}_j\|$ should be independently and identically distributed according to $\mathcal{N}(0,1)$, which is attained as each entry in $P$ is independently and identically distributed according to $\mathcal{N}(0,1/D_s)$ (see Section 4). Second, $t/C$ should be in $(0,1)$, which is attained by the definition of $C$ in (195). Hence, the term above can be upper bounded by

$$\leq\sum_{s,k=1}^{K}2N^2\exp\left(-t^2(\sum_{d=1}^{D_s}\omega_d^{(k,s)})/(8C^2)\right)\quad(211)$$

$$\leq2N^2K^2\exp\left(-t^2D\sqrt{\epsilon}\beta_s/(32C^2)\right)\quad(212)$$

$$\leq2N^2K^2\exp\left(-t^2D\sqrt{\epsilon}\beta_{\min}/(32C^2)\right).\quad(213)$$

where we plug in (202).

By combining (203), and (213) we establish the almost sure convergence of the first term in (198) to zero via the Borel–Cantelli lemma. This is done by showing the sum below is finite for any $t\in(0,1)$, where the case for $t\geq1$ follows since it is bounded from above by the case $t<1$.

$$\sum_{N=N_0}^{\infty}Pr\left[\exists i\neq j\in[N],k\in[K]:\right.\quad(214)$$

$$\left.\left|\sum_{s=1}^{K}\left(\frac{\sum_{d=1}^{D_s}\omega_d^{(k,s)}(\boldsymbol{P}^{(s)}(\tilde{\boldsymbol{x}}_i^{(s)}-\tilde{\boldsymbol{x}}_j^{(s)}))_d^2-\frac{1}{D_s}\sum_{d=1}^{D_s}\omega_d^{(k,s)}\|\tilde{\boldsymbol{x}}_i^{(s)}-\tilde{\boldsymbol{x}}_j^{(s)}\|^2}{\epsilon(1/D)\sum_{t=1}^{D_s}\sum_{d=1}^{D_s}\omega_d^{(k,t)}}\right)\right|\geq t\right]$$

$$\leq\sum_{N=N_0}^{\infty}\sum_{i,j=1;j\neq i}^{N}\sum_{k=1}^{K}Pr\left[\left|\sum_{s=1}^{K}\left(\frac{\sum_{d=1}^{D_s}\omega_d^{(k,s)}(\boldsymbol{P}^{(s)}(\tilde{\boldsymbol{x}}_i^{(s)}-\tilde{\boldsymbol{x}}_j^{(s)}))_d^2-\frac{1}{D_s}\sum_{d=1}^{D_s}\omega_d^{(k,s)}\|\tilde{\boldsymbol{x}}_i^{(s)}-\tilde{\boldsymbol{x}}_j^{(s)}\|^2}{\epsilon(1/D)\sum_{t=1}^{D_s}\sum_{d=1}^{D_s}\omega_d^{(k,t)}}\right)\right|\geq t\right]\quad(215)$$

$$\leq\sum_{N=N_0}^{\infty}\sum_{i,j=1;j\neq i}^{N}\sum_{k=1}^{K}Pr\left[\left|\sum_{s=1}^{K}\left(\frac{\sum_{d=1}^{D_s}\omega_d^{(k,s)}(\boldsymbol{P}^{(s)}(\tilde{\boldsymbol{x}}_i^{(s)}-\tilde{\boldsymbol{x}}_j^{(s)}))_d^2-\frac{1}{D_s}\sum_{d=1}^{D_s}\omega_d^{(k,s)}\|\tilde{\boldsymbol{x}}_i^{(s)}-\tilde{\boldsymbol{x}}_j^{(s)}\|^2}{\epsilon\sqrt{\epsilon}/4}\right)\right|\geq t\right]\quad(216)$$

$$= \sum_{N=N_0}^{\infty} \sum_{i,j=1;j\neq i}^{N} \sum_{k=1}^{K} Pr\left[\left|\sum_{s=1}^{K}\left(\sum_{d=1}^{D_s}\omega_d^{(k,s)}(\boldsymbol{P}^{(s)}(\tilde{\boldsymbol{x}}_i^{(s)} - \tilde{\boldsymbol{x}}_j^{(s)}))_d^2 - \frac{1}{D_s}\sum_{d=1}^{D_s}\omega_d^{(k,s)}\|\tilde{\boldsymbol{x}}_i^{(s)} - \tilde{\boldsymbol{x}}_j^{(s)}\|^2\right)\right| \geq \frac{t\epsilon^{3/2}}{4}\right] \tag{217}$$

$$= \sum_{N=N_0}^{\infty} \sum_{i,j=1;j\neq i}^{N} \sum_{k,s=1}^{K} Pr\left[\left|\sum_{d=1}^{D_s}\omega_d^{(k,s)}(\boldsymbol{P}^{(s)}(\tilde{\boldsymbol{x}}_i^{(s)} - \tilde{\boldsymbol{x}}_j^{(s)}))_d^2 - \frac{1}{D_s}\sum_{d=1}^{D_s}\omega_d^{(k,s)}\|\tilde{\boldsymbol{x}}_i^{(s)} - \tilde{\boldsymbol{x}}_j^{(s)}\|^2\right| \geq \frac{t\epsilon^{3/2}}{4K}\right] \tag{218}$$

$$\leq \sum_{N=N_0}^{\infty} 2N^2 K^2 \exp\left(-t^2 D\epsilon^3 \beta_{\min}/(2^9 K^2 C^2)\right) \tag{219}$$

$$\tag{220}$$

$$\leq \sum_{N=N_0}^{\infty} 2N^2 K^2 N^{-t^2(D/\log(N))\epsilon^3\beta_{\min}/(2^9 K^2 C^2)} \tag{221}$$

$$< \infty, \tag{222}$$

where the first derivation follows from Boole's inequality. The second and third steps increase the left-hand side based on lower bounding its denominator using (203), and the definition of $\epsilon$ being strictly positive. The fourth derivation invokes Lemma I.5. Next, we apply the upper bound from (213) as $1/K, t, \epsilon < 1$. The final derivation follows from the asymptotic assumption $D/\log(N) \to \infty$, as stated in Section 4, which guarantees that the sum is finite.

We now turn our attention to the second term in (198), which we analyze separately from the first. We begin by examining the behavior of the numerators and denominators of the individual terms within the summation. Once these components are understood, we will combine the results to analyze the term as a whole. Specifically, for all $k, s \in \{1, \ldots, K\}$,

$$(1/D_s)\sum_{d=1}^{D_s}\omega_d^{(k,s)} \xrightarrow{D,N\to\infty} p_k^{(s)} \tag{223}$$

$$(1/D)\sum_{t=1}^{K}\sum_{d=1}^{D_t}\omega_d^{(k,t)} = \sum_{t=1}^{K}(D_t/D)\left((1/D_t)\sum_{d=1}^{D_t}\omega_d^{(k,t)}\right) \xrightarrow{D,N\to\infty} \sum_{t=1}^{K}\beta_t p_t^{(k)} \tag{224}$$

$$\frac{(1/D_s)\sum_{d=1}^{D_s}\omega_d^{(k,s)}}{(1/D)\sum_{t=1}^{K}\sum_{d=1}^{D_t}\omega_d^{(k,t)}} \xrightarrow{D,N\to\infty} \frac{p_k^{(s)}}{\sum_{t=1}^{K}\beta_t p_t^{(k)}}, \tag{225}$$

where we use the algebraic limit theorem. The bottom limit follows from the two limits above it and from the strict positivity of the denominator, as established in (203).

Hence, we can establish the following limit, which bounds the asymptotic value of the second term in (198):

$$\max_{\substack{i,j\in\{1,\ldots,N\},k\in\{1,\ldots,K\}\\ j\neq i}} \left|\sum_{s=1}^{K}\frac{\sum_{d=1}^{D_s}\omega_d^{(k,s)}\|\tilde{\boldsymbol{x}}_i^{(s)} - \tilde{\boldsymbol{x}}_j^{(s)}\|^2}{\epsilon(1/D)\sum_{t=1}^{K}\sum_{d=1}^{D_t}\omega_d^{(k,t)}} - \sum_{s=1}^{K}\frac{p_k^{(s)}\|\tilde{\boldsymbol{x}}_i^{(s)} - \tilde{\boldsymbol{x}}_j^{(s)}\|^2}{\epsilon\sum_{t=1}^{K}\beta_t p_t^{(k)}}\right| \tag{226}$$

$$\leq \max_{\substack{i,j\in\{1,\ldots,N\},k\in\{1,\ldots,K\}\\ j\neq i}} \sum_{s=1}^{K}\frac{\|\tilde{\boldsymbol{x}}_i^{(s)} - \tilde{\boldsymbol{x}}_j^{(s)}\|^2}{\epsilon}\cdot\left|\frac{\sum_{d=1}^{D_s}\omega_d^{(k,s)}}{(1/D)\sum_{t=1}^{K}\sum_{d=1}^{D_t}\omega_d^{(k,t)}} - \frac{p_k^{(s)}}{\sum_{t=1}^{K}\beta_t p_t^{(k)}}\right| \tag{227}$$

$$\leq \max_{k\in\{1,\ldots,K\}} \frac{C}{\epsilon}\cdot\sum_{s=1}^{K}\left|\frac{\sum_{d=1}^{D_s}\omega_d^{(k,s)}}{(1/D)\sum_{t=1}^{K}\sum_{d=1}^{D_t}\omega_d^{(k,t)}} - \frac{p_k^{(s)}}{\sum_{t=1}^{K}\beta_t p_t^{(k)}}\right| \tag{228}$$

$$\leq \frac{KC}{\epsilon}\cdot\max_{s,k\in\{1,\ldots,K\}}\left|\frac{\sum_{d=1}^{D_s}\omega_d^{(k,s)}}{(1/D)\sum_{t=1}^{K}\sum_{d=1}^{D_t}\omega_d^{(k,t)}} - \frac{p_k^{(s)}}{\sum_{t=1}^{K}\beta_t p_t^{(k)}}\right| \tag{229}$$

$$\xrightarrow{D,N\to\infty} 0, \tag{230}$$

where in the first derivation we apply the triangle inequality, and in the second we invoke the definition of $C$ as given in (195). We then use the fact that the maximum is greater than or equal to the mean. Finally, we apply (225) which establishes that for each of the $K^2$ sequences— indexed by $s, k = 1, \ldots, K$— converges deterministically to their limit.

We can conclude the first part of the proof, by showing that (196) can be bounded almost surely from above by zero using (222) and (230).

$$\max_{\substack{i,j\in\{1,\dots,N\},k\in\{1,\dots,K\}\\j\neq i}}\left|\frac{\|\boldsymbol{y}_i-\boldsymbol{y}_j\|^2_{\boldsymbol{\omega}(\boldsymbol{k})}}{\epsilon(1/D)\sum_{t=1}^{D_s}\sum_{d=1}^{D_s}\omega_d^{(k,t)}}-\sum_{s=1}^{K}\frac{p_k^{(s)}\|\tilde{\boldsymbol{x}}_i^{(s)}-\tilde{\boldsymbol{x}}_j^{(s)}\|^2}{\epsilon\sum_{t=1}^{K}\beta_t p_s^{(k)}}\right|\xrightarrow[a.s.]{D,N\to\infty}0. \tag{231}$$

Now, we can conclude the proof by

$$\left|\frac{1}{N}\sum_{k=1}^{K}\sum_{i=1}^{N}\log\left(\frac{1}{N}\sum_{j=1;j\neq i}^{N}\exp\left(-\frac{\|\boldsymbol{y}_i-\boldsymbol{y}_j\|^2_{\boldsymbol{\omega}(\boldsymbol{k})}}{\epsilon(1/D)\sum_{t=1}^{D_s}\sum_{d=1}^{D_s}\omega_d^{(k,t)}}\right)\right)\right. \tag{232}$$

$$\left.-\frac{1}{N}\sum_{k=1}^{K}\sum_{i=1}^{N}\log\left(\frac{1}{N}\sum_{j=1;j\neq i}^{N}\exp\left(-\sum_{s=1}^{K}\frac{p_k^{(s)}\|\tilde{\boldsymbol{x}}_i-\tilde{\boldsymbol{x}}_j\|^2}{\epsilon\sum_{t=1}^{K}\beta_t p_s^{(k)}}\right)\right)\right|$$

$$\leq\frac{1}{N}\sum_{k=1}^{K}\sum_{i=1}^{N}\left|\log\left(\frac{1}{N}\sum_{j=1;j\neq i}^{N}\exp\left(-\frac{\|\boldsymbol{y}_i-\boldsymbol{y}_j\|^2_{\boldsymbol{\omega}(\boldsymbol{k})}}{\epsilon(1/D)\sum_{t=1}^{D_s}\sum_{d=1}^{D_s}\omega_d^{(k,t)}}\right)\right)\right. \tag{233}$$

$$\left.-\log\left(\frac{1}{N}\sum_{j=1;j\neq i}^{N}\exp\left(-\sum_{s=1}^{K}\frac{p_k^{(s)}\|\tilde{\boldsymbol{x}}_i-\tilde{\boldsymbol{x}}_j\|^2}{\epsilon\sum_{t=1}^{K}\beta_t p_s^{(k)}}\right)\right)\right|$$

$$=\frac{1}{N}\sum_{k=1}^{K}\sum_{i=1}^{N}\left|\log\left(\sum_{j=1;j\neq i}^{N}\exp\left(-\frac{\|\boldsymbol{y}_i-\boldsymbol{y}_j\|^2_{\boldsymbol{\omega}(\boldsymbol{k})}}{\epsilon(1/D)\sum_{t=1}^{D_s}\sum_{d=1}^{D_s}\omega_d^{(k,t)}}\right)\right)\right. \tag{234}$$

$$\left.-\log\left(\sum_{j=1;j\neq i}^{N}\exp\left(-\sum_{s=1}^{K}\frac{p_k^{(s)}\|\tilde{\boldsymbol{x}}_i-\tilde{\boldsymbol{x}}_j\|^2}{\epsilon\sum_{t=1}^{K}\beta_t p_s^{(k)}}\right)\right)\right|$$

$$\leq K\cdot\max_{\substack{i,j\in\{1,\dots,N\},k\in\{1,\dots,K\}\\j\neq i}}\left|\frac{\|\boldsymbol{y}_i-\boldsymbol{y}_j\|^2_{\boldsymbol{\omega}(\boldsymbol{k})}}{\epsilon(1/D)\sum_{t=1}^{D_s}\sum_{d=1}^{D_s}\omega_d^{(k,t)}}-\sum_{s=1}^{K}\frac{p_k^{(s)}\|\tilde{\boldsymbol{x}}_i-\tilde{\boldsymbol{x}}_j\|^2}{\epsilon\sum_{t=1}^{K}\beta_t p_s^{(k)}}\right| \tag{235}$$

$$\xrightarrow[D,N\to\infty]{a.s.}0, \tag{236}$$

where in the first two steps, we apply the triangle inequality together with the logarithm's product rule. Next, we exploit the fact that the log-sum-exp function is 1-Lipschitz (see Appendix A of (El Ghaoui & Gueye, 2008)). Finally, we invoke (231) and note that $K$ is finite. $\qquad\square$

**Lemma I.7.** *Assume the configuration in Section 4. Let $\epsilon\in(0,1)$, and $p_s^{(k)}\in[\sqrt{\epsilon},1-(K-1)\sqrt{\epsilon}]$ and let $\beta_1,\dots,\beta_K\in(0,1)$ that satisfy $\sum_{t=1}^{K}\beta_t=1$. Then, the following convergene is attained*

$$\frac{1}{N}\sum_{k=1}^{K}\sum_{i=1}^{N}\log\frac{1}{N-1}\sum_{j=1;j\neq i}^{N}\exp\left(-\frac{\sum_{s=1}^{K}\left\|\boldsymbol{x}_i^{(s)}-\boldsymbol{x}_j^{(s)}\right\|^2 p_s^{(k)}}{\epsilon\sum_{t=1}^{K}p_t^{(k)}\beta_t}-\frac{\left\|\boldsymbol{x}_i^{(K+1)}-\boldsymbol{x}_j^{(K+1)}\right\|^2\sum_{s=1}^{K}p_s^{(k)}}{\epsilon\sum_{t=1}^{K}p_t^{(k)}\beta_t}\right) \tag{237}$$

$$\xrightarrow[D,N\to\infty]{a.s.}\sum_{k=1}^{K}\mathbb{E}_{\boldsymbol{x}\in\mathcal{M}}\left[\log\mathbb{E}_{\boldsymbol{z}\in\mathcal{M}}\left[\exp\left(-\frac{\sum_{s=1}^{K}\|\boldsymbol{x}^{(s)}-\boldsymbol{z}^{(s)}\|^2 p_s^{(k)}}{\epsilon\sum_{t=1}^{K}p_t^{(k)}\beta_t}-\frac{\left\|\boldsymbol{x}_i^{(K+1)}-\boldsymbol{z}^{(K+1)}\right\|^2\sum_{s=1}^{K}p_s^{(k)}}{\epsilon\sum_{t=1}^{K}p_t^{(k)}\beta_t}\right)\right]\right] \tag{238}$$

*Proof.* For simplicity, we begin the proof by defining some terms and constants that will be used the derivations below

$$O_{i,j,N,k}=\exp\left(-\frac{\sum_{s=1}^{K}\left\|\boldsymbol{x}_i^{(s)}-\boldsymbol{x}_j^{(s)}\right\|^2 p_s^{(k)}}{\epsilon\sum_{t=1}^{K}p_t^{(k)}\beta_t}-\frac{\left\|\boldsymbol{x}_i^{(K+1)}-\boldsymbol{x}_j^{(K+1)}\right\|^2\sum_{s=1}^{K}p_s^{(k)}}{\epsilon\sum_{t=1}^{K}p_t^{(k)}\beta_t}\right) \tag{239}$$

$$O_{i,N,k} = \frac{1}{N-1} \sum_{j=1;j\neq i}^{N} \exp\left(-\frac{\sum_{s=1}^{K} \left\|\boldsymbol{x}_i^{(s)} - \boldsymbol{x}_j^{(s)}\right\|^2 p_s^{(k)}}{\epsilon \sum_{t=1}^{K} p_t^{(k)} \beta_t} - \frac{\left\|\boldsymbol{x}_i^{(K+1)} - \boldsymbol{x}_j^{(K+1)}\right\|^2 \sum_{s=1}^{K} p_s^{(k)}}{\epsilon \sum_{t=1}^{K} p_t^{(k)} \beta_t}\right) \tag{240}$$

$$\tilde{O}_{i,k} = \mathbb{E}_{\boldsymbol{z}\in\mathcal{M}}\left[\exp\left(-\frac{\sum_{s=1}^{K} \|\boldsymbol{x}_i^{(s)} - \boldsymbol{z}^{(s)}\|^2 p_s^{(k)}}{\epsilon \sum_{t=1}^{K} p_t^{(k)} \beta_t} - \frac{\left\|\boldsymbol{x}_i^{(K+1)} - \boldsymbol{z}^{(K+1)}\right\|^2 \sum_{s=1}^{K} p_s^{(k)}}{\epsilon \sum_{t=1}^{K} p_t^{(k)} \beta_t}\right)\right] \tag{241}$$

$$\tilde{\tilde{O}}_{k} = \mathbb{E}_{\boldsymbol{x}\in\mathcal{M}}\left[\log\mathbb{E}_{\boldsymbol{z}\in\mathcal{M}}\left[\exp\left(-\frac{\sum_{s=1}^{K} \|\boldsymbol{x}^{(s)} - \boldsymbol{z}^{(s)}\|^2 p_s^{(k)}}{\epsilon \sum_{t=1}^{K} p_t^{(k)} \beta_t} - \frac{\left\|\boldsymbol{x}_i^{(K+1)} - \boldsymbol{z}^{(K+1)}\right\|^2 \sum_{s=1}^{K} p_s^{(k)}}{\epsilon \sum_{t=1}^{K} p_t^{(k)} \beta_t}\right)\right]\right] \tag{242}$$

$$C = \max_{\boldsymbol{x},\boldsymbol{z}\in\mathcal{M}} \|\boldsymbol{x} - \boldsymbol{z}\|^2 \tag{243}$$

$$\tilde{C} = \exp(2CK/\epsilon^{3/2}) \tag{244}$$

First, We can see that $\mathbb{E}[O_{i,j,N,k}] = \mathbb{E}[O_{i,N,k}] = \tilde{O}_{i,k}$, where the expectations are over $\{\boldsymbol{x}_1,\ldots,\boldsymbol{x}_N\}/\{\boldsymbol{x}_i\}$ that are drawn independently from the same distribution. Second, we can see that $\tilde{\tilde{O}}_k = \mathbb{E}_{\boldsymbol{x}_i\in\mathcal{M}}[\log\tilde{O}_{i,k}]$. Finally, we can see that $\tilde{C} > 1$.

Next, we prove that the terms $O_{i,j,N,k}, O_{i,N,k}$ and $\tilde{O}_{i,k}$ are bounded within $[\tilde{C}^{-1}, 1]$ for each $i \in \{1,\ldots,N\}$ and $k \in \{1,\ldots,K\}$. A direct result of this will be that both $\log\tilde{O}_{i,k}$ and $\tilde{\tilde{O}}_k$ will be bounded by $[-2CK/\epsilon^{3/2}, 0]$. The terms $O_{i,N,k}$ and $\tilde{O}_{i,k}$ are actually an average or expectation operator over $O_{i,j,N,k}$, hence their bounds should be the same as the latter term.

To begin with, the upper bound of $O_{i,j,N,k}$ is 1 as the exponential argument is non-positive. As for the lower bound of $O_{i,j,N,k}$,

$$O_{i,j,N,k} = \exp\left(-\frac{\sum_{s=1}^{K} \|\boldsymbol{x}_i^{(s)} - \boldsymbol{x}_j^{(s)}\|^2 p_s^{(k)}}{\epsilon \sum_{t=1}^{K} p_t^{(k)} \beta_t} - \frac{\left\|\boldsymbol{x}_i^{(K+1)} - \boldsymbol{x}_j^{(K+1)}\right\|^2 \sum_{s=1}^{K} p_s^{(k)}}{\epsilon \sum_{t=1}^{K} p_t^{(k)} \beta_t}\right) \tag{245}$$

$$\geq \exp\left(-\frac{\sum_{s=1}^{K} C p_s^{(k)}}{\epsilon \sum_{t=1}^{K} p_t^{(k)} \beta_t} - \frac{C \sum_{s=1}^{K} p_s^{(k)}}{\epsilon \sum_{t=1}^{K} p_t^{(k)} \beta_t}\right) \tag{246}$$

$$\geq \exp\left(-\frac{\sum_{s=1}^{K} C \cdot 1}{\epsilon \sum_{t=1}^{K} \sqrt{\epsilon}\beta_t} - \frac{C \sum_{s=1}^{K} 1}{\epsilon \sum_{t=1}^{K} \sqrt{\epsilon}\beta_t}\right) \tag{247}$$

$$\geq \exp\left(-\frac{CK}{\epsilon\sqrt{\epsilon}} - \frac{CK}{\epsilon\sqrt{\epsilon}}\right) \tag{248}$$

$$= \tilde{C}^{-1}, \tag{249}$$

where the first derivation uses the definition of $C$ in (243), and the second uses the bounds of $p_s^{(k)}$. Then, the third and fourth derivations are based on the property that $\sum_{t=1}^{K} \beta_t = 1$ and the definition of $\tilde{C}$ in (244), respecitvel.

Now we are ready to use these terms to define the probability of the two terms mentioned in the statement being close by for finite $N$ for some $\delta > 0$

$$Pr\left[\left|\frac{1}{N}\sum_{k=1}^{K}\sum_{i=1}^{N}\log(O_{i,N,k}) - \sum_{k=1}^{K}\tilde{\tilde{O}}_k\right| \geq \delta\right] \tag{250}$$

$$= Pr\left[\left|\frac{1}{N}\sum_{k=1}^{K}\sum_{i=1}^{N}\log(O_{i,N,k}) - \frac{1}{N}\sum_{k=1}^{K}\sum_{i=1}^{N}\log(\tilde{O}_{i,N,k}) + \frac{1}{N}\sum_{k=1}^{K}\sum_{i=1}^{N}\log(\tilde{O}_{i,N,k}) - \sum_{k=1}^{K}\tilde{\tilde{O}}_k\right| \geq \delta\right] \tag{251}$$

$$\leq Pr\left[\left|\frac{1}{N}\sum_{k=1}^{K}\sum_{i=1}^{N}\log(O_{i,N,k}/\tilde{O}_{i,N,k})\right| \geq \delta/2\right] + Pr\left[\left|\frac{1}{N}\sum_{k=1}^{K}\sum_{i=1}^{N}\log(\tilde{O}_{i,N,k}) - \sum_{k=1}^{K}\tilde{\tilde{O}}_k\right| \geq \delta/2\right], \tag{252}$$

where the derivations use Lemma I.5.

First, we bound the second term, and then we will get to the first term. By definition, $\tilde{\tilde{O}} = \mathbb{E}[\log(\tilde{O}_{i,k})]$ where the expectation is with respect to $\boldsymbol{x}_i \in \mathcal{M}$, for any $i \in \{1, \dots, N\}$ and $k \in \{1, \dots, K\}$.

$$Pr\left[\left|\frac{1}{N}\sum_{k=1}^{K}\sum_{i=1}^{N}\log(\tilde{O}_{i,N,k}) - \sum_{k=1}^{K}\tilde{\tilde{O}}_k\right| \geq \delta/2\right] \leq \sum_{k=1}^{K}Pr\left[\left|\frac{1}{N}\sum_{i=1}^{N}\log(\tilde{O}_{i,N,k}) - \tilde{\tilde{O}}_k\right| \geq \delta/2K\right] \tag{253}$$

$$\leq 2K\exp\left(-\frac{2\delta^2 N^2/(4K^2)}{N(\log(1) - \log(\tilde{C}^{-1}))^2}\right) \tag{254}$$

$$= 2K\exp\left(-\frac{\delta^2 N/(2K^2)}{\log(\tilde{C}^{-1})^2}\right) \tag{255}$$

$$= K\exp\left(-\frac{2\delta^2 N}{\log(\tilde{C})^2}\right) \tag{256}$$

where we use Lemma I.5 for the first derivation. In next two derivations, we use the Hoeffding's inequality along with the bound shown above that $\tilde{O}_{i,N,k} \in [\tilde{C}^{-1}, 1]$.

Now, we bound the second term. This term is well defined as both $O_{i,N,k}$ and $\tilde{O}_{i,k}$ are strictly positive as derived above.

$$Pr\left[\left|\frac{1}{N}\sum_{k=1}^{K}\sum_{i=1}^{N}\log(O_{i,N,k}/\tilde{O}_{i,N,k})\right| \geq \delta/2\right] \tag{257}$$

$$\leq \sum_{k=1}^{K}\sum_{i=1}^{N}Pr\left(\left|\log\frac{O_{i,N,k}}{\tilde{O}_{i,k}}\right| \geq \delta/2K\right) \tag{258}$$

$$= \sum_{k=1}^{K}\sum_{i=1}^{N}Pr\left(\left\{\frac{O_{i,N,k}}{\tilde{O}_{i,k}} \geq \exp(\delta/2K)\right\} \bigcup \left\{\frac{O_{i,N,k}}{\tilde{O}_{i,k}} \leq \exp(-\delta/2K)\right\}\right) \tag{259}$$

$$= \sum_{k=1}^{K}\sum_{i=1}^{N}Pr\left(\left\{\frac{O_{i,N,k}}{\tilde{O}_{i,k}} - 1 \geq \exp(\delta/2K) - 1\right\} \bigcup \left\{\frac{O_{i,N,k}}{\tilde{O}_{i,k}} - 1 \leq \exp(-\delta/2K) - 1\right\}\right) \tag{260}$$

$$\leq \sum_{k=1}^{K}\sum_{i=1}^{N}Pr\left(\left\{\frac{O_{i,N,k}}{\tilde{O}_{i,k}} - 1 \geq 1 - \exp(-\delta/2K)\right\} \bigcup \left\{\frac{O_{i,N,k}}{\tilde{O}_{i,k}} - 1 \leq -(1 - \exp(-\delta/2K))\right\}\right) \tag{261}$$

$$= \sum_{k=1}^{K}\sum_{i=1}^{N}Pr\left(\left|\frac{O_{i,N,k}}{\tilde{O}_{i,k}} - 1\right| \geq 1 - \exp(-\delta/2K)\right) \tag{262}$$

$$= \sum_{k=1}^{K}\sum_{i=1}^{N}Pr\left(\left|\frac{O_{i,N,k}}{\tilde{O}_{i,k}} - 1\right| \geq \tilde{\delta}\right) \tag{263}$$

where the first derivation is based on Lemma I.5. In (261) we use the fact that $\exp(\delta) - 1 = \exp(\delta/2K)(1 - \exp(-\delta/2K)) \geq 1 - \exp(-\delta/2K) \geq 0$, and in (263) we denote $\tilde{\delta} = 1 - exp(-\delta/2K)$. Now, we can use the fact that $O_{i,N,k}$ can be written as an average over $O_{i,j,N,k}$ and that $\mathbb{E}[O_{i,j,N,k}] = \tilde{O}_{i,k}$.

$$= \sum_{k=1}^{K}\sum_{i=1}^{N}Pr\left(\left|\frac{O_{i,N,k}}{\tilde{O}_{i,k}} - 1\right| \geq \tilde{\delta}\right) \tag{264}$$

$$= \sum_{k=1}^{K}\sum_{i=1}^{N}Pr\left(\left|\frac{1}{N-1}\sum_{j=1,j\neq i}^{N}\frac{O_{i,j,N,k}}{\tilde{O}_{i,k}} - 1\right| \geq \tilde{\delta}\right) \tag{265}$$

$$\leq \sum_{k=1}^{K}\sum_{i=1}^{N}\exp\left(-\frac{2\tilde{\delta}^2}{(N-1)\left(\tilde{C}/(N-1) - \tilde{C}^{-1}/(N-1)\right)^2}\right) \tag{266}$$

$$\leq \sum_{k=1}^{K}\sum_{i=1}^{N}\exp\left(-2\tilde{\delta}^2(N-1)/\tilde{C}^2\right) \tag{267}$$

$$= KN \exp\left(-2\tilde{\delta}^2 (N-1)/\tilde{C}^2\right), \tag{268}$$

where we use the Hoeffding inequality as $O_{i,N,k} \in [\tilde{C}^{-1}, 1]$ for all $i \in \{1, \ldots, N\}$ and $k \in \{1, \ldots, K\}$ as noted above. Additionally, we use the bound from above to bound $O_{i,N,k}/\tilde{O}_{i,k} \in [\tilde{C}^{-1}, \tilde{C}]$ and that $\tilde{C} \geq 1$.

By combining the above results, we can conclude the almost-sure convergence in (237) via the Borel-Cantelli Lemma. It applies as the sum below is finite for any $\delta > 0$.

$$\sum_{N=N_0}^{\infty} Pr\left[\left|\frac{1}{N}\sum_{k=1}^{K}\sum_{i=1}^{N}\log(O_{i,N,k}) - \sum_{k=1}^{K}\tilde{\tilde{\theta}}_k\right| \geq \delta\right] \tag{269}$$

$$\leq \sum_{N=N_0}^{\infty} K\exp\left(-\frac{2\delta^2 N}{\log(\tilde{C})^2}\right) + KN\exp\left(-\frac{2\tilde{\delta}^2(N-1)}{\tilde{C}^2}\right) \tag{270}$$

$$< \infty, \tag{271}$$

where the exponential decay ensures the finiteness of the sum.

$\square$

**Lemma I.8.** *Let $f$ be a non-vanishing smooth distribution over a smooth compact Riemannian manifold $\mathcal{M}$ and let $\mathbf{y} \in \mathcal{M}$. Let $\beta_1, \ldots, \beta_K \in (0,1)$ that satisfy $\sum_{k=1}^{K}\beta_k = 1$. There exists $\tilde{\epsilon}(\mathcal{M}, f)$ such that for any $\epsilon < \tilde{\epsilon}^2$ any $q_1, \ldots, q_K \in [\sqrt{\epsilon}, 1 - (K-1)\sqrt{\epsilon}]$ and any $s \in \{1, \ldots, K\}$:*

$$\log\left(\int_{x\in\mathcal{M}}\exp\left(-\frac{\|\mathbf{x}-\mathbf{y}\|^2 q_s}{\epsilon\sum_{t=1}^{K}q_t\beta_t}\right)f(\mathbf{x})d\mathbf{x}\right) \tag{272}$$

$$= \frac{dim(\mathcal{M})}{2}\log\left(\frac{\pi\epsilon\sum_{t=1}^{K}q_t\beta_t}{q_s}\right) + \log\left(f(\mathbf{y})\right) + O\left(\sqrt{\epsilon}\right),$$

*and*

$$\log\left(\int_{x\in\mathcal{M}}\exp\left(-\frac{\|\mathbf{x}-\mathbf{y}\|^2\sum_{s=1}^{K}q_s}{\epsilon\sum_{t=1}^{K}q_t\beta_t}\right)f(\mathbf{x})d\mathbf{x}\right) \tag{273}$$

$$= \frac{dim(\mathcal{M})}{2}\log\left(\frac{\pi\epsilon\sum_{t=1}^{K}q_t\beta_t}{\sum_{s=1}^{K}q_s}\right) + \log\left(f(\mathbf{y})\right) + O\left(\sqrt{\epsilon}\right).$$

*Proof.* We begin by showing (272). Based on Lemma I.2, there exists $\tilde{\epsilon}(\mathcal{M}, f) \leq 1$ so that for any $\epsilon < \tilde{\epsilon}(\mathcal{M}, f)$

$$\int_{\mathbf{x}\in\mathcal{M}}\exp\left(-\frac{\|\mathbf{x}-\mathbf{y}\|^2}{\epsilon}\right)f(\mathbf{x})d\mathbf{x} = (\pi\epsilon)^{dim(\mathcal{M})/2}f(\mathbf{y})(1 + O(\epsilon)). \tag{274}$$

Second, if $\epsilon \leq \tilde{\epsilon}^2$ then for any $s \in \{1, \ldots, K\}$

$$\frac{\epsilon\sum_{t=1}^{K}q_t\beta_t}{q_s} \leq \frac{\epsilon\cdot\sum_{t=1}^{K}\beta_t\max_{r\in\{1,\ldots,K\}}q_r}{q_s} < \frac{\epsilon\cdot 1\cdot 1}{\sqrt{\epsilon}} = \sqrt{\epsilon} \leq \tilde{\epsilon}, \tag{275}$$

by using the upper bound of $q_1, \ldots, q_K$ along with their positivity and non negativity of $\beta_1, \ldots, \beta_K$ for the first derivation, and the bounds of $q_1, \ldots, q_K$ for the second. Therefore,

$$\log\left(\int_{x\in\mathcal{M}}\exp\left(-\frac{\|\mathbf{x}-\mathbf{y}\|^2 q_s}{\epsilon\sum_{t=1}^{K}q_t\beta_t}\right)f(\mathbf{y})d\mathbf{y}\right) \tag{276}$$

$$= \log\left(\left(\frac{\pi\epsilon\sum_{t=1}^{K}q_t\beta_t}{q_s}\right)^{dim(\mathcal{M})/2}f(\mathbf{x})\left(1 + O\left(\sqrt{\epsilon}\right)\right)\right) \tag{277}$$

$$= \frac{\dim(\mathcal{M})}{2} \log \left( \frac{\pi \epsilon \sum_{t=1}^{K} q_t \beta_t}{q_s} \right) + \log \left( f(\boldsymbol{x}) \right) + \log \left( 1 + O\left( \sqrt{\epsilon} \right) \right) \tag{278}$$

$$= \frac{\dim(\mathcal{M})}{2} \log \left( \frac{\pi \epsilon \sum_{t=1}^{K} q_t \beta_t}{q_s} \right) + \log \left( f(\boldsymbol{x}) \right) + O\left( \sqrt{\epsilon} \right). \tag{279}$$

where we used Lemma I.2 and (276) in the first derivation, and the identity $\log(1 + a) \le a$ for all $a \ge 0$ in the second derivation, which follows from the identity $\exp(a) \ge 1 + a$.

Now, as for (273). If $\epsilon < \tilde{\epsilon}^2$ then

$$\frac{\epsilon \sum_{t=1}^{K} q_t \beta_t}{\sum_{s=1}^{K} q_s} \le \frac{\epsilon \sum_{t=1}^{K} q_t \beta_t}{\min_{r \in \{1,\dots,K\}} q_r} < \sqrt{\epsilon} < \tilde{\epsilon}, \tag{280}$$

where the left inequality is due to the non-negativity of $q_1, \dots, q_K$, and the right uses the same derivation as in (275). Then, (273) follows using a similar derivation as done in (276).

$\square$

**Lemma I.9.** *Let $f_1, \dots, f_{K+1}$ be non-vanishing smooth distributions over smooth compact Riemannian manifolds $\mathcal{M}_1, \dots, \mathcal{M}_{K+1}$, respectively. Define $f$ to be a non vanishing smooth distribution over the product manifold $\mathcal{M} = \mathcal{M}_1 \times \dots \times \mathcal{M}_{K+1}$ by $f(\boldsymbol{x}^{(1)}, \dots, \boldsymbol{x}^{(K+1)}) = \prod_{k=1}^{K+1} f_k(\boldsymbol{x}^{(k)})$ for any $(\boldsymbol{x}^{(1)}, \dots, \boldsymbol{x}^{(K+1)}) \in \mathcal{M}$.*

*Let $\beta_1, \dots, \beta_K \in (0,1)$ that satisfy $\sum_k \beta_k = 1$. There exists $\tilde{\epsilon}(\mathcal{M}, f)$ such that for any $\epsilon < \tilde{\epsilon}^2$ and any $q_1, \dots, q_K \in [\sqrt{\epsilon}, 1 - (K-1)\sqrt{\epsilon}]$:*

$$\int_{\boldsymbol{z} \in \mathcal{M}} f(\boldsymbol{z}) \log \left( \int_{\boldsymbol{x} \in \mathcal{M}} \left( \prod_{s=1}^{K} \exp \left( -\frac{\|\boldsymbol{x}^{(s)} - \boldsymbol{z}^{(s)}\|^2 q_s}{\epsilon \sum_{t=1}^{K} q_t \beta_t} - \frac{\|\boldsymbol{x}^{(K+1)} - \boldsymbol{z}^{(K+1)}\|^2 q_s}{\epsilon \sum_{t=1}^{K} q_t \beta_t} \right) \right) f(\boldsymbol{x}) d\boldsymbol{x} \right) d\boldsymbol{z} \tag{281}$$

$$= \sum_{s=1}^{K} \frac{\dim(\mathcal{M}_k)}{2} \log \left( \frac{\pi \epsilon \sum_{t=1}^{K} q_t \beta_t}{q_s} \right) + \frac{\dim(\mathcal{M}_{K+1})}{2} \log \left( \frac{\pi \epsilon \sum_{t=1}^{K} q_t \beta_t}{\sum_{s=1}^{K} q_s} \right) - h(f) + O(\sqrt{\epsilon}), \tag{282}$$

*where $h(f)$ is the differential entropy of $f$ defined by $h(f) = -\int_{\boldsymbol{z} \in \mathcal{M}} f(\boldsymbol{z}) \log(f(\boldsymbol{z})) d\boldsymbol{z}$.*

*Proof.* We begin by rewriting the terms inside the logarithmic term in (281) using the separability properties of $\mathcal{M}$ and $f$

$$\log \int_{\boldsymbol{x} \in \mathcal{M}} \left( \prod_{s=1}^{K} \exp \left( -\frac{\|\boldsymbol{x}^{(s)} - \boldsymbol{z}^{(s)}\|^2 q_s}{\epsilon \sum_{t=1}^{K} q_t \beta_t} - \frac{\|\boldsymbol{x}^{(K+1)} - \boldsymbol{z}^{(K+1)}\|^2 q_s}{\epsilon \sum_{t=1}^{K} q_t \beta_t} \right) \right) f(\boldsymbol{x}) d\boldsymbol{x} \tag{283}$$

$$= \log \left( \left( \prod_{s=1}^{K} \int_{\boldsymbol{x}^{(s)} \in \mathcal{M}_s} \exp \left( -\frac{\|\boldsymbol{x}^{(s)} - \boldsymbol{z}^{(s)}\|^2 q_s}{\epsilon \sum_{t=1}^{K} q_t \beta_t} \right) f_s(\boldsymbol{x}^{(s)}) d\boldsymbol{x}^{(s)} \right) \right. \tag{284}$$

$$\left. \cdot \left( \int_{\boldsymbol{x}^{(K+1)} \in \mathcal{M}_{K+1}} \exp \left( -\frac{\|\boldsymbol{x}^{(K+1)} - \boldsymbol{z}^{(K+1)}\|^2 \sum_{s=1}^{K} q_s}{\epsilon \sum_{t=1}^{K} q_t \beta_t} \right) f_{K+1}(\boldsymbol{x}^{(K+1)}) d\boldsymbol{x}^{(K+1)} \right) \right)$$

$$= \sum_{s=1}^{K} \log \left( \int_{\boldsymbol{x}^{(s)} \in \mathcal{M}_s} \exp \left( -\frac{\|\boldsymbol{x}^{(s)} - \boldsymbol{z}^{(s)}\|^2 q_s}{\epsilon \sum_{t=1}^{K} q_t \beta_t} \right) f_k(\boldsymbol{x}^{(s)}) d\boldsymbol{x}^{(s)} \right) \tag{285}$$

$$+ \log \left( \int_{\boldsymbol{x}^{(K+1)} \in \mathcal{M}_{K+1}} \exp \left( -\frac{\|\boldsymbol{x}^{(K+1)} - \boldsymbol{z}^{(K+1)}\|^2 \sum_{s=1}^{K} q_s}{\epsilon \sum_{t=1}^{K} q_t \beta_t} \right) f_{K+1}(\boldsymbol{x}^{(K+1)}) d\boldsymbol{x}^{(K+1)} \right).$$

Based on Lemma I.8 there exists $\tilde{\epsilon}(\mathcal{M}, f) < 1$ such that for any $\epsilon < \tilde{\epsilon}^2$ and $q_1, \dots, q_K \in [\sqrt{\epsilon}, 1 - (K-1)\sqrt{\epsilon}]$ then the equation above is equal to

$$= \sum_{s=1}^{K} \left( \frac{\dim(\mathcal{M}_s)}{2} \log \left( \frac{\pi \epsilon \sum_{t=1}^{K} q_t \beta_t}{q_s} \right) + \log \left( f_s(\boldsymbol{z}^{(s)}) \right) \right) \tag{286}$$

$$+\frac{\dim(\mathcal{M}_{K+1})}{2} \log \left( \frac{\pi\epsilon \sum_{t=1}^{K} q_t \beta_t}{\sum_{s=1}^{K} q_s} \right) + \log \left( f_{K+1}(\boldsymbol{z}^{(K+1)}) \right) + O(\sqrt{\epsilon})$$

$$= \sum_{s=1}^{K} \left( \frac{\dim(\mathcal{M}_s)}{2} \log \left( \frac{\pi\epsilon \sum_{t=1}^{K} q_t \beta_t}{q_s} \right) \right) + \frac{\dim(\mathcal{M}_{K+1})}{2} \log \left( \frac{\pi\epsilon \sum_{t=1}^{K} q_t \beta_t}{\sum_{s=1}^{K} q_s} \right) + \log(f(\boldsymbol{z})) + O(\sqrt{\epsilon}). \tag{287}$$

Now, we can push this term inside (281) and derive that it can be rewritten by

$$\int_{\boldsymbol{z}\in\mathcal{M}} f(\boldsymbol{z}) \Bigg( \sum_{s=1}^{K} \left( \frac{\dim(\mathcal{M}_s)}{2} \log \left( \frac{\pi\epsilon \sum_{t=1}^{K} q_t \beta_t}{q_s} \right) \right) \tag{288}$$

$$+ \frac{\dim(\mathcal{M}_{K+1})}{2} \log \left( \frac{\pi\epsilon \sum_{t=1}^{K} q_t \beta_t}{\sum_{s=1}^{K} q_s} \right) + \log(f(\boldsymbol{z})) + O(\sqrt{\epsilon}) \Bigg) d\boldsymbol{z}$$

$$= \sum_{s=1}^{K} \left( \frac{\dim(\mathcal{M}_s)}{2} \log \left( \frac{\pi\epsilon \sum_{t=1}^{K} q_t \beta_t}{q_s} \right) \right) + \frac{\dim(\mathcal{M}_{K+1})}{2} \log \left( \frac{\pi\epsilon \sum_{t=1}^{K} q_t \beta_t}{\sum_{s=1}^{K} q_s} \right) - h(f) + O(\sqrt{\epsilon}). \tag{289}$$

$\square$

**Theorem 4.3** There exists $\bar{\epsilon}(\mathcal{M}, f) \leq 1$ such that for any $\epsilon < \bar{\epsilon}$ and $p_s^{(k)} \in [\sqrt{\epsilon}, 1 - (K-1)\sqrt{\epsilon}]$, any partitioning solution $\{\boldsymbol{\omega}^{(k)}\}$ (obeying Problem 4.1's constraints) satisfies

$$\min_{\{\tilde{\boldsymbol{W}}^{(k)}\}} \frac{1}{\epsilon N} \left( \tilde{G}\left(\{\tilde{\boldsymbol{W}}^{(k)}\}, \{\boldsymbol{\omega}^{(k)}\}, \{\boldsymbol{y}_i\}\right) + \epsilon \left( \sum_{k=1}^{K} \sum_{i,j=1}^{N} \tilde{W}_{i,j}^{(k)} \log \left( \tilde{W}_{i,j}^{(k)} \right) \right) \right) + K \log(N-1) \tag{16}$$

$$\xrightarrow[N,D\to\infty]{a.s.} \sum_{k=1}^{K} \frac{\dim(\mathcal{M}_{K+1})}{2} \log \left( \frac{\sum_{s=1}^{K} p_s^{(k)}}{\sum_{t=1}^{K} p_t^{(k)} \beta_t} \right) + \sum_{k,s=1}^{K} \frac{\dim(\mathcal{M}_s)}{2} \log \left( \frac{p_s^{(k)}}{\sum_{t=1}^{K} p_t^{(k)} \beta_t} \right) \tag{17}$$

$$+ K \sum_{s=1}^{K+1} \left( h_s(f_s) - \frac{dim(\mathcal{M}_s) \log(\pi\epsilon)}{2} \right) + O(\sqrt{\epsilon}),$$

where $h_s(f_s) = -\int_{\boldsymbol{z}\in\mathcal{M}_s} f_s(\boldsymbol{z}) \log f_s(\boldsymbol{z}) d\boldsymbol{z}$ is the differential entropy of the density $f_s$ over $\mathcal{M}_s$.

*Proof of Theorem 4.3.*
We denote the entire latent data manifold by $\mathcal{M} = \mathcal{M}_1 \times \ldots \times \mathcal{M}_{K+1}$, and its distribution by $f$ defined by $f(\boldsymbol{x}^{(1)}, \ldots, \boldsymbol{x}^{(K+1)}) \equiv \prod_{k=1}^{K+1} f_k(\boldsymbol{x}^{(k)})$. As can be understood, this definition complies with the data definition of Section 4.
Based on Corollary 4.2, a set of affinity matrices $\{\boldsymbol{W}^{(k)}\}_{k=1}^{K}$ that minimize (16) is of the form:

$$W_{i,j}^{(k)} = \frac{A_{i,j}^{(k)}}{\sum_t A_{i,t}^{(k)}} \tag{290}$$

for all $k \in \{1, \ldots, K\}$ and $i, j \in \{1, \ldots, N\}$ and

$$A_{i,j}^{(k)} = \begin{cases} \exp\left( -\frac{\|\boldsymbol{y}_i - \boldsymbol{y}_j\|_{\tilde{\boldsymbol{\omega}}^{(k)}}^2}{\epsilon \cdot (1/D) \sum_d \tilde{\omega}_d^{(k)}} \right) & \text{if } i \neq j \\ 0 & \text{else} \end{cases} . \tag{291}$$

By plugging these affinity matrices in to (16) we get

$$\frac{1}{\epsilon N} \sum_{k=1}^{K} \sum_{i,j=1}^{N} \frac{A_{i,j}^{(k)}}{\sum_{t=1}^{N} A_{i,t}^{(k)}} \cdot \frac{\|\boldsymbol{y}_i - \boldsymbol{y}_j\|_{\boldsymbol{\omega}^{(k)}}^2}{(1/D) \sum_{d=1}^{D} \omega_d^{(k)}} + \frac{1}{N} \sum_{k=1}^{K} \sum_{i,j=1}^{N} \frac{A_{i,j}^{(k)}}{\sum_{t=1}^{N} A_{i,t}^{(k)}} \log \left( \frac{A_{i,j}^{(k)}}{\sum_{t=1}^{N} A_{i,t}^{(k)}} \right) + K \log(N-1) \tag{292}$$

$$= -\frac{1}{N}\sum_{k=1}^{K}\sum_{i,j=1}^{N}\frac{A_{i,j}^{(k)}}{\sum_{t=1}^{N}A_{i,t}^{(k)}}\cdot\log(A_{i,j}^{(k)}) + \frac{1}{N}\sum_{k=1}^{K}\sum_{i=1}^{N}\left(\log(N-1) + \sum_{j=1}^{N}\frac{A_{i,j}^{(k)}}{\sum_{t=1}^{N}A_{i,t}^{(k)}}\log\left(\frac{A_{i,j}^{(k)}}{\sum_{t=1}^{N}A_{i,t}^{(k)}}\right)\right) \tag{293}$$

$$= \frac{1}{N}\sum_{k=1}^{K}\sum_{i=1}^{N}\left(\log(N-1) - \sum_{j=1}^{N}\frac{A_{i,j}^{(k)}}{\sum_{t=1}^{N}A_{i,t}^{(k)}}\log\left(\sum_{t=1}^{N}A_{i,t}^{(k)}\right)\right) \tag{294}$$

$$= \frac{1}{N}\sum_{k=1}^{K}\sum_{i=1}^{N}\left(\log(N-1) - \log\left(\sum_{t=1}^{N}A_{i,t}^{(k)}\right)\right) \tag{295}$$

$$= -\frac{1}{N}\sum_{k=1}^{K}\sum_{i=1}^{N}\left(\log\left(\frac{1}{N-1}\sum_{t=1}^{N}A_{i,t}^{(k)}\right)\right). \tag{296}$$

Now, we can now introduce back the expressions of all $A_{i,j}^{(k)}$, for $i,j=1,\ldots,N$ and $k=1,\ldots,K$ and derive (17).

$$\frac{-1}{N}\sum_{k=1}^{K}\sum_{i=1}^{N}\log\frac{1}{N-1}\sum_{j=1;j\neq i}^{N}\exp\left(\frac{-\|\boldsymbol{y}_i-\boldsymbol{y}_j\|_{\boldsymbol{\omega}^{(k)}}^2}{\epsilon\cdot(1/D)\sum_d\omega_d^{(k)}}\right) \tag{297}$$

$$\xrightarrow[D,N\to\infty]{a.s.} -\sum_{k=1}^{K}\int_{\boldsymbol{x}\in\mathcal{M}}\log\left(\int_{\boldsymbol{z}\in\mathcal{M}}\exp\left(-\frac{\sum_{s=1}^{K}\|\boldsymbol{x}^{(s)}-\boldsymbol{z}^{(s)}\|^2 p_s^{(k)}}{\epsilon\sum_{t=1}^{K}p_t^{(k)}\beta_t} - \frac{\left\|\boldsymbol{x}_i^{(K+1)}-\boldsymbol{z}^{(K+1)}\right\|^2\sum_{s=1}^{K}p_s^{(k)}}{\epsilon\sum_{t=1}^{K}p_t^{(k)}\beta_t}\right)\right. \tag{298}$$

$$\left.f(\boldsymbol{z})d\boldsymbol{z}\right)f(\boldsymbol{x})d\boldsymbol{x}$$

$$= \sum_{k=1}^{K}\frac{\dim(\mathcal{M}_{K+1})}{2}\log\left(\frac{\sum_{s=1}^{K}p_s^{(k)}}{\sum_{t=1}^{K}p_t^{(k)}\beta_t}\right) - K\frac{\dim(\mathcal{M}_{K+1})}{2}\log(\pi\epsilon) \tag{299}$$

$$+ \sum_{k,s=1}^{K}\frac{\dim(\mathcal{M}_s)}{2}\log\left(\frac{p_s^{(k)}}{\sum_{t=1}^{K}p_t^{(k)}\beta_t}\right) - K\sum_{s=1}^{K}\frac{\dim(\mathcal{M}_s)}{2}\log(\pi\epsilon) + Kh(f)\log +O(\sqrt{\epsilon})$$

where we employ Lemmas I.6 and I.7 to derive the asymptotic convergence in the next lines. The next derivation results from Lemma I.9. The latter lemma indicate that there exists $\tilde{\epsilon}(\mathcal{M},f) < 1$ such that for any $\epsilon < \tilde{\epsilon}^2(\mathcal{M},f)$ and any $p_s^{(k)} \in [\sqrt{\epsilon}, 1-(K-1)\sqrt{\epsilon}]$ in which the approximation holds, where $s,k=1,\ldots,K$. Hence, we get

$\square$

**Lemma I.10.** *Let $C_1, C_2, C_3 > 0$, and $\beta_1, \beta_2 \in (0,1)$ where $\beta_1 + \beta_2 = 1$. Define the function $f : (0,1)^2 \to \mathbb{R}$ by*

$$f(p_1, p_2) = C_1\log\left(\frac{p_1(1-p_1)}{(p_1\beta_1+p_2\beta_2)(1-p_1\beta_1-p_2\beta_2)}\right) \tag{300}$$

$$+ C_2\log\left(\frac{p_2(1-p_2)}{(p_1\beta_1+p_2\beta_2)(1-p_1\beta_1-p_2\beta_2)}\right)$$

$$+ C_3\log\left(\frac{(p_1+p_2)(2-p_1-p_2)}{(p_1\beta_1+p_2\beta_2)(1-p_1\beta_1-p_2\beta_2)}\right).$$

*Let $\alpha \geq \max_{t\in\{0,1\}} 2\cdot(\beta_2(1-\beta_2))^{-(C_1+C_2+C_3)/(C_t)}$ be a neighborhood constant (and therefore $\alpha \geq 2$). Then, for any $\delta \in (0, \min(\beta_1, \beta_2)^{(C_1+C_2+C_3)/(C_1+C_2)}/\alpha)$ the minimizers $p_1^*, p_2^*$ defined by*

$$p_1^*, p_2^* = \underset{(p_1,p_2)\in[\delta,1-\delta]^2}{\arg\min} f(p_1, p_2) \tag{301}$$

*should sastisfy $(p_1^*, p_2^*) \in L \times U$ or $(p_1^*, p_2^*) \in U \times L$. where $L = [\delta, \alpha\delta]$ and $U = [1-\alpha\delta, 1-\delta]$.*

*Proof.* The proof will begin by determining the characteristics of a function that will be used throughout the proof. Then, we are going to divide the domain into different parts and prove that the value at either $(\delta, 1 - \delta)$ or $(1 - \delta, \delta)$ will attain a lower value.

We begin with going through the properties of $\rho : [0, 1] \rightarrow \mathbb{R}$ defined by $\rho(p) = p(1 - p)$ for any $p \in [0, 1]$. The function is concave, as indicated by the negativity of its second order derivative ( see Theorem 4.5 in (Rockafellar, 1970)). It attains its maximum at $p = 1/2$, where its first derivative vanishes — a direct application of Theorem 25.1 (Rockafellar, 1970). Moreover, $\rho$ is symmetric around $p = 1/2$. These properties are illustrated below:

$$\frac{d^2}{dp^2}\rho(p) \;=\; \frac{d}{dp}1 - 2p = -2 \tag{302}$$

$$0 \;=\; \frac{d}{dp}\rho(p) = 1 - 2p \tag{303}$$

$$f(p) \;=\; p(1 - p) = (1 - p)p = f(1 - p). \tag{304}$$

For the completeness of the proof we begin by showing that $\alpha \geq 2$, by showing that for any $t \in \{1, 2\}$

$$\alpha \;\geq\; \frac{2}{(\beta_2(1 - \beta_2))^{(C_1 + C_2 + C_3)/(C_t)}} \tag{305}$$

$$\geq\; \frac{2}{((1/2)(1 - 1/2))^{(C_1 + C_2 + C_3)/(C_t)}} \tag{306}$$

$$=\; 2 \cdot 4^{(C_1 + C_2 + C_3)/(C_t)} \tag{307}$$

$$\geq\; 2, \tag{308}$$

where we use the fact that $C_1, C_2, C_3 > 0$ and that $\rho$ is maximized at $1/2$.

Now, we can begin characterizing the minimal values of $f$ by examining its values for all $(p_1, p_2) \in [\alpha\delta, 1 - \alpha\delta] \times [\delta, 1 - \delta]$. Specifically, we will show that the first two terms of $f$ are higher than their value at $(p_1, p_2) = (\delta, 1 - \delta)$ by

$$\sum_{i=1}^{2} C_i \log\left(\frac{p_i(1 - p_i)}{(p_1\beta_1 + p_2\beta_2)(1 - p_1\beta_1 - p_2\beta_2)}\right) - \sum_{i=1}^{2} C_i \log\left(\frac{\delta(1 - \delta)}{(\delta\beta_1 + (1 - \delta)\beta_2)(1 - \delta\beta_1 - (1 - \delta)\beta_2)}\right) \tag{309}$$

$$= \sum_{i=1}^{2} C_i \log\left(\frac{p_i(1 - p_i)}{\delta(1 - \delta)}\right) + \sum_{i=1}^{2} C_i \log\left(\frac{(\delta\beta_1 + (1 - \delta)\beta_2)(1 - \delta\beta_1 - (1 - \delta)\beta_2)}{(p_1 + \beta_2(p_2 - p_1))(1 - p_1 - \beta_2(p_2 - p_1))}\right) \tag{310}$$

$$\geq C_1 \log\left(\frac{\alpha\delta(1 - \alpha\delta)}{\delta(1 - \delta)}\right) + C_2 \log\left(\frac{\delta(1 - \delta)}{\delta(1 - \delta)}\right) + \sum_{i=1}^{2} C_i \log\left(\frac{(\delta\beta_1 + (1 - \delta)\beta_2)(1 - \delta\beta_1 - (1 - \delta)\beta_2)}{1/4}\right) \tag{311}$$

$$\geq C_1 \log\left(\frac{\alpha\delta(1 - \alpha\delta)}{\delta(1 - \delta)}\right) + C_2 \log\left(\frac{\delta(1 - \delta)}{\delta(1 - \delta)}\right) + \sum_{i=1}^{2} C_i \log\left(\frac{\beta_2(1 - \beta_2)}{1/4}\right) \tag{312}$$

where in the first inequality we used the concavity assumption of $\rho$ and that is maximized at $1/2$, making $\rho(1/2) = 1/4$. In the second inequality we used the fact that $\delta\beta_1 + (1 - \delta)\beta_2 \in [\min(\beta_1, \beta_2), \max(\beta_1, \beta_2)]$ as the term is a convex combination of $\beta_1$ and $\beta_2$. As $\beta_1 = 1 - \beta_2$, we can see that $[\min(\beta_1, \beta_2), \max(\beta_1, \beta_2)] = [\min(1 - \beta_2, \beta_2), \max(1 - \beta_2, \beta_2)]$. Therefore, by using the concavity assumption of $\rho$ we can lower bound the numerator within the last logarithm term. Below, we continue the derivation

$$\geq\; C_1 \log\left(\frac{\alpha(1 - \alpha\delta)}{1}\right) + \sum_{i=1}^{2} C_i \log\left(4\beta_2(1 - \beta_2)\right) \tag{313}$$

$$\geq\; C_1 \log\left(\alpha/2\right) + \sum_{i=1}^{2} C_i \log\left(4\beta_2(1 - \beta_2)\right) \tag{314}$$

$$\geq\; C_1 \log\left(\alpha \cdot (1/2)\right) + \sum_{i=1}^{2} C_i \log\left(\beta_2(1 - \beta_2)\right) \tag{315}$$

where in the first inequality we used $\log(1/(1 - \delta)) \geq \log(1/1) = 0$, which follows from $\delta \leq \min(\beta_1, \beta_2)^{(C_1+C_2+C_3)/(C_1+C_2)}/\alpha \leq \min(\beta_1, \beta_2)/1 \leq 1/2$. The second derivation is based on $1 - \alpha\delta \geq 1 - \min(\beta_1, \beta_2)^{(C_1+C_2+C_3)/(C_1+C_2)} \geq 1 - \min(\beta_1, \beta_2) \geq 1/2$ from the domain of $\delta$ and that $\beta_1 + \beta_2 = 1$.

As for the third terms, we will show that last term of $f$ is higher than its value at $(p_1, p_2) = (\delta, 1 - \delta)$ by

$$C_3 \log \left( \frac{(p_1 + p_2)(2 - p_1 - p_2)}{(p_1\beta_1 + p_2\beta_2)(1 - p_1\beta_1 - p_2\beta_2)} \right) - C_3 \log \left( \frac{(\delta + 1 - \delta)(2 - \delta - (1 - \delta))}{(\delta\beta_1 + (1 - \delta)\beta_2)(1 - \delta\beta_1 - (1 - \delta)\beta_2)} \right) \tag{316}$$

$$= C_3 \log \left( \frac{p_1 + p_2}{p_1\beta_1 + p_2\beta_2} \right) + C_3 \log \left( \frac{2 - p_1 - p_2}{1 - p_1\beta_1 - p_2\beta_2} \right) - C_3 \log \left( \frac{1}{(\delta\beta_1 + (1 - \delta)\beta_2)(1 - \delta\beta_1 - (1 - \delta)\beta_2)} \right) \tag{317}$$

$$= C_3 \log \left( \frac{p_1 + p_2}{p_1\beta_1 + p_2\beta_2} \right) + C_3 \log \left( \frac{2 - p_1 - p_2}{1 - p_1\beta_1 - p_2\beta_2} \right) + C_3 \log \left( (\delta\beta_1 + (1 - \delta)\beta_2)(1 - \delta\beta_1 - (1 - \delta)\beta_2) \right) \tag{318}$$

$$\geq C_3 \log \left( \frac{p_1 + p_2}{p_1\beta_1 + p_2\beta_2} \right) + C_3 \log \left( \frac{2 - p_1 - p_2}{1 - p_1\beta_1 - p_2\beta_2} \right) + C_3 \log \left( \beta_2(1 - \beta_2) \right) \tag{319}$$

$$\geq C_3 \log \left( \frac{p_1\beta_1 + p_2\beta_2}{p_1\beta_1 + p_2\beta_2} \right) + C_3 \log \left( \frac{1 - p_1 + 1 - p_2}{1 - \min(p_1, p_2)} \right) + C_3 \log \left( \beta_2(1 - \beta_2) \right) \tag{320}$$

$$\geq C_3 \log \left( \beta_2(1 - \beta_2) \right). \tag{321}$$

where in the first inequality we use the same derivation made in (312), and in the second we use the fact that $\beta_1 = 1 - \beta_2 \in (0, 1)$ and that $p_1\beta_1 + p_2\beta_2 \geq \min(p_1, p_2)$ which follows from it. Finally, in the last inequality, we use the fact that $p_1, p_2 \in (0, 1)$.

By combining these two statements we get that $f(p_1, p_2) \geq f(\delta, 1 - \delta)$ by

$$f(p_1, p_2) - f(\delta, 1 - \delta) \geq C_1 \log(\alpha/2) + \sum_{i=1}^{3} C_i \log(\beta_2(1 - \beta_2)) \tag{322}$$

$$= C_1 \log \left( \alpha \cdot \frac{(\beta_2(1 - \beta_2))^{\sum_{i=1}^{3} C_i/C_t}}{2} \right) \tag{323}$$

$$\geq C_1 \log \left( \max_{t \in \{1,2\}} \frac{2}{(\beta_2(1 - \beta_2))^{\sum_{i=1}^{3} C_i/C_t}} \cdot \frac{(\beta_2(1 - \beta_2))^{\sum_{i=1}^{3} C_i/C_1}}{2} \right) \tag{324}$$

$$\geq C_1 \log \left( \frac{2}{(\beta_2(1 - \beta_2))^{\sum_{i=1}^{3} C_i/C_1}} \cdot \frac{(\beta_2(1 - \beta_2))^{\sum_{i=1}^{3} C_i/C_1}}{2} \right) \tag{325}$$

$$\geq 0. \tag{326}$$

By following the same steps above and switching between $p_1$ and $p_2$, we can get that $f$ attains a lower value at $(p_1, p_2) = (1 - \delta, \delta)$ compared to it value in all the points within the domain $(p_1, p_2) \in [\delta, 1 - \delta] \times [\alpha\delta, 1 - \alpha\delta]$.

Finally, we are left with showing that $f$ attains a lower value at $(p_1, p_2) = (\delta, 1 - \delta)$ compared to any value $(p_1, p_2) \in [\delta, \alpha\delta]^2 \cup [1 - \alpha\delta, 1 - \delta]^2$. As before, we begin by bounding the difference between the first two terms by

$$\sum_{i=1}^{2} C_i \log \left( \frac{p_i(1 - p_i)}{(p_1\beta_1 + p_2\beta_2)(1 - p_1\beta_1 - p_2\beta_2)} \right) - \sum_{i=1}^{2} C_i \log \left( \frac{\delta(1 - \delta)}{(\delta\beta_1 + (1 - \delta)\beta_2)(1 - \delta\beta_1 - (1 - \delta)\beta_2)} \right) \tag{327}$$

$$= \sum_{i=1}^{2} C_i \log \left( \frac{p_i(1 - p_i)}{\delta(1 - \delta)} \right) + \sum_{i=1}^{2} C_i \log \left( \frac{(\delta\beta_1 + (1 - \delta)\beta_2)(1 - \delta\beta_1 - (1 - \delta)\beta_2)}{(p_1\beta_1 + p_2\beta_2)(1 - p_1\beta_1 - p_2\beta_2)} \right) \tag{328}$$

$$\geq \sum_{i=1}^{2} C_i \log \left( \frac{\delta(1 - \delta)}{\delta(1 - \delta)} \right) + \sum_{i=1}^{2} C_i \log \left( \frac{(\delta\beta_1 + (1 - \delta)\beta_2)(1 - \delta\beta_1 - (1 - \delta)\beta_2)}{(p_1\beta_1 + p_2\beta_2)(1 - p_1\beta_1 - p_2\beta_2)} \right) \tag{329}$$

$$\geq \sum_{i=1}^{2} C_i \log \left( \frac{\delta(1 - \delta)}{\delta(1 - \delta)} \right) + \sum_{i=1}^{2} C_i \log \left( \frac{(\delta\beta_1 + (1 - \delta)\beta_2)(1 - \delta\beta_1 - (1 - \delta)\beta_2)}{\alpha\delta(1 - \alpha\delta)} \right) \tag{330}$$

$$\geq \sum_{i=1}^{2} C_i \log\left(\frac{\beta_2(1-\beta_2)}{\alpha\delta(1-\alpha\delta)}\right) \tag{331}$$

where in the first inequality we use the concavity property of $\rho$, and in the second we use its properties along with the fact that $p_1\beta + p_2\beta \in [\delta,\alpha\delta]^2 \cup [1-\alpha\delta, 1-\delta]^2$. In the third inequality, we use the fact that $\delta\beta_1 + (1-\delta)\beta_2 \in [\min(\beta_1,\beta_2), \max(\beta_1,\beta_2)]$ as the term is a convex combination of $\beta_1$ and $\beta_2$. As $\beta_1 = 1 - \beta_2$, we can see that $[\min(\beta_1,\beta_2), \max(\beta_1,\beta_2)] = [\min(1-\beta_2,\beta_2), \max(1-\beta_2,\beta_2)]$.

As for the third term, we will derive their differences below.

$$C_3 \log\left(\frac{(p_1+p_2)(2-p_1-p_2)}{(p_1\beta_1+p_2\beta_2)(1-p_1\beta_1-p_2\beta_2)}\right) - C_3 \log\left(\frac{(\delta+1-\delta)(2-\delta-(1-\delta))}{(\delta\beta_1+(1-\delta)\beta_2)(1-\delta\beta_1-(1-\delta)\beta_2)}\right) \tag{332}$$

$$\geq C_3 \log(\beta_2(1-\beta_2)), \tag{333}$$

by going through the same steps taken in (316)-(321). Now, we can combine the last two results to show that over this domain $f(p_1,p_2) \geq f(\delta, 1-\delta)$

$$f(p_1,p_2) - f(\delta, 1-\delta) \;\geq\; \sum_{i=1}^{2} C_i \log\left(\frac{\beta_2(1-\beta_2)}{\alpha\delta(1-\alpha\delta)}\right) + C_3 \log(\beta_2(1-\beta_2)) \tag{334}$$

$$=\; \log\left(\frac{\beta_2(1-\beta_2)^{(C_1+C_2+C_3)}}{(\alpha\delta(1-\alpha\delta))^{C_1+C_2}}\right) \tag{335}$$

$$\geq\; \log\left(\frac{\beta_2(1-\beta_2)^{(C_1+C_2+C_3)}}{\left((\beta_2(1-\beta_2))^{\frac{(C_1+C_2+C_3)}{C_1+C_2}}\right)^{C_1+C_2)}}\right) \tag{336}$$

$$=\; \log\left(\frac{\beta_2(1-\beta_2)^{(C_1+C_2+C_3)}}{\beta_2(1-\beta_2)^{(C_1+C_2+C_3)}}\right) \tag{337}$$

$$=\; 0 \tag{338}$$

by using the concavity property of $\rho$ and that it maximizes at $1/2$, along with the definition of $\delta$ in which $\delta\alpha \leq \min(\beta_1,\beta_2)^{(C_1+C_2+C_3)/(C_1+C_2)} \leq 1/2$, as $\min(\beta_1,\beta_2) \leq 1/2$. $\qquad\square$

**Theorem 4.4** Let $K = 2$, and define $f : (0,1)^2 \to \mathbb{R}$ by

$$f(p_1,p_2) = \sum_{k=1}^{K} \frac{\dim(\mathcal{M}_{K+1})}{2} \log\left(\frac{\sum_{s=1}^{K} p_s^{(k)}}{\sum_{t=1}^{K} p_t^{(k)}\beta_t}\right) + \sum_{k,s=1}^{K} \frac{\dim(\mathcal{M}_s)}{2} \log\left(\frac{p_s^{(k)}}{\sum_{t=1}^{K} p_t^{(k)}\beta_t}\right), \tag{18}$$

where $p_1^{(1)} = p_1$ and $p_2^{(1)} = p_2$ and therefore $p_1^{(2)} = 1 - p_1^{(1)}$, $p_2^{(2)} = 1 - p_2^{(2)}$. Then, the limiting minimizer $(p_1^*, p_2^*) = \lim_{\epsilon\to 0} \arg\min_{p_1,p_2\in[\sqrt{\epsilon}, 1-\sqrt{\epsilon}]^2} f(p_1,p_2)$ is either $(0,1)$ or $(1,0)$.

*Proof of Theorem 4.4.* We begin by rewriting (18) by

$$f(p_1,p_2) \tag{339}$$

$$= \sum_{k,s=1}^{2} \frac{\dim(\mathcal{M}_s)}{2} \log\left(\frac{p_s^{(k)}}{\sum_{t=1}^{2} p_t^{(k)}\beta_k}\right) + \sum_{k=1}^{2} \frac{\dim(\mathcal{M}_3)}{2} \log\left(\frac{\sum_{s=1}^{2} p_s^{(k)}}{\sum_{t=1}^{2} p_t^{(k)}\beta_t}\right) \tag{340}$$

$$= \sum_{s=1}^{2} \frac{\dim(\mathcal{M}_s)}{2} \log\left(\frac{p_s^{(1)}}{\sum_{t=1}^{2} p_t^{(1)}\beta_t} \cdot \frac{p_s^{(2)}}{\sum_{t=1}^{2} p_t^{(2)}\beta_t}\right) + \frac{\dim(\mathcal{M}_3)}{2} \log\left(\frac{\sum_{s=1}^{2} p_s^{(1)}}{\sum_{t=1}^{2} p_t^{(1)}\beta_t} \cdot \frac{\sum_{s=1}^{2} p_s^{(2)}}{\sum_{t=1}^{2} p_t^{(2)}\beta_t}\right) \tag{341}$$

$$= \sum_{s=1}^{2} \frac{\dim(\mathcal{M}_s)}{2} \log\left(\frac{p_s^{(1)}}{\sum_{t=1}^{2} p_t^{(1)}\beta_t} \cdot \frac{1-p_s^{(1)}}{\sum_{t=1}^{2}(1-p_t^{(1)})\beta_t}\right) \tag{342}$$

$$+ \frac{\dim(\mathcal{M}_3)}{2} \log \left( \frac{\sum_{s=1}^{2} p_s^{(1)}}{\sum_{t=1}^{2} p_t^{(1)} \beta_t} \cdot \frac{\sum_{s=1}^{2}(1 - p_s^{(1)})}{\sum_{t=1}^{2}(1 - p_t^{(1)}) \beta_t} \right) \tag{343}$$

$$= \sum_{s=1}^{2} \frac{\dim(\mathcal{M}_s)}{2} \log \left( \frac{p_s^{(1)}}{\sum_{t=1}^{2} p_t^{(1)} \beta_t} \frac{1 - p_s^{(1)}}{1 - \sum_{t=1}^{2} p_t^{(1)} \beta_t} \right) + \frac{\dim(\mathcal{M}_3)}{2} \log \left( \frac{\sum_{s=1}^{2} p_s^{(1)}}{\sum_{t=1}^{2} p_t^{(1)} \beta_t} \cdot \frac{2 - \sum_{s=1}^{2} p_s^{(1)}}{1 - \sum_{t=1}^{2} p_t^{(1)} \beta_t} \right) \tag{344}$$

$$= \sum_{s=1}^{2} \frac{\dim(\mathcal{M}_s)}{2} \log \left( \frac{p_s(1 - p_s)}{(\sum_{t=1}^{2} p_t \beta_t)(1 - \sum_{t=1}^{2} p_t \beta_t)} \right) + \frac{\dim(\mathcal{M}_3)}{2} \log \left( \frac{(\sum_{s=1}^{2} p_s)(2 - \sum_{s=1}^{2} p_s)}{(\sum_{t=1}^{2} p_t \beta_t)(1 - \sum_{t=1}^{2} p_t \beta_t)} \right), \tag{345}$$

where in the second derivation we use $p_s^{(2)} = 1 - p_s^{(1)}$ for $s = 1, 2$, as defined in the theorem's statement. Then, we use $\sum_{t=1}^{2} \beta_t = 1$ as defined in Section 4, and substitue $p_s^{(1)} = p_s$ as defined in the theorem's statment.

We observe that the function $f(p_1, p_2)$ matches the form analyzed in Lemma I.10. Therefore, there exists a constant $\alpha \geq 2$, dependent on $\beta_1, \beta_2, \mathcal{M}_1, \mathcal{M}_2, \mathcal{M}_3$, such that for any sufficiently small $\delta > 0$, the minimizers of $f$ over the domain $[\delta, 1 - \delta]^2$ lie within either $L \times U$ or $U \times L$, where $L = [\delta, \alpha\delta]$ and $U = [1 - \alpha\delta, 1 - \delta]$. This implies that, for any sequence of minimization problems with $\delta$s that tends to 0, the corresponding minimizers converge to either $(0, 1)$ or $(1, 0)$. By setting $\delta = \sqrt{\epsilon}$ and similarly considering a sequence of problems with $\epsilon$s that tends to 0 yields the desired result stated in the theorem. $\square$

## I.3. Proofs of Appendix C

**Lemma I.11.** *Let $\{\boldsymbol{W}^{(k)}\} \subset [0,1]^{N \times N}$ be affinity matrices under the constraints in Problem C.1. Define a partitioning solution $\{\boldsymbol{\omega}^{(k)}\} \subset [0,1]^D$ by*

$$\omega_d^{(k)} = \frac{\exp\left(-\sum_{i,j=1}^N W^{(k)}\left((\boldsymbol{y}_i)_d - (\boldsymbol{y}_j)_d\right)^2 / \delta\right)}{\sum_{\tilde{k}=1}^K \exp\left(-\sum_{i,j=1}^N W^{(\tilde{k})}\left((\boldsymbol{y}_i)_d - (\boldsymbol{y}_j)_d\right)^2 / \delta\right)} \qquad k = 1, \ldots, K. \tag{346}$$

*Then, a minimizer of the optimization problem suggested in Problem C.1 among partitioning solutions that satisfy its constraints is*

$$\boldsymbol{\omega}^{(k)*} = \underset{\{\tilde{\boldsymbol{\omega}}^{(k)}\}}{\arg\min}\ G_{reg}(\delta, \{\boldsymbol{W}^{(k)}\}_{k=1}^K, \{\tilde{\boldsymbol{\omega}}^{(k)}\}_{k=1}^K, \{\boldsymbol{y}_i\}_{i=1}^N) \tag{347}$$

*Proof.* The proof will demonstrate that the optimal solution can be derived from solving $D$ subproblems, each corresponding to a coordinate of partitioning solution. We will then use convexity conditions to derive the optimal solution under certain assumptions on the data. Finally, we will show that if these assumptions do not hold, the solution takes a similar form, but with bandwidth parameters approaching zero.

We begin by showing that the problem can be solved using $D$ subproblems separately. For each $d \in \{1, \ldots, D\}$ we define a subproblem related to the $d - th$ coordinate by

$$\underset{\tilde{\omega}_d^{(1)}, \ldots, \tilde{\omega}_d^{(K)}}{\arg\min}\ \tilde{\tilde{G}}_d(\{\boldsymbol{W}^{(k)}\}_{k=1}^K, \{\tilde{\omega}_d^{(k)}\}_{k=1}^K, \{\boldsymbol{y}_j\}_{j=1}^N) \tag{348}$$

subject to the constraints $\sum_{d=1}^D \tilde{\omega}_d^{(k)} = 1$, where

$$\tilde{\tilde{G}}_d(\{\boldsymbol{W}^{(k)}\}_{k=1}^K, \{\tilde{\omega}_d^{(k)}\}_{k=1}^K, \{\boldsymbol{y}_j\}_{j=1}^N) = \sum_{k=1}^K \tilde{\omega}_d^{(k)} \left(\sum_{i=1}^N \sum_{j=1}^N W_{i,j}^{(k)}((\boldsymbol{y}_i)_d - (\boldsymbol{y}_j)_d)^2\right) \tag{349}$$

$$+ \delta \sum_{k=1}^K \tilde{\omega}_d^{(k)} \log\left(\tilde{\omega}_d^{(k)}\right). \tag{350}$$

Based on the next derivation, we can see that the optimal solution should be optimal for each of these sub-problems

$$\underset{\{\tilde{\boldsymbol{\omega}}^{(k)}\}_{k=1}^K}{\min}\ G_{reg}(\delta, \{\tilde{\boldsymbol{W}}^{(k)}\}_{k=1}^K, \{\tilde{\boldsymbol{\omega}}^{(k)}\}_{k=1}^K, \{\boldsymbol{y}_i\}_{i=1}^N) \tag{351}$$

$$= \underset{\{\tilde{\boldsymbol{\omega}}^{(k)}\}_{k=1}^K}{\min}\ G(\{\tilde{\boldsymbol{W}}^{(k)}\}, \{\tilde{\boldsymbol{\omega}}^{(k)}\}, \{\boldsymbol{y}_i\}_{i=1}^N) + \delta\left(D\log(K) + \sum_{d=1}^D \sum_{k=1}^K \tilde{\omega}_d^{(k)} \log(\tilde{\omega}_d^{(k)})\right) \tag{352}$$

$$= \underset{\{\tilde{\boldsymbol{\omega}}^{(k)}\}_{k=1}^K}{\min}\ \sum_{k=1}^K \sum_{i,j=1}^N \tilde{W}_{i,j}^{(k)} \|\boldsymbol{y}_i - \boldsymbol{y}_j\|_{\boldsymbol{\omega}^{(k)}}^2 + \delta\left(D\log(K) + \sum_{d=1}^D \sum_{k=1}^K \tilde{\omega}_d^{(k)} \log(\tilde{\omega}_d^{(k)})\right) \tag{353}$$

$$= \underset{\{\tilde{\boldsymbol{\omega}}^{(k)}\}_{k=1}^K}{\min}\ \sum_{d=1}^D \sum_{k=1}^K \tilde{\omega}_d^{(k)} \left(\sum_{i=1}^N \sum_{j=1}^N W_{i,j}^{(k)}((\boldsymbol{y}_i)_d - (\boldsymbol{y}_j)_d)^2\right) + \delta \sum_{d=1}^D \sum_{k=1}^K \tilde{\omega}_d^{(k)} \log\left(\tilde{\omega}_d^{(k)}\right) \tag{354}$$

$$= \sum_{d=1}^D \underset{\{\tilde{\omega}_d^{(k)}\}_{k=1}^K}{\min}\ \sum_{k=1}^K \tilde{\omega}_d^{(k)} \left(\sum_{i=1}^N \sum_{j=1}^N W_{i,j}^{(k)}((\boldsymbol{y}_i)_d - (\boldsymbol{y}_j)_d)^2\right) + \delta \sum_{d=1}^D \sum_{k=1}^K \tilde{\omega}_d^{(k)} \log\left(\tilde{\omega}_d^{(k)}\right) \tag{355}$$

$$= \sum_{d=1}^D \underset{\{\tilde{\omega}_d^{(k)}\}_{k=1}^K}{\min}\ \sum_{k=1}^K \tilde{\tilde{G}}_d(\{\boldsymbol{W}^{(k)}\}_{k=1}^K, \{\tilde{\omega}_d^{(k)}\}_{k=1}^K, \{\boldsymbol{y}_j\}_{j=1}^N) \tag{356}$$

where the domain of each minimization problem above contains the constraints shown in Problem C.1.

Without loss of generality, we are going to find the optimal $\omega_d^{(1)}, \ldots, \omega_d^{(K)}$ that minimize the $\tilde{\tilde{G}}_d$ for some $d \in \{1, \ldots, D\}$. To find the optimal solution, we begin by building on the Karush-Kuhn-Tucker (KKT) Theorem (Corollary 28.3.1 in (Rockafellar, 1970)). We note that this solution will depend on certain conditions; therefore, after this derivation, we will provide an alternative solution that attains a similar form when these conditions are not met.

The theorem assumes that the objective is convex and that the equality constraint is an affine function. Furthermore, it assumes that there exists a solution within the domain that satisfies the inequality constraints in a strict manner (Slater's conditions). We begin with the latter assumption, as there are no inequality constraints the Slater's conditions are satisfied.

The objective $\tilde{\tilde{G}}_d$ is a sum of a linear function and an entropy function. Based on Theorem 4.5 in (Rockafellar, 1970), by showing that the Hessian of the entropy function is positive semi-definite we can deduce that it is convex. Specifically, the Hessian elements are

$$\frac{d^2}{d^2\tilde{\omega}_d^{(k)}} \delta \sum_{k=1}^{K} \tilde{\omega}_d^{(k)} \log\left(\tilde{\omega}_d^{(k)}\right) = \frac{d}{d\tilde{\omega}_d^{(k)}} \delta(1 + \log(\tilde{\omega}_d^{(k)})) \tag{357}$$

$$= \frac{\delta}{\tilde{\omega}_d^{(k)}} \tag{358}$$

$$\geq 0. \tag{359}$$

$$\frac{d}{d\tilde{\omega}_d^{(s)}} \frac{d}{d\tilde{\omega}_d^{(k)}} \delta \sum_{k=1}^{K} \tilde{\omega}_d^{(k)} \log\left(\tilde{\omega}_d^{(k)}\right) = \frac{d}{d\tilde{\omega}_d^{(s)}} \delta(1 + \log(\tilde{\omega}_d^{(k)})) \tag{360}$$

$$= 0 \tag{361}$$

for all $s, k \in \{1, \ldots, K\}$ where $s \neq k$. As the Hessian is a diagonal matrix with positive values we can conclude that it is a positive definite matrix.

Now, as the objective is the sum of two convex functions it is convex as well based on Theorem 5.2 in (Rockafellar, 1970) and the convexity of linear functions.

The KKT theorem states that a solution satisfying the KKT conditions—including stationarity, primal feasibility, dual feasibility, and complementary slackness—is an optimal solution to the problem, provided Slater's condition holds. To derive such a solution we need to first define the Lagrangian of the minimization problem by

$$\tilde{L}_d(\boldsymbol{W}^{(k)}, \tilde{\omega}_d^{(k)}) \equiv \sum_{k=1}^{K} \tilde{\omega}_d^{(k)} \left(\sum_{i=1}^{N} \sum_{j=1}^{N} W_{i,j}^{(k)} ((\boldsymbol{y}_i)_d - (\boldsymbol{y}_j)_d)^2\right)$$

$$+ \delta \sum_{k=1}^{K} \tilde{\omega}_d^{(k)} \log\left(\tilde{\omega}_d^{(k)}\right)$$

$$+ \mu_d(\sum_{\tilde{k}=1}^{K} \tilde{\omega}_d^{(\tilde{k})} - 1) \tag{362}$$

where $\mu_d \in \mathbb{R}$.

A solution that satisfies the stationary condition should attain

$$0 = \frac{d\tilde{L}_d}{d\omega_d^{(k)}} \tag{363}$$

$$= \sum_{i=1}^{N} \sum_{j=1}^{N} W_{i,j}^{(k)} ((\boldsymbol{y}_i)_d - (\boldsymbol{y}_j)_d)^2 + \delta(1 + \log(\omega_d^{(k)})) + \mu_d \tag{364}$$

$$\omega_d^{(k)} = \exp\left(-\frac{\sum_{i,j=1}^{N} W_{i,j}^{(k)} ((\boldsymbol{y}_i)_d - (\boldsymbol{y}_j)_d)^2 + \delta + \mu_d}{\delta}\right). \tag{365}$$

The primal feasibility condition on the on the equality constraint induces-

$$1 \;=\; \sum_{k=1}^{K} \omega_d^{(k)} \tag{366}$$

$$=\; \sum_{k=1}^{K} \exp\left(-\frac{\sum_{i=1}^{N}\sum_{j=1}^{N} W_{i,j}^{(k)}((\boldsymbol{y}_i)_d - (\boldsymbol{y}_j)_d)^2 + \delta + \mu_d}{\delta}\right) \tag{367}$$

$$\exp\left(\frac{\mu_d}{\delta}\right) \;=\; \sum_{k=1}^{K} \exp\left(-\frac{\sum_{i=1}^{N}\sum_{j=1}^{N} W_{i,j}^{(k)}((\boldsymbol{y}_i)_d - (\boldsymbol{y}_j)_d)^2 + \delta}{\delta}\right) \tag{368}$$

$$\mu_d \;=\; \delta \cdot \log\left(\sum_{k=1}^{K} \exp\left(-\frac{\sum_{i=1}^{N}\sum_{j=1}^{N} W_{i,j}^{(k)}((\boldsymbol{y}_i)_d - (\boldsymbol{y}_j)_d)^2 + \delta}{\delta}\right)\right). \tag{369}$$

By pushing it back into (365) we get that for any $k \in \{1, \ldots, N\}$

$$\omega_d^{(k)} \;=\; \frac{\exp\left(-\frac{\sum_{i=1}^{N}\sum_{j=1}^{N} W_{i,j}^{(k)}((\boldsymbol{y}_i)_d - (\boldsymbol{y}_j)_d)^2 + \delta}{\delta}\right)}{\sum_{k=1}^{K} \exp\left(-\frac{\sum_{i=1}^{N}\sum_{j=1}^{N} W_{i,j}^{(k)}((\boldsymbol{y}_i)_d - (\boldsymbol{y}_j)_d)^2 + \delta}{\delta}\right)} \tag{370}$$

$$=\; \frac{\exp\left(-\frac{\sum_{i=1}^{N}\sum_{j=1}^{N} W_{i,j}^{(k)}((\boldsymbol{y}_i)_d - (\boldsymbol{y}_j)_d)^2}{\delta}\right)}{\sum_{k=1}^{K} \exp\left(-\frac{\sum_{i=1}^{N}\sum_{j=1}^{N} W_{i,j}^{(k)}((\boldsymbol{y}_i)_d - (\boldsymbol{y}_j)_d)^2}{\delta}\right)}. \tag{371}$$

Therefore we can conclude the proof with the derived optimal solution form. $\qquad\square$

**Proposition C.2** Let $\delta \geq 0$, $\{\boldsymbol{\omega}^{(k)}\}_{k=1}^{K} \subset [0,1]^D$ be a partitioning weights and $\{\boldsymbol{W}^{(k)}\}_{k=1}^{K}, \subset [0,1]^{N \times N}$ be affinity matrices that satisfy the constraints of Problem C.1.

Define $\{\boldsymbol{W}^{(k)*}\} \subset [0,1]^{N \times N}$ as in (9) based on $\{\boldsymbol{\omega}^{(k)}\}_{k=1}^{K}$, and $\{\boldsymbol{\omega}^{(k)*}\}_{k=1}^{K}$ by

$$\omega_d^{(k)*} = \exp\left(-\frac{\sum_{i,j} W_{i,j}^{(k)}\left((\boldsymbol{y}_i)_d - (\boldsymbol{y}_j)_d\right)^2}{\delta}\right) \Big/ \sum_{s=1}^{K} \exp\left(-\frac{\sum_{i,j} W_{i,j}^{(s)}\left((\boldsymbol{y}_i)_d - (\boldsymbol{y}_j)_d\right)^2}{\delta}\right). \tag{21}$$

Then, we have that

$$\{\boldsymbol{W}^{(k)*}\} = \underset{\{\tilde{\boldsymbol{W}}^{(k)}\}}{\arg\min}\, G_{reg}(\delta, \{\tilde{\boldsymbol{W}}^{(k)}\}_{k=1}^{K}, \{\boldsymbol{\omega}^{(k)}\}_{k=1}^{K}, \{\boldsymbol{y}_i\}_{i=1}^{N}), \tag{22}$$

$$\{\boldsymbol{\omega}^{(k)*}\} = \underset{\{\tilde{\boldsymbol{\omega}}^{(k)}\}}{\arg\min}\, G_{reg}(\delta, \{\boldsymbol{W}^{(k)}\}_{k=1}^{K}, \{\tilde{\boldsymbol{\omega}}^{(k)}\}_{k=1}^{K}, \{\boldsymbol{y}_i\}_{i=1}^{N}). \tag{23}$$

*Proof of Proposition C.2.* The proposition aims to characterize optimal parameters of Problem C.1. We begin by defining two sub-problems that are related to it, each focusing on minimizing one set of parameters while keeping the other set fixed:

$$\{\boldsymbol{W}^{(k)*}\} \;=\; \underset{\{\tilde{\boldsymbol{W}}^{(k)}\}}{\arg\min}\, G_{reg}(\delta, \{\tilde{\boldsymbol{W}}^{(k)}\}_{k=1}^{K}, \{\boldsymbol{\omega}^{(k)}\}_{k=1}^{K}, \{\boldsymbol{y}_i\}_{i=1}^{N}), \tag{372}$$

$$\{\boldsymbol{\omega}^{(k)*}\} \;=\; \underset{\{\tilde{\boldsymbol{\omega}}^{(k)}\}}{\arg\min}\, G_{reg}(\delta, \{\boldsymbol{W}^{(k)}\}_{k=1}^{K}, \{\tilde{\boldsymbol{\omega}}^{(k)}\}_{k=1}^{K}, \{\boldsymbol{y}_i\}_{i=1}^{N}). \tag{373}$$

where the parameters are limited to the constraints stated in Problem C.1. Interestingly, the suggested optimization problem in (373) can be rewritten by

$$\{\boldsymbol{W}^{(k)*}\} \;=\; \underset{\{\tilde{\boldsymbol{W}}^{(k)}\}}{\arg\min}\, G_{reg}(\delta, \{\tilde{\boldsymbol{W}}^{(k)}\}_{k=1}^{K}, \{\boldsymbol{\omega}^{(k)}\}_{k=1}^{K}, \{\boldsymbol{y}_i\}_{i=1}^{N} \tag{374}$$

$$= \underset{\{\tilde{\boldsymbol{W}}^{(k)}\}}{\arg\min} \, G(\{\tilde{\boldsymbol{W}}^{(k)}\}, \{\tilde{\boldsymbol{\omega}}^{(k)}\}, \{\boldsymbol{y}_i\}_{i=1}^N) + \delta \left( D\log(K) + \sum_{d=1}^D \sum_{k=1}^K \tilde{\omega}_d^{(k)} \log(\tilde{\omega}_d^{(k)}) \right) \quad (375)$$

$$= \underset{\{\tilde{\boldsymbol{W}}^{(k)}\}}{\arg\min} \, G(\{\tilde{\boldsymbol{W}}^{(k)}\}, \{\tilde{\boldsymbol{\omega}}^{(k)}\}, \{\boldsymbol{y}_i\}_{i=1}^N). \quad (376)$$

These two sub-problems are considered in Lemmas I.3 and I.11. Specifically, in Lemma I.3 the optimal graph matrices are derived in the form of

$$W_{i,j}^{(k)*} = \begin{cases} \dfrac{\exp\left(-\dfrac{\|\boldsymbol{y}_i - \boldsymbol{y}_j\|_{\tilde{\boldsymbol{\omega}}^{(k)}}^2}{\epsilon_{k,i}}\right)}{\sum_{t=1}^N \exp\left(-\dfrac{\|\boldsymbol{y}_i - \boldsymbol{y}_t\|_{\tilde{\boldsymbol{\omega}}^{(k)}}^2}{\epsilon_{k,i}}\right)} & \text{for } j \neq i \\ 0 & \text{else} \end{cases}, \quad (377)$$

$$(378)$$

for $i, j = 1, \ldots, N$ and $k = 1, \ldots, K$, where $\epsilon_{k,i}$ attains the minimum value that satisfies

$$\sum_{j=1}^N W_{i,j}^{(k)*} \log W_{i,j}^{(k)*} \leq -\log(\alpha). \quad (379)$$

On the other hand, in Lemma I.11 an optimal partitioning parameters are derived in the form of

$$\omega_d^{(k)} = \frac{\exp\left(-\sum_{i,j=1}^N W^{(k)} \left((\boldsymbol{y}_i)_d - (\boldsymbol{y}_j)_d\right)^2 / \delta\right)}{\sum_{\tilde{k}=1}^K \exp\left(-\sum_{i,j=1}^N W^{(\tilde{k})} \left((\boldsymbol{y}_i)_d - (\boldsymbol{y}_j)_d\right)^2 / \delta\right)} \quad (380)$$

for $d = 1, \ldots, D$ and $k = 1, \ldots, K$.

Therefore there exists parameters of this form that minimizes Problem C.1. $\quad\square$

**Proposition C.3** Let $\{\overline{\boldsymbol{\varpi}}^{(k)}\}_{k=1}^K \subset [0,1]^D$ be a soft uniform partitioning, i.e. $\overline{\varpi}_d^{(k)} = 1/K$, and let $\{\overline{\boldsymbol{W}}^{(k)}\}_{k=1}^K$ be the corresponding affinity matrices from Proposition C.2. Let $\{\boldsymbol{\omega}^{(k)}\}_{k=1}^K \subset \{0,1\}^D$ and $\{\boldsymbol{W}^{(k)}\}_{k=1}^K$ be the optimal partitioning solution as discussed in Proposition 3.4.

Define

$$\delta_{init} \equiv \frac{G(\{\overline{\boldsymbol{W}}^{(k)}\}_{k=1}^K, \{\overline{\boldsymbol{\varpi}}^{(k)}\}_{k=1}^K, \{\boldsymbol{y}_i\}_{i=1}^N)}{D \cdot \log(K)}. \quad (24)$$

Then, for any hard partitioning solution $\{\tilde{\omega}^{(k)}\}_{k=1}^K \subset \{0,1\}^D$ and any corresponding affinity matrices $\{\tilde{\boldsymbol{W}}^{(k)}\}_{k=1}^K$, we have:

$$G_{reg}(\delta_{init}, \{\overline{\boldsymbol{W}}^{(k)}\}_{k=1}^K, \{\overline{\boldsymbol{\varpi}}^{(k)}\}_{k=1}^K, \{\boldsymbol{y}_i\}_{i=1}^N) \leq G_{reg}(\delta_{init}, \{\tilde{\boldsymbol{W}}^{(k)}\}_{k=1}^K, \{\tilde{\omega}^{(k)}\}_{k=1}^K, \{\boldsymbol{y}_i\}_{i=1}^N) \quad (25)$$

*Proof of Proposition C.3.* We can see that the inequality stands based on the following derivation-

$$G_{reg}(\delta_{init}, \{\overline{\boldsymbol{W}}^{(k)}\}_{k=1}^K, \{\overline{\boldsymbol{\varpi}}^{(k)}\}_{k=1}^K, \{\boldsymbol{y}_i\}_{i=1}^N) \quad (381)$$

$$= G(\{\overline{\boldsymbol{W}}^{(k)}\}_{k=1}^K, \{\overline{\boldsymbol{\varpi}}^{(k)}\}_{k=1}^K, \{\boldsymbol{y}_i\}_{i=1}^N) + \delta_{init} \left( D\log(K) + \sum_{d=1}^D \sum_{k=1}^K \overline{\varpi}_d^{(k)} \log(\overline{\varpi}_d^{(k)}) \right) \quad (382)$$

$$= G(\{\overline{\boldsymbol{W}}^{(k)}\}_{k=1}^K, \{\overline{\boldsymbol{\varpi}}^{(k)}\}_{k=1}^K, \{\boldsymbol{y}_i\}_{i=1}^N) \quad (383)$$

$$= \delta_{init}(D \cdot \log(K)) \quad (384)$$

$$\leq \quad G(\{\tilde{\boldsymbol{W}}^{(k)}\}_{k=1}^{K}, \{\tilde{\boldsymbol{\omega}}^{(k)}\}_{k=1}^{K}, \{\boldsymbol{y}_i\}_{i=1}^{N}) + \delta_{init}(D \cdot \log(K)) \tag{385}$$

$$= \quad G(\{\tilde{\boldsymbol{W}}^{(k)}\}_{k=1}^{K}, \{\tilde{\boldsymbol{\omega}}^{(k)}\}_{k=1}^{K}, \{\boldsymbol{y}_i\}_{i=1}^{N}) + \delta_{init}\left(D \log(K) + \sum_{d=1}^{D}\sum_{k=1}^{K}\tilde{\omega}_d^{(k)}\log(\tilde{\omega}_d^{(k)})\right) \tag{386}$$

$$= \quad G_{reg}(\delta_{init}, \{\tilde{\boldsymbol{W}}^{(k)}\}_{k=1}^{K}, \{\tilde{\boldsymbol{\omega}}^{(k)}\}_{k=1}^{K}, \{\boldsymbol{y}_i\}_{i=1}^{N}), \tag{387}$$

where in the second derivation we introduce the identity $\sum_{k=1}^{K}\overline{\omega}_d^{(k)}\log\overline{\omega}_d^{(k)} = -\log(K)$ for all $d \in \{1,\ldots,D\}$, which holds by definition of the uniform partitioning. In the third derivation, we introduce the definition of $\delta_{init}$. Then, in the fourth derivation we use the non-negativity of $G$, along with the identity $\sum_{k=1}^{K}\tilde{\omega}_d^{(k)}\log\tilde{\omega}_d^{(k)} = 0$ for all $d \in \{1,\ldots,D\}$, which follows directly from the definition of a hard partition. $\qquad\square$

**Proposition C.4** Let the data consist of $N$ data points in $\mathbb{R}^D$. Then, the computational complexity of obtaining the partitioning weights $\{\boldsymbol{\omega}^{(k)*}\}_{k=1}^{K}$ and the affinity matrices $\{\boldsymbol{W}^{(k)*}\}_{k=1}^{K}$, as defined in Proposition C.2, is $O(KN^2D)$.

*Proof of Proposition C.4.* The feature partitioning weights, $\{\boldsymbol{\omega}^{(k)*}\}$, described in (21), are constructed based on:

$$\sum_{i,j=1}^{N} W_{i,j}^{(k)*}((\boldsymbol{y}_i)_d - (\boldsymbol{y}_j)_d)^2 \tag{388}$$

for every $d \in \{1,\ldots,D\}$ and $k \in \{1,\ldots,K\}$. The computational complexity of evaluating this quantity is $O(N^2)$ for a specific $d \in \{1,\ldots,D\}$ and $k \in \{1,\ldots,K\}$, and overall $O(N^2DK)$ for all. The remaining operations used to define the feature partitions —taking the exponential of this value and normalizing across partitions —- result in a computational complexity of $O(DK)$. Therefore, the overall computational complexity of this step is $O(N^2DK)$.

We now derive the computational complexity associated with computing the affinity matrices. These matrices are constructed based on the following weighted squared distances:

$$\|\boldsymbol{y}_i - \boldsymbol{y}_j\|_{\boldsymbol{\omega}^{(k)*}}^2 = \sum_{d=1}^{D}\omega_d^{(k)*}((\boldsymbol{y}_i)_d - (\boldsymbol{y}_j)_d)^2 \tag{389}$$

for all $i,j \in \{1,\ldots,N\}$ and $k \in \{1,\ldots,K\}$. The computational complexity of evaluating its value is $O(D)$ for a specific $i,j \in \{1,\ldots,N\}$ and $k \in \{1,\ldots,K\}$, and overall $O(N^2DK)$ for all. The remaining operations used to construct the affinity matrices involve taking the exponential value of each distance and normalizing across $j \in \{1,\ldots,N\}$, for every $i \in \{1,\ldots,N\}$ and $k \in \{1,\ldots,K\}$. These operations attain a computational complexity of $O(N^2K)$. Therefore, the computational complexity of this step is $O(N^2DK)$.

Now, by combining the computational complexities for both parameter updates results in $O(KN^2D)$. $\qquad\square$

**Proposition C.5** Let the data consist of $N$ data points in $\mathbb{R}^D$. Suppose the data is given in the form of a singular value decomposition (SVD) approximation of rank $S \ll N, D$. Then, the computational complexity of obtaining the partitioning weights $\{\boldsymbol{\omega}^{(k)*}\}_{k=1}^{K}$ and the affinity matrices $\{\boldsymbol{W}^{(k)*}\}_{k=1}^{K}$, as defined in Proposition C.2, is $O(K(S^2N^2 + S^2D))$.

*Proof of Proposition C.5.* Let $\boldsymbol{Y} \in \mathbb{R}^{N \times D}$ represent the data matrix, with the points embedded as rows. The SVD approximation is given by $\boldsymbol{Y} = \boldsymbol{U}\boldsymbol{E}\boldsymbol{V}^T$, where $\boldsymbol{U} \in \mathbb{R}^{N \times S}$ and $\boldsymbol{V} \in \mathbb{R}^{D \times S}$ are the left and right singular vector matrices, respectively, and $\boldsymbol{E} \in \mathbb{R}^{S \times S}$ is the diagonal matrix of the leading singular values.

The feature partitioning weights, $\{\boldsymbol{\omega}^{(k)*}\}$, described in (21), are constructed based on:

$$\sum_{i,j=1}^{N} W_{i,j}^{(k)*}((\boldsymbol{y}_i)_d - (\boldsymbol{y}_j)_d)^2 \tag{390}$$

for every $d \in \{1,\ldots,D\}$ and $k \in \{1,\ldots,K\}$. The remaining operations used to define the feature partitions —taking the exponential of this value and normalizing across partitions —- result in a computational complexity of $O(DK)$. In the next lines we will rewrite the equation above and compute the computational complexity associated with it.

We begin with rewriting the (390) by

$$\sum_{i,j=1}^{N} W_{i,j}^{(k)*}((\boldsymbol{y}_i)_d - (\boldsymbol{y}_j)_d)^2 \tag{391}$$

$$= \sum_{j=1}^{N}(\sum_{i=1}^{N} W_{i,j}^{(k)*})(\boldsymbol{y}_j)_d^2 + \sum_{i=1}^{N}(\sum_{j=1}^{N} W_{i,j}^{(k)*})(\boldsymbol{y}_i)_d^2 - 2\sum_{i,j=1}^{N} W_{i,j}^{(k)*}(\boldsymbol{y}_i)_d(\boldsymbol{y}_j)_d \tag{392}$$

$$= \sum_{j=1}^{N}(\sum_{i=1}^{N} W_{i,j}^{(k)*} + W_{j,i}^{(k)*})(\boldsymbol{y}_j)_d^2 - \sum_{i,j=1}^{N}(W_{i,j}^{(k)*} + W_{j,i}^{(k)*})(\boldsymbol{y}_i)_d(\boldsymbol{y}_j)_d \tag{393}$$

$$= (\boldsymbol{Y}^T(diag(\boldsymbol{W}^{(k)*}\boldsymbol{1} + (\boldsymbol{W}^{(k)*})^T\boldsymbol{1}))\boldsymbol{Y})_{d,d} - (\boldsymbol{Y}^T(\boldsymbol{W}^{(k)*} + (\boldsymbol{W}^{(k)*})^T)\boldsymbol{Y})_{d,d} \tag{394}$$

$$= (\boldsymbol{Y}^T\boldsymbol{L}^{(k)}\boldsymbol{Y})_{d,d} \tag{395}$$

where $\boldsymbol{L}^{(k)} \in \mathbb{R}^{N \times N}$ is defined by $\boldsymbol{L}^{(k)} = diag((\boldsymbol{W}^{(k)*} + (\boldsymbol{W}^{(k)*})^T)\boldsymbol{1}) - \boldsymbol{W}^{(k)*} - (\boldsymbol{W}^{(k)*})^T$ for any $k \in \{1, \dots, K\}$, the vector $\boldsymbol{1} \in \mathbb{R}^N$ is an all ones vector, and $diag : \mathbb{R}^N \to \mathbb{R}^{N \times N}$ generates a diagonal matrix from a given vector. The construction of $\boldsymbol{L}^{(k)}$ using $\boldsymbol{W}^{(k)}$ is $O(N^2)$.

We now incorporate the singular value decomposition (SVD) of $\boldsymbol{Y}$ into this expression and derive its computation complexity by analyzing the matrix multiplications involved:

$$(\boldsymbol{V}\boldsymbol{E}\boldsymbol{U}^T\boldsymbol{L}^{(k)}\boldsymbol{U}\boldsymbol{E}\boldsymbol{V}^T)_{d,d}. \tag{396}$$

Define $\boldsymbol{A}^{(k)} \equiv \boldsymbol{U}^T\boldsymbol{L}^{(k)}\boldsymbol{U} \in \mathbb{R}^{S \times S}$. The computational complexity of computing it is $O(SN^2 + S^2N)$ as $\boldsymbol{U} \in \mathbb{R}^{N \times S}$ and $\boldsymbol{L}^{(k)} \in \mathbb{R}^{N \times N}$. Next, define $\boldsymbol{B}^{(k)} \equiv \boldsymbol{E}\boldsymbol{A}^{(k)}\boldsymbol{E} \in \mathbb{R}^{S \times S}$. Since $\boldsymbol{E}$ is a diagonal matrix, the computational complexsity of this multiplication is $O(S^2)$. Therefore we are left with computing:

$$(\boldsymbol{V}\boldsymbol{B}^{(k)}\boldsymbol{V}^T)_{d,d}. \tag{397}$$

The computational complexity of evaluating each such element is $O(S^2)$ for each $d \in \{1, \dots, D\}$, and overall $O(S^2D)$.

Therefore, the computational complexity of evaluating (390) for a given $k \in \{1, \dots, K\}$ is $O(SN^2 + S^2D)$. Thus, we can conclude that the total cost deriving the feature partitioning weights is $O(K(SN^2 + S^2D) + DK) = O(K(SN^2 + S^2D))$.

We now turn to deriving the computational complexity of the affinity matrices. As described in (14), these matrices are constructed based on:

$$\|\boldsymbol{y}_i - \boldsymbol{y}_j\|_{\boldsymbol{\omega}^{(k)*}}^2 \tag{398}$$

for all $i, j \in \{1, \dots, N\}$ and $k \in \{1, \dots, K\}$. The remaining operations used to construct the affinity matrices involve taking the exponential value of each distance and normalizing across $j \in \{1, \dots, N\}$, for every $i \in \{1, \dots, N\}$ and $k \in \{1, \dots, K\}$. These operations attain a computational complexity of $O(N^2K)$. In the next lines we will rewrite (398) and derive its computational complexity.

We now incorporate the singular value decomposition (SVD) of $\boldsymbol{Y}$ into this expression and derive its computation complexity by analyzing the matrix multiplications involved:

$$\|\boldsymbol{y}_i - \boldsymbol{y}_j\|_{\boldsymbol{\omega}^{(k)*}}^2 \tag{399}$$

$$= (\boldsymbol{Y}diag(\boldsymbol{\omega}^{(k)*})\boldsymbol{Y}^T)_{i,i} + (\boldsymbol{Y}diag(\boldsymbol{\omega}^{(k)*})\boldsymbol{Y}^T)_{j,j} - 2(\boldsymbol{Y}diag(\boldsymbol{\omega}^{(k)*})\boldsymbol{Y}^T)_{i,j} \tag{400}$$

$$= (\boldsymbol{U}\boldsymbol{E}\boldsymbol{V}^T diag(\boldsymbol{\omega}^{(k)*})\boldsymbol{V}\boldsymbol{E}\boldsymbol{U}^T)_{i,i} + (\boldsymbol{U}\boldsymbol{E}\boldsymbol{V}^T diag(\boldsymbol{\omega}^{(k)*})\boldsymbol{V}\boldsymbol{E}\boldsymbol{U}^T)_{j,j} - 2(\boldsymbol{U}\boldsymbol{E}\boldsymbol{V}^T diag(\boldsymbol{\omega}^{(k)*})\boldsymbol{V}\boldsymbol{E}\boldsymbol{U}^T)_{i,j} \tag{401}$$

for all $i, j \in \{1, \dots, N\}$ and $k \in \{1, \dots, K\}$, where $diag : \mathbb{R}^D \to \mathbb{R}^{D \times D}$ generates a diagonal matrix from a given vector. Define $\boldsymbol{C}^{(k)} = \boldsymbol{V}^T diag(\boldsymbol{\omega}^{(k)*})\boldsymbol{V} \in \mathbb{R}^{S \times S}$. Its computational complexity is $O(S^2D)$ as $\boldsymbol{V} \in \mathbb{R}^{D \times S}$. Next, define $\boldsymbol{F}^{(k)} = \boldsymbol{E}\boldsymbol{C}^{(k)}\boldsymbol{E} \in \mathbb{R}^{S \times S}$. Since $\boldsymbol{E}$ is a diagonal matrix, the computational complexity of this multiplication is $O(S^2)$. Finally, the computational complexity of $(\boldsymbol{V}^T\boldsymbol{F}^{(k)}\boldsymbol{V})_{i,j}$ is $O(S^2)$, and the computational complexity of deriving it for all entrees is $O(S^2N^2)$ .

Therefore, the total computational complexity for evaluating (398) for all $i, j \in \{1, \ldots, N\}$ for a given $k \in \{1, \ldots, K\}$ is $O(S^2 N^2 + S^2 D)$. Thus, we can conclude that the total cost deriving the affinity matrices is $O(K(S^2 N^2 + S^2 D) + N^2 K) = O(K(S^2 N^2 + S^2 D))$. Now, by combining the computational complexities for both sets of parameters we get: $O(K(SN^2 + S^2 D) + K(S^2 N^2 + S^2 D)) = O(KS^2 N^2 + KS^2 D)$.

$\square$

