# OpenReview forum: "Partition First, Embed Later: Laplacian-Based Feature Partitioning for Refined Embedding and Visualization of High-Dimensional Data"
_ICML.cc/2025/Conference — ICML 2025 oral_

### Official Review · Reviewer_92va · 2025-03-09

**Overall Recommendation:** 3

**Summary:**

This paper claims when the data is complex and governed by multiple latent variables (which is almost always the case), the visualization methods that aim to capture all features in single lower dimensional space often: fail in disentangle the latent variables, or requires larger dimensionality to capture the full structure in the high dimensional data. To address this issue, this paper assumes the dataset is generated by mutually exclusive  manifolds each corresponds to a subset of features, and proposes to first partition the feature space into sub spaces, by minimizing Laplacian score of the partition and then perform DR w.r.t each subset of the features. The paper provided an extensive theoretical analysis of the partition problem and provided an alternating optimization algorithm that obtains soft assignment score which approximates the hard assignment solution. In the experiment section, the proposes approach is compared against multiple clustering methods (over features) on a synthetic dataset. Also, the approach is compared against tSNE on identifying biological processes in RNA sequencing data.

**Claims And Evidence:**

There are three main claims that the paper tries to establish:

* "Our approach generalizes traditional embedding and visualization techniques, allowing them to learn multiple embeddings simultaneously" This claim is supported by the empirical study in both experimental section and appendix.
* "We establish that if several independent or partially dependent manifolds are embedded in distinct feature subsets in high dimensional space, then our framework can reliably identify the correct subsets with theoretical guarantees." This claim is partially support by the objective function ( laplacian scores) as well as the theoretical analysis. However, in the proposed algorithm that finds the optimal solution of the objective function is an soft approximation of the underlying combinatorial assignment problem. The approximation quality is not prominently discussed, which undermines the "reliably identify" claim.
* "Finally, we demonstrate the effectiveness of our approach in extracting multiple low-dimensional structures and partially independent processes from both simulated and real data." This claim is partially supported by the partitioning experiment as well as the visualization experiment. However, there are no strong baselines used in the comparison, i.e., methods that also assume data consists of multiple manifold and performs DR/factorization accordingly. Without stronger baselines (see questions below), the effectiveness of the proposed approach in practice is difficult to assess.

**Essential References Not Discussed:**

Not to my knowledge.

**Experimental Designs Or Analyses:**

As mentioned above there are no strong baselines used in the comparison, i.e., methods that also assume data consists of multiple manifold and performs DR/factorization accordingly. Without stronger baselines (see questions below), the effectiveness of the proposed approach in practice is difficult to assess. Possible baselines are:

* He et al., Product manifold learning with independent coordinate selection.
* Kohli et al., Low distortion local eigenmaps
* van der Maarten and Hinton, Visualizing non-metric similarities in multiple maps

**Methods And Evaluation Criteria:**

* The proposed method is sensible. However, the approximation quality of the proposed algorithm is not clear in the main paper.
* The experiment settings are also sensible, however, also miss strong baselines.

**Other Comments Or Suggestions:**

N/A

**Other Strengths And Weaknesses:**

The theoretical oriented approach is appreciated, as many DR papers (objective function + optimization) did not provide in depth theoretical analysis.

**Questions For Authors:**

* Could you compare the proposed approach with stronger baseline mentioned above?
* Could you summarize and highlight the approximation quality of the proposed algorithm in the main paper?

**Relation To Broader Scientific Literature:**

The idea of applying partition using by minimizing Laplacian scores of the assignments then visualization is interesting in the DR literature. I'm not familiar with the high dimensional feature partition literature, so cannot comment in that respect.

**Theoretical Claims:**

I didn't went through the proofs.

---

> ### Author Rebuttal · Authors · 2025-04-01
>
> Q1. “Could you compare the proposed approach with stronger baseline mentioned above?”
>
> R1.To address the reviewer’s concerns, we conducted a comprehensive comparison of our approach with the methods [A], [B], and [C] using the biological dataset from Section 5.2 and the rotating figurines dataset in Appendix D. It will be incorporated into the text.
>
> We now explain our quantitative error measure for the biological dataset. To compute an error measure for a given embedding of the dataset, we do the following. For each point (cell) i and each of the two types of labels (cell phase and type), we find the k nearest neighbors of point i and compute the proportion of nearest neighbors whose label differs from the label of point i. This provides two scores (of label inconsistencies) for each point, corresponding to the two processes. We then average these scores across all data points, producing two error measures: one quantifying the inconsistency of the embedding with respect to the cell phase and the other with the cell type. For methods that provide a single embedding of the data, we average these two scores, quantifying how much this embedding is consistent with both latent processes. For methods that produce two embeddings, we expect each embedding to be consistent with only one of the latent processes. Therefore, in such cases, we assign to each embedding only one of its two scores (without repetition) such that the average of the assigned scores is minimized for the two embeddings.
>
> For the rotating figurines dataset in Appendix D, each image is determined by three latent variables, the rotation angles of the three figurines. To compute an error measure for a given embedding of the dataset, we do the following. For each data point i and for each one of the three rotation angles, we find the k nearest neighbors of point i and the k nearest angles of angle i, and compute the relative set difference between the two groups. This provides three scores of angle inconsistencies for each point (corresponding to the angles). The rest of the procedure to compute the final error measure is analogous to the case of the biological data described above.
>
> For each dataset, we computed the error measure for seven different methods: 1) t-SNE embedding using all features; 2) two t-SNE embeddings based on our partitions (‘tSNE FP’); 3) two embeddings of IC-PML; 4) a single embedding of LDLE; 5) two embeddings of Multi-tsne; 6) raw data features; and 7) raw data features after partitioning (‘FP’). For each method, we computed the error measure using different numbers of k nearest neighbors, where k = 2,4,6,...,50, resulting in a graph of the error measure as a function of k.
>
> The resulting performance graphs can be found in anonymous.4open.science/r/FP-70F4. It is evident that for both datasets, our proposed partitioning provides the smallest error measure across all values of k, either after the embedding with t-SNE or using the raw features. We highlight that for IC-PML, Multi-tsne, and LDLE, we tested a wide range of hyperparameter values and retained the configuration that provided the smallest error measure for each k. Details are provided in the link.
>
> [A] He et al., Product manifold learning with independent coordinate selection.
>
> [B] Kohli et al., Low distortion local eigenmaps
>
> [C] van der Maarten and Hinton, Visualizing non-metric similarities in multiple maps
>
> Q2. Could you summarize and highlight the approximation quality of the proposed algorithm in the main paper?
>
> Response. First, we would like to emphasize that our proposed algorithm (see Algorithm 1) does provide a solution to the hard partitioning problem (Problem 3.3). The next paragraph should make it clear.
>
> We now clarify the rationale behind our algorithm and its derivation. The analysis of our proposed optimization problem in Section 3 suggests a natural alternating minimization strategy, by alternating between minimizing the objective function over the graph parameters and minimizing over the feature partitions (see Eqs. (9)–(11)). Unfortunately, due to the binary nature of the feature partitions, this procedure is sensitive to the presence of local minima. To address this issue, we introduce a regularized variant of the objective function (see Problem C.1 in Appendix C) that produces a soft assignment of features instead of hard assignments into partitions. Our proposed algorithm solves several instances of the regularized optimization problem sequentially, each one with a reduced regularization parameter. In the final step, the regularization parameter reaches zero, hence effectively minimizing the exact unregularized problem. As demonstrated in Appendix C, this sequence of solutions to the regularized optimization problems is less likely to get stuck in a local minimum compared to solving the exact unregularized problem directly. To clarify this issue, we will add this explanation to the main text at the end of Section 3.

---

> > ### Comment · Reviewer_92va · 2025-04-08
> >
> > Thanks for the explanation and new comparison figures. I have updated my score.

---

### Official Review · Reviewer_E6QD · 2025-03-10

**Overall Recommendation:** 4

**Summary:**

High-dimensional data can sometimes be composed of multiple sets of features, each following a distinct substructure. Traditional visualization methods such as t-SNE, when applied to the full feature set, struggle to capture these substructures. The authors propose a method that enables feature space separation for improved visualization of individual feature sets. Both qualitative and quantitative results suggest that the proposed method outperforms existing baselines.

**Claims And Evidence:**

Most claims in the submission are clear and supported by evidence. However, the prevalence of substructure concatenation in real-world datasets beyond the simulated examples (rotated figurines/COIL-20) and single-cell transcriptomics is not thoroughly discussed. Additional insights into its occurrence across diverse applications would strengthen the motivation for the proposed method.

**Essential References Not Discussed:**

I'm not aware of any essential references that were not discussed.

**Experimental Designs Or Analyses:**

The experimental design seems legit, but the number of datasets examined is relatively limited. The main text evaluates only three datasets, two of which are simulated. The supplementary materials primarily contain additional simulated datasets based on object rotations. Expanding the evaluation to different modalities or another scRNA-seq dataset would help validate the broader applicability of the method.

Additionally, visualizing the separated datasets using methods beyond t-SNE could provide further insights into the effectiveness of the proposed approach.

**Methods And Evaluation Criteria:**

The proposed method and evaluation criteria are well-aligned with the problem at hand.

**Other Comments Or Suggestions:**

Algorithm 1, which is central to the proposed method, is only provided in the supplementary material. While this is likely due to space constraints, summarizing the approach in the main text would improve readability for the audience.

The authors should further emphasize the importance of the problem to strengthen the motivation for the method.

**Other Strengths And Weaknesses:**

The paper is well-written, and the figures are clear and visually appealing.

**Questions For Authors:**

Appendix G describes the procedure for determining the number of partitions in a dataset. However, the proposed elbow method may fail when the optimal $K$ is 1, potentially leading to spurious partitions. How do the authors mitigate this issue?

**Relation To Broader Scientific Literature:**

The proposed method addresses high-dimensional data visualization in cases where the dataset can be decomposed into multiple substructures. While this approach is likely valuable for computational biologists, its broader applicability to other fields may be limited (see my comments in Experimental Design and Analysis section).

**Theoretical Claims:**

The proof in Appendix I appears sound, with no major issues identified.

---

> ### Author Rebuttal · Authors · 2025-04-01
>
> We sincerely thank the reviewer for their thoughtful and detailed assessment of our work. We are pleased that the reviewer found our "proposed method and evaluation criteria well aligned with the problem at hand". We also appreciate the acknowledgement that "most claims in the submission are clear and supported by evidence".
>
> Q1(Q. for authors). Appendix G describes the procedure for determining the number of partitions in a dataset. However, the proposed elbow method may fail when the optimal K is 1, potentially leading to spurious partitions. How do the authors mitigate this issue?
>
> Response . We propose a suitable test to address this challenge in the second paragraph of page 20, Section G in the appendix. This test is designed to assess whether the data should be partitioned into two subsets (K=2). It compares the smoothness score (Eq 8) obtained from partitioning the data with an analogous score obtained from a randomly transformed version of the data, which mixes the features. This transformation simulates a scenario where no partition is possible. A significant difference between the two scores suggests that the data can be meaningfully partitioned.
>
> Response to other comments and concerns:
> 1.(Other Comments) Algorithm 1, which is central to the proposed method, is only provided in the supplementary material. While this is likely due to space constraints, summarizing the approach in the main text would improve readability for the audience.
>
> Response. We agree with the reviewer, it was indeed excluded due to space constraints in the initial submission. We will add such a summary at the end of Section 3.2. We refer the reviewer to our response to Q2 of reviewer 92va.
>
> 2.(Other Comments) “The authors should further emphasize the importance of the problem to strengthen the motivation for the method”.
>
> Response. We thank the reviewer for this comment. To strengthen the motivation for our method, we will add the following discussion to the introduction section (specifically in page 2, between the first and second paragraphs in the left column):
> “The setting where distinct feature subsets of the data may contain unique geometric structures is widespread in applications. For example, in hyperspectral imaging, different feature groups correspond to different wavelengths, which capture distinct chemical or physical phenomena of the observed materials or environment [A, B]. Similarly, in astrophysics, different spectral bands of electromagnetic radiation serve as feature groups in the data, capturing distinct astrophysical phenomena such as interstellar extinction, fast radio bursts, and gravitational waves [C,D]. In cellular biology and genomics, different groups of genes may be associated with distinct cellular processes [E,F], as we exemplify in Section 5.2.”\
> [A] Khan, M., et al. "Modern trends in hyperspectral image analysis: A review," in Ieee Access, vol. 6, pp. 14118–14129, 2018.\
> [B] Lu, B., et al. "Recent advances of hyperspectral imaging technology and applications in agriculture," in Remote Sensing, vol. 12, no. 16, pp. 2659, 2020.\
> [C] Indebetouw, R., et al. "The wavelength dependence of interstellar extinction from 1.25 to 8.0 $μ$m using GLIMPSE data," in The Astrophysical Journal, vol. 619, no. 2, pp. 931, 2005.\
> [D] Burke-Spolaor, S., et al. "The astrophysics of nanohertz gravitational waves," in The Astronomy and astrophysics review, vol. 27, pp. 1–78, 2019.\
> [E] Sastry, A., et al. "The Escherichia coli transcriptome mostly consists of independently regulated modules," in Nature communications, vol. 10, no. 1, pp. 5536, 2019.\
> [F] Kotliar, D., et al. "Identifying gene expression programs of cell-type identity and cellular activity with single-cell RNA-Seq," in Elife, vol. 8, pp. e43803, 2019.
>
> 3.(Experimental Design)  The experimental design seems legit, but the number of datasets examined is relatively limited. The main text evaluates only three datasets, two of which are simulated. The supplementary materials primarily contain additional simulated datasets based on object rotations. Expanding the evaluation to different modalities or another scRNA-seq dataset would help validate the broader applicability of the method.
>
> Response.  We refer the reviewer for the Response for (Other Streng. And Weak) to reviewer zz3ii. Additionally, during the revision period, we commit to enhancing our manuscript by incorporating analyses using datasets from the paper
>
> Qu, R., et al. "Gene trajectory inference for single-cell data by optimal transport metrics," in Nature Biotechnology, pp. 1–11, 2024.
>
> 4.(Experimental Design) Visualizing the separated datasets using methods beyond tSNE could provide further insights into the effectiveness of the proposed approach.
>
> Response. To address the reviewer’s concern, we will include UMAP and Diffusion Maps embeddings in the Appendix for additional visualization and analysis of the separated datasets. The images can be found in https://anonymous.4open.science/r/FP-70F4.

---

> > ### Comment · Reviewer_E6QD · 2025-04-04
> >
> > This response has well addressed my concerns and I have increased my rating to 4.

---

### Official Review · Reviewer_2zX4 · 2025-03-12

**Overall Recommendation:** 4

**Summary:**

The authors propose an approach for embedding high dimensional data via partitioning features using a Laplacian smoothness optimization. This improves over classical techniques for embedding where extreme dimension reduction can distort results. They provide theoretical results characterizing the solution of their stated optimization problem as well as related asymptotic analysis. They provided experiments that examine the efficacy of their approach in real data.

**Claims And Evidence:**

Yes. Theory is sound to the best of my knowledge, and experiments are rather comprehensive.

**Essential References Not Discussed:**

To the best of my knowledge, the literature and prior work that are essential have been cited in this paper.

**Experimental Designs Or Analyses:**

Given that the approach proposed is unsupervised in nature, it is indeed challenging to validate. The authors did a good job with the simulations in Table 1 where they compared their algorithm with others under a simulated setting where partitioning error can be measured.

**Methods And Evaluation Criteria:**

The comparisons provided in table 1 and in appendix section G are sensible.

**Other Comments Or Suggestions:**

N/A

**Other Strengths And Weaknesses:**

originality: The approach of partitioning features is original, to the best of my knowledge, regarding dimensionality reduction.

clarity: The paper is written clearly. The material is naturally challenging, but the authors did a good job of exposition.

significance: given the broad applications of dimensionality reduction, I find the paper's contribution sufficiently impactful and significant.

**Questions For Authors:**

N/A

**Relation To Broader Scientific Literature:**

In terms of broader science, there is a large literature in machine learning on manifold learning and dimension reduction. In biology, such methods are often used to analyze sequencing data to uncover new cell types etc.

**Theoretical Claims:**

I did not check the proofs of the theorem that are in the appendix/supplement. However, to the best of my knowledge, the theorems appear sound and I did not find any mistakes in the theorems in the main text.

---

> ### Author Rebuttal · Authors · 2025-04-01
>
> We thank the reviewer for your thoughtful and positive review of our paper. We are pleased that the reviewer found our approach ‘original’ and that they considered the paper’s contribution to be “sufficiently impactful and significant”. Additionally, we appreciate the reviewer's recognition of the “clarity” with which the paper is written, as well as their acknowledgment that we did a “good job”  with the simulations in Table 1 to compare our results with other algorithms.

---

### Official Review · Reviewer_zz3i · 2025-03-14

**Overall Recommendation:** 4

**Summary:**

The manuscript proposes a new dimensionality reduction method, targeting the case where data feature originates from K sets, which are either independent or low-dependent. The method is composed of two steps. In the first, a decomposition of the data features into K disjoint sets is identified; in the second, a classic dimensionality reduction method is applied to each of the K subsets. The decomposition generalises upon a common first step in existing methods, which uncover a graph structure ("Laplacian") from the data disimilarity matrix. Here, both the decomposition into K sets and the smoothness objectives are co-optimised, achieving better smoothness than the original frameworks. The method is demonstrated to be brilliant on synthetic data, which was created by the assumed generative model, and superior to previous approaches on certain real-world problems.

**Claims And Evidence:**

As far as I could see, all claims are supported by clear and convincing evidence.

**Essential References Not Discussed:**

None that I could see.

**Experimental Designs Or Analyses:**

The methods' demonstration in terms of visualisation is straightforward, while other evaluations of the experimental results are barely done (beyond what is shown in Table 2).

**Methods And Evaluation Criteria:**

Evaluation criteria are hard for dimensionality reduction methods, and the proposed method might be superior in certain cases and inferior in others. The manuscript does not compare the new method with previous ones using the subfield common (albeit not well justified).

**Other Comments Or Suggestions:**

My score is only "weak accept" rather than "accept" due to the limited evaluation of the method.

**Other Strengths And Weaknesses:**

Strengths
 * Clear motivation and superb depication of the inner working of existing methods.
 * Very nice proposal of the objective function we wish we could solve and several relaxations toward an objective we can solve.
 * Proofs for the convergence of the algorithm to the correct solution in certain cases.

Weaknesses
 * The results are mostly aesthetic and subjective rather than through improvement of a previously-proposed benchmark (beyond the results in Table 2 which are quite minimal).
 * Only a single real-world example is presented in the main test and another in the appendices.

**Questions For Authors:**

* Does your method help mitigate the criticism of "The specious art of single-cell genomics" (which you nicely cite as motivation for your method)? Or does the criticism equally apply to your method as well?
 * Can you offer an objective criteria (a test) for when your method is expected to perform better than previous ones? Obviously, for the correct data generative model, this is the case, but can you offer some guidance for someone running the method on real-world data?
 * What are the limitations of the proposed algorithm on data which does not satisfy the assumptions (e.g. K=1)? In what sense would your approximation degrade the results of classic dimensionality reduction methods?

**Relation To Broader Scientific Literature:**

The manuscript provides a great literature review of both dimenionality reduction methods and graph decomposition methods.

**Theoretical Claims:**

The theoretical claims are beautifully supported with clear explanations and detailed proofs.

---

> ### Author Rebuttal · Authors · 2025-04-01
>
> We would like to thank the reviewer for their time and thoughtful feedback. We appreciate that the reviewer found our claims to be "supported by clear and convincing evidence" and that our "theoretical claims are beautifully supported with clear explanations and detailed proofs."
>
> Q1. Does your method help mitigate the criticism of "The specious art of single-cell genomics" (which you nicely cite as motivation for your method)? Or does the criticism equally apply to your method as well?
>
> R1. Our approach addresses some of the criticisms outlined in "The specious art of single-cell genomics" but not all of them. Specifically, substructures in the data that have low intrinsic dimensions could be embedded more accurately after our proposed partitioning. This advantage is demonstrated in our simulations and experiments. However, even if our approach successfully partitions the features into groups representing distinct substructures, there is no guarantee that these substructures can be accurately visualized in 2 or 3 dimensions. Nonetheless, even if the visualization fails, the partitioning obtained by our approach can be used for many other analytical tasks beyond visualization. Hence, the core of our approach, which partitions the data into simpler structures (regardless of visualization), is less susceptible to the criticism of "The specious art of single-cell genomics". To clarify this issue in the text, we plan to add this explanation to the discussion section.
>
> Q2. Can you offer an objective criteria (a test) for when your method is expected to perform better than previous ones? Obviously, for the correct data generative model, this is the case, but can you offer some guidance for someone running the method on real-world data?
>
> Response. We refer the reviewer to the second paragraph of page 20 in appendix G.
>
> Q3. What are the limitations of the proposed algorithm on data which does not satisfy the assumptions (e.g. K=1)? In what sense would your approximation degrade the results of classic dimensionality reduction methods?
>
> Response. In cases where the data consists of a single smooth structure and is partitioned into K>1 feature groups, we expect that the partitions will each capture a structure similar to the original data, at least in the model we analyzed in Section 4. This will result in redundant embeddings that are similar to an embedding using all features. In Section G of the appendix, we explain it in more detail, along with other related scenarios of over-selecting or under-selecting K.
>
> *Response to other comments and concerns:*
>
> (Experimental Design). The methods' demonstration in terms of visualization is straightforward, while other evaluations of the experimental results are barely done (beyond what is shown in Table 2).
>
> Response: We refer the reviewer to our response to Question 1 from reviewer 92va.
>
> (Other Streng. And Weak.). Only a single real-world example is presented in the main test and another in the appendices.
>
> Response. Based on the reviewer comment, we conducted a new experiment using a liver scRNA-seq dataset [A] that contains two biological organizing factors: the circadian cycle process and cellular zonation (spatial organization of cells in liver layers). The data consists of 6,889 cells, where each cell is annotated by the circadian time at which it was captured (ZT00, ZT06, ZT12, ZT18), which follows a cyclic pattern, and its location within the liver (Layer 1-8), where cells from adjacent layers share biological relationships. We apply standard preprocessing on the data before applying our partitioning approach. A comparison of the traditional tSNE embedding with tSNE embeddings based on the extracted partitions is provided in https://anonymous.4open.science/r/FP-70F4/
>
> The comparison includes:
> * liver_circaidan_PCA10 - This figure demonstrates how the circadian process is reflected across different embeddings.
> * liver_layer_PCA10- This figure illustrates how the zonation is captured in the embeddings, by overlaying each layer subpopulation separately on top of the entire embedding.
> * Note: The "PCA10" indicates that we used PCA with ten dimensions within the preprocessing. Additional files ending with ``PCA20’’ are included as well, and in these we used PCA with twenty dimensions.
>
> To conclude, the traditional tSNE embedding provides a single visualization where both processes are shown together. While the embedding contains four clusters, the cyclic structure defining the circadian cycle is less evident. However, the embedding based on Partition 2 does reveal both clusters and the cyclic structure. Additionally, while zonation is visible in the traditional t-SNE, it appears less distinct compared to the clearer and more progressive representation of zonation in the embedding based on Partition 1, which aligns better with the zonation layers.
>
> [A] Droin, C. et al. Space-time logic of liver gene expression at sub-lobular scale. Nature metabolism 3, 43-58 (2021).

---

> > ### Comment · Reviewer_zz3i · 2025-04-08
> >
> > After reading the authors' rebuttal and other reviewers' comments, I believe there is a consensus on the validity of the work, with the main criticism (raised by reviewer 92va and myself) referring to a more comprehensive evaluation against strong baselines. This issue was reasonably addressed by the authors (especially considering time constraints), so I will raise my score accordingly.

---

### Decision · Program_Chairs · 2025-05-01

**Decision:**

Accept (oral)

**Comment:**

This paper proposed to embed high-dimensional data via partitioning features using Laplacian smoothness optimization. It improves classical techniques and provides theoretical results of optimization and related asymptotic analysis. Reviewers praised that the proposed approach is origin, the paper is well written, and the theoretical results are supported with clear explanations and detailed proofs. Reviewers also raised concerns regarding the validation of the proposed work and the selected datasets. Authors’ responses to these concerns are satisfactory. Overall, this is a high-quality work, so the strong acceptance is recommended.